# Upper tropospheric CH$_4$ and CO affected by the South Asian summer monsoon during OMO

Laura Tomsche[1], Andrea Pozzer[1], Narendra Ojha[1], Uwe Parchatka[1], Jos Lelieveld[1], Horst Fischer[1]

[1]Department of Atmospheric Chemistry, Max-Planck-Institute for Chemistry, Mainz, 55128, Germany

5 *Correspondence to*: Laura Tomsche (laura.tomsche@mpic.de)

**Abstract.** The Asian monsoon anticyclone (AMA) is a yearly recurring phenomenon in the northern hemispheric upper troposphere and lower stratosphere. It is part of the South Asian summer monsoon system, and it has a clearly observable signature due to vertical transport of polluted air masses from the surface to the upper troposphere by the monsoon convection. We performed in situ measurements of carbon monoxide (CO) and methane (CH$_4$) in the region of monsoon outflow and in 10 background air in the upper troposphere (Mediterranean, Arabian Peninsula, Arabian Sea) by optical absorption spectroscopy on board the High Altitude and Long range (HALO) research aircraft during the OMO (Oxidation Mechanism Observations) mission in summer 2015. We identified the transport pathways and the origin of the trace gases with back trajectories, calculated with the Lagrangian particle dispersion model FLEXPART, and we compared the in situ data with simulations of the atmospheric chemistry general circulation model EMAC. CH$_4$ and CO mixing ratios were found to be enhanced within the 15 AMA, the in situ data increased on average by 72.1 ppbv and 20.1 ppbv, respectively, originating in the South Asian region (Indio-Gangetic Plain, North East India, Bangladesh and Bay of Bengal). It appears that CH$_4$ is an ideal monsoon tracer in the upper troposphere due to its extended lifetime and the strong South Asian emissions. Furthermore, we used the measurements and model results to study the dynamics of the AMA over several weeks during the monsoon season, with an emphasis on the southern and western areas in the upper troposphere. We distinguished four AMA modes based on different meteorological 20 conditions. During one occasion we observed that under the influence of dwindling flow the transport barrier between the anticyclone and its surroundings weakened, expelling air masses from the AMA. The trace gases exhibited a distinct fingerprint of the AMA, and we also found that CH$_4$ accumulated over the course of the OMO campaign.

## 1 Introduction

25 The Asian monsoon anticyclone (AMA) is an annual, large-scale weather phenomenon in the upper troposphere and lower stratosphere during the boreal summer. It is enclosed by the westerly subtropical jet in the north and the easterly jet in the south and extends over southern Asia and the Middle East up to the Mediterranean. It is formed by diabatic heating in the South Asian monsoon region (Gill, 1980, Hoskins and Rodwell, 1995). The anticyclone is a strong and nearly closed circulation system, which is variable in strength and location (Hsu and Plumb, 2000, Popovic and Plumb, 2001, Garny and Randel, 2013,

Ploeger et al., 2015). The strong winds at its edges act as transport barrier for chemical constituents in the upper troposphere. Stratospheric tracers, like ozone, show generally lower concentrations inside the AMA than outside (Park et al., 2008, Randel and Park, 2006). Tropospheric tracers, like CO and $CH_4$, are uplifted to the upper troposphere by the strong monsoon convection. These chemical constituents can be trapped in the anticyclone, change the atmospheric chemistry in the upper

troposphere and lower stratosphere and clearly signify the monsoon influence (Park et al., 2007). The signature of the anticyclone has been identified from different measurement platforms, like satellites and aircrafts. Airborne measurements are rare and limited in time and space but resolve small scales. For example, the in-service airborne projects CARIBIC (Civil Aircraft for the Regular Investigation of the atmosphere Based on an Instrument Container; e.g. Schuck et al., 2012, Rauthe-Schöch et al., 2016) and IAGOS-MOZAIC (IAGOS (In-service Aircraft for a Global Observing System) and MOZAIC

(Measurements of OZone by Airbus In-service aircraft); Barret et al., 2016, Dethof et al., 1999) reported trace gas measurements in the Asian monsoon region. In addition aircraft campaigns investigated the Asian monsoon during the aircraft campaign MINOS (Lelieveld et al., 2002, Scheeren et al., 2003), the Earth System Model Validation (ESMVal) campaign (Gottschaldt et al., 2017), and the Transport and Composition in the Upper Troposphere and Lowermost Stratosphere (TACTS) campaign (Vogel et al., 2014). In contrast, satellite data cover a larger spatial area and can be used for long term measurements,

nevertheless they are limited to their overpassing track and they have a coarse resolution. The obscured view from clouds during the South Asian monsoon additionally restricts the satellite view (e. g. Ojha et al., 2016), which requires long-term averaging in time and should be complemented by in situ measurements. Satellite data for different trace gases, like $H_2O$ (Park et al., 2004, Randel and Park, 2006), CO (Li et al., 2005, Park et al, 2008) and $CH_4$ (Park et al., 2004), show the vertical and horizontal extension of the AMA and are generally in agreement with model simulations (e.g. Pan et al., 2016, Nützel et al.,

2016, Bergman et al., 2013). To improve model outputs and satellite data retrievals, airborne measurements are necessary.

A more physically motivated criterion to distinguish between the AMA and its surrounding in the upper troposphere is the potential vorticity (PV) (e.g. Ploeger et al., 2015, Garny and Randel, 2013). In the anticyclone PV values on isentropic surfaces are lower than outside. Therefore, a maximum in the PV gradient can be used to identify the horizontal transport barrier associated with the AMA. However, applying the PV criterion is not straightforward since PV values in the AMA increase

during the monsoon season and decrease from the extra-tropics towards the tropics, which limits its usefulness. Nevertheless, it is quite helpful in combination with trace gas distributions from in situ and satellite measurements.

During the aircraft campaign MINOS the outflow of the AMA was investigated over the eastern Mediterranean basin (Lelieveld et al., 2002, Scheeren et al., 2003), while during the Earth System Model Validation (ESMVal) campaign a single flight was performed from Male/Maldives to Larnaca/Cyprus in September 2012 that intersected the AMA at an altitude of

150 hPa (Gottschaldt et al., 2017). In situ airborne measurements in the region of the Mediterranean, the Arabian Peninsula, and the Arabian Sea during the monsoon season are still limited, even though the AMA impacts these regions either by its extension or via outflow. Here we present results from an aircraft mission, which focuses on the AMA between the Indian

Ocean and the Mediterranean. The measurement campaign OMO (Oxidation Mechanism Observations) took place in July/August 2015 with the High Altitude and Long range (HALO) research Aircraft, performing flights at altitudes between 11 km and 15 km over the above-mentioned regions to investigate the dynamics and atmospheric chemistry in the upper troposphere over five weeks during the monsoon season.

The present study focuses on the measurements of $CH_4$ and CO, which document long-distance transport of air pollution, as these species have extended lifetimes of 8-9 years ($CH_4$, Lelieveld et al., 2016) and 2-3 months (CO, Xiao et al., 2007). These trace gases can be used to identify emission sources from the surface as they are co-emitted with other pollutants. They have both natural and anthropogenic sources. Major CO sources are anthropogenic and emitted via combustion processes of fossil fuel, biomass, and domestic fuel. Its natural sources are mainly from vegetation and oceans, but they are minor (Pandis and

Seinfeld, 2006). $CH_4$ is also emitted by combustion of fossil fuel and biomass (Khalil, 2000). Further sources are rice cultivation and ruminants, but also swamps and flood areas. For wetlands, the uncertainty in $CH_4$ emissions is still a large concern in atmospheric chemical transport models (Bloom et al., 2017, and references there in). In South Asia anthropogenic emissions increase with a growing population and economic development (Rauthe-Schöch et al., 2016, Ohara et al., 2007). The observations of $CH_4$ and CO show zonal and meridional concentration gradients as well as vertical gradients in the upper

troposphere, allowing to investigate the extent of the AMA. In order to differentiate background from AMA influenced air masses, we derived a $CH_4$ based threshold. Further, we compared our observations with EMAC model simulations, which extend the view on the trace gas distribution from a regional (along the flight tracks) to a global scale. To study the transport pathways we calculated back trajectories with the Lagrangian particle dispersion model FLEXPART along the flight tracks. With FLEXPART we gained a more detailed insight into the dynamics. We compared the back trajectories with observations

of $CH_4$ and CO to distinguish between different transport pathways. Thus we also studied the origin of emissions within South Asia. Finally, we investigated the variability of the AMA over several weeks as the anticyclone changes its position, extent, and strength due to the monsoon dynamics.

## 2 Methods

### 2.1 OMO campaign

The Oxidation Mechanism Observation (OMO) aircraft measurement campaign focused on the self-cleaning capacity of the atmosphere in connection with the Indian summer monsoon. The mission took place in July and August 2015 with flight tracks in the upper troposphere (10-15 km) over the Mediterranean, the Arabian Peninsula, and the Indian Ocean (Figure 1). In South Asia the pollution emissions are growing and during the monsoon season they are uplifted into the upper troposphere. The pollution is partly removed by wet deposition or transformation into soluble gases, or they are involved in air chemistry and

transported downwind of the sources. For a broad analysis of the efficiency of the self-cleaning mechanism a large variety of

chemical compounds, like $CH_4$, CO, OH, $HO_2$, NOy, $SO_2$, $RO_2$, $H_2O_2$, and total peroxides, were measured during the multi-institutional campaign, involving the Max-Planck-Institute for Chemistry, Mainz, the Research Centre Jülich, the German Aerospace Centre, the Research Centre Karlsruhe, and the universities of Bremen, Heidelberg, and Wuppertal. The main objectives were the oxidation processes and free radical chemistry, the efficiency of convective cloud transport and wet

deposition, as well as long-distance transport of air pollution and impacts on air quality and climate change. The OMO mission comprised 111 flight hours during 17 flights. HALO was based alternately at Paphos (Cyprus) and on Gan (Maldives) with refuelling stops at the airport of Bahrain. Further information about OMO can be found in Lelieveld et al. (2018) and on the webpage http://www.halo.dlr.de/science/missions/omo/omo.html.

**2.2 Trace gas measurements**

We employed the TRISTAR instrument (Tracer In Situ TDLAS for Atmospheric Research, and TDLAS is Tunable Diode Laser Absorption Spectrometry), which is an IR-quantum cascade laser absorption spectrometer for airborne measurements of trace gases (CO, $CH_4$, HCHO) on board HALO with a compact and robust design, consisting of an optical set-up and the instrument electronics. TRISTAR was described in more detail in previous publications (Tadic et al., 2017, Schiller et al., 2008

and references therein). The electronic part, including laser controller, data acquisition, etc., is integrated into the upper part of half a 19" rack. Mounted on the top of the rack is the optical set-up consisting of a liquid nitrogen cryostat, which houses three infrared quantum cascade lasers for CO, $CH_4$, and HCHO and two cryogenic photovoltaic mercury-cadmium-telluride detectors, a double corner cube multi pass cell according to White (1976), and several mirrors to reflect and collimate the beam. The optical unit is fixed on shock mounts for protection against vibrations. The trace gases are detected sequentially via

pneumatically driven pop-up mirrors.

A detailed description of the electronic set-up is given in Schiller et al. (2008). With a 66 ms saw-tooth current ramp the laser emission is scanned across a rotational-vibrational absorption line of the target species. In addition, the frequency of the laser is modulated via its injection current with a sinusoidal shaped 20 kHz frequency. At the detector the signal is demodulated at 40 kHz by a lock-in amplifier. For CO and $CH_4$ two measurement modes are used: ambient air and in-flight calibration. The

in-flight calibrations are realized with a secondary standard from pressurized bottles (6 l bottle, Auer GmbH, Germany), calibrated against certified reference gases. CO is calibrated against a reference gas (121.44±1.46 ppbv), which is calibrated against a secondary standard (155.8±0.45 ppbv). The latter was measured against a dilution gas (10ppm±1%) from EMPA (Swiss Federal Laboratories for Materials Science and Technology) referring to a NIST standard. $CH_4$ is calibrated against a working standard based on the scale of NOAA-2004 by Dlugokencky et al. (2005) and has an uncertainty of +-0.3 ppbv relating

to the CMDL83-standard (Dlugokencky et al., 2005). Furthermore the in situ CO and $CH_4$ data are drift corrected by interpolation between regular in-flight calibrations (Tadic et al., 2017).

Under the assumption of a Gaussian error propagation the total uncertainty consists of the statistical error (noise and drift correction) and the systematical error (calibration to reference gases). The total campaign average uncertainties are 5.1% and 0.275% for CO and $CH_4$, respectively. During OMO the CO accuracy degraded, because of problems with the CO laser in the second half of the mission. A detailed overview of the total uncertainties for all the flights is presented in the supplement Table S1.

## 2.3 FLEXPART back trajectories

The origin of air masses was derived with the Lagrangian particle dispersion model FLEXPART Version 9.2 beta (Stohl et al., 1998). The model is driven by ECMWF (European Centre for Medium-Range Forecasts) operational data with a horizontal resolution of 1°x1° and a vertical resolution of 137 levels between 1013.25 hPa and 0.01 hPa. The temporal resolution is 3 h, with analyses at 00, 06, 12, 18 UTC and forecasts for 03, 09, 15, 21 UTC. FLEXPART accounts for turbulence using the mean wind plus turbulent fluctuations and also mesoscale wind fluctuations (Stohl et al., 2010). The planetary boundary height is parameterized following the concept of Vogelezang and Holtslag (1996) using the critical Richardson number (Stohl et al., 2010). Vertical transport is calculated by using the Langevin equation (Thomson, 1987), which takes into account the turbulent vertical wind and its standard deviation. It includes also a decrease of air density with height. Additional moist convection is parameterized according to Emanuel and Zivkovic-Rothman (1999). Their parametrization builds on temperature and humidity fields to provide mass flux information (Stohl et al., 2005). Trajectories are started every 10 min along the flight tracks for air parcels, neglecting loss processes due to deposition or chemical reactions. The trajectories are calculated 10 days back in time for 10000 parcels that are initialized per release point (size: 1°x1°x500 m and 1 hour). The model output is a dispersion field, which consists of several parameters, i.e. geographical position, planetary boundary layer (PBL) height, and temperature, for each parcel per 3 h interval. The amount of data can be condensed via cluster analyses according to Stohl et al. (2002). These cluster trajectories are called centroid trajectories. They are comparable to traditional trajectories, but include contributions of turbulence and convection via the centroid of all particles per time step.

## 2.4 EMAC model data

The ECHAM/MESSy Atmospheric Chemistry (EMAC) model consists of the general circulation model ECHAM5 (fifth generation of the European Center HAMburg, Roeckener et al, 2006) and the Modular Earth Submodel System (MESSy, Jöckel et al. 2005, 2010), which extends the model into a fully coupled chemistry climate model. The horizontal resolution applied is 2.8°x2.8°, the vertical resolution is determined by 90 layers on a hybrid pressure grid between the surface and 0.01 hPa. The EMAC model was not run in an offline CTM mode, as the radiation calculations were based on simulated GHGs

concentrations. The model was weakly nudged towards ECMWF ERA Interim data (Jeuken et al., 1996) and therefore reproduced very similar dynamics to the ECMWF model (although not binary identical). The simulation is an extension of simulation RC1SD-base-10 (Jöckel et al. 2016) so to cover the full OMO campaign. Few changes to the original simulation have been applied (i.e. increased South Asia $SO_2$ emissions and reduced lightning $NO_x$), as described in Lelieveld et al. (2018).

Although the simulation is the continuation of a well evaluated experiment, the simulation was running from March 1st, 2015 so to give time to the $SO_2$ and $NO_x$ to balance to the new emissions (i.e. 4 months spin up time). Only the data from July and August 2015, which covers the field campaign is actually used. The EMAC model is a hydrostatic model and the convective transport is parameterized (Ouwersloot et al. 2015, Tost et al. 2006). Indication of the vertical transport time in EMAC can be found in Krol et al. (2018), where also a comparison with model of similar complexity is shown. The emissions are based on

the Representative Concentration Pathways (RCP) 8.5 for anthropogenic activity (Van Vuuren et al., 2011) and Global Fire Emissions Database (GFED) v3.1 for biomass burning emission of 2015 (Van der Werf et al., 2010). For methane additional sources of both wetlands in the Amazon and North American shale gas drilling were added to simulate the methane trend since 2007 (Zimmermann et al., 2018).

We used two different model output. One is the output from the SD4 submodel, which was developed by Jöckel et al. (2010)
for simulations along moving platforms, like ships or aircrafts. The data collection takes place in four dimensions (space and time) and the data are interpolated online, thus no information is lost due to interpolation after the simulation. The data along the flight track have a time resolution of 12 min (i.e. the model time-step), which are compared to the in situ data for CO and $CH_4$, averaged over 12 min. The second model output is given as three-dimensional daily mean data for different parameters, like CO, $CH_4$ and the wind field. With these data the position of the AMA can be identified on different pressure levels, as well
as its vertical extension, via vertical profiles. Additionally, the identification of emission sources at the surface is possible.

## 2.5 Satellite data

Cloud top pressure information is used as a proxy for convection. We compared the location of the convective clouds with the location of the uplift of the back trajectories simulated by FLEXPART. The cloud top pressure data are collected from the MODIS instrument on board of AQUA, via measured radiances in the spectral absorption bands at 15 µm $CO_2$ (Menzel et al.,
2008). In general, the atmosphere becomes more opaque with increasing wavelength due to the absorption of $CO_2$ between 13.3-15 µm. Thus, the measured radiances in these spectral bands are sensitive to different pressure levels. The cloud top pressure is determined by the ratio of two pairs of adjacent wavelengths in the infrared. For AQUA MODIS the ratios of 14.24/13.94 µm, 13.94/13.64 µm and 13.64/13.34 µm are used for high, midlevel, and low-level clouds, respectively. The data are derived from the level 3 MODIS Atmosphere Daily Global Product data (MYD08D3, Platnick, 2015), which are available
from the NASA webpage (https://ladsweb.modaps.eosdis.nasa.gov/). The resolution of the data is 1°x1° for daily means.

## 3 Results and Discussion

### 3.1 CH$_4$ and CO profiles

Vertical profiles were flown over Oberpfaffenhofen (Germany), Paphos (Cyprus), Etna (Italy), Egypt, Bahrain, and Gan (Maldives) (marked in Figure 1). As observed, the CO and CH$_4$ profiles measured during OMO indicate different altitude distributions depending on the geographical location and partly also on the meteorological situation, especially for Paphos and Egypt. Profiles over Egypt were measured when the AMA extended over this region. Profiles over Paphos were sampled during periods with and without the AMA being positioned over Cyprus. Here profiles over Paphos, Etna, and Oberpaffenhofen are used to derive a northern hemisphere (NH) background, while profiles over Egypt and Bahrain are used to derive altitude dependent information under monsoon influence (AMA profiles), and profiles over Gan are used to derive a southern hemisphere (SH) background (Figure 2). Average profiles were calculated in 500 m bins, starting above 4 km to avoid boundary layer effects. Inspection of the CH$_4$ AMA profile indicates a significant enhancement in the upper troposphere between 9 and 12.5 km corresponding to pressure levels between 300 and 170 hPa. Park et al. (2007) used CO observations from satellites and wind fields to identify monsoon influenced air masses inside the AMA at a similar pressure range (200-100 hPa). The average CH$_4$ mixing ratio of the AMA profile between 9 km and 12.5 km is 1919.0±17.2 ppbv, while the average CH$_4$ mixing ratio for the NH background is 1863.4±14.0 ppbv, comparable to CH$_4$ mixing ratios below 9 km measured for the AMA profile 1876.5±8.7 ppbv. The average CH$_4$ mixing ratio for the SH background is 1778.3±19.5 ppbv, significantly lower than either the NH background or the AMA profiles. While the NH background shows only a small increase of CH$_4$ above 11 km, the SH background profile steadily increases with height. Gan is located at the equator and thus influenced by the southern hemisphere during boreal summer when the ITCZ is shifted to the north (Waliser and Gautier, 1993). Since most of the methane sources are in the northern hemisphere north of the ITCZ, the profile over Gan thus to some extend represents the SH background. The observed CH$_4$ increase with height can be explained by the global circulation. In the boundary layer CH$_4$ mixing ratios are influenced by turbulent mixing close to emission sources or by horizontal advection in remote places (Saito et al., 2013). At the surface the air at Gan is influenced by wind from southern directions with low CH$_4$ mixing ratios originating from the southern Indian Ocean. High altitude advection leads to interhemispheric transport (Saito et al., 2013) thus to transfer of higher CH$_4$ mixing ratios from the NH into the SH, which have been convectively uplifted from the boundary layer. The observed difference in CH$_4$ background between the NH and the SH is 85.1 ppbv, which agrees with an interhemispheric gradient of 86-90 ppbv for the period 2007 to 2010 given in Bergamaschi et al. (2013).

The measured mean CO profiles for AMA (74.2±10.9 ppbv), NH background (68.8±7.3 ppbv) and SH background (63.2±4.3 ppbv) are rather similar and agree within the standard deviations. Nevertheless, in the upper troposphere the AMA profile indicates a slight increase of CO mixing ratios relative to the background. Enhanced CO and CH$_4$ mixing ratios in the upper troposphere over the eastern Mediterranean in summer 2001, associated with air masses influenced by the monsoon, were also observed during the MINOS aircraft campaign (Lelieveld et al, 2002, Scheeren et al., 2003). This is consistent with

the observed upper tropospheric increase of CO and CH$_4$ in the NH background profiles, which are found during ascends and descends over Paphos but not over Oberpfaffenhofen. In general, our observation of enhanced CO mixing ratios under monsoon influence are consistent with Park et al. (2008), who showed that satellite-based averaged CO profiles exhibit increased CO mixing ratios in the upper troposphere inside the AMA (around 100 ppbv in 10-15 km) in comparison to air outside the AMA (65-90 ppbv in 10-15 km).

To differentiate between air masses in and outside the AMA various approaches have been used in the literature. Often potential vorticity (PV) is used for this purpose (e.g. Randel and Park, 2006, Garny and Randel, 2013 or Ploeger et al., 2015). Ploeger et al. (2015) calculated PV from reanalysis data to determine a transport barrier isolating the AMA. In the restricted area of interest low PV values are found inside the anticyclone while higher PV values represent the background. A more direct approach is the use of a CO threshold (Park et al., 2008). Based on satellite data, Park et al. (2008) found that CO mixing ratios < 60 ppbv represent background air while CO mixing ratios > 60 ppbv represent air inside the AMA at 16.5 km. In our study the monsoon influence in the upper troposphere is most obvious in the CH$_4$ profile, while CO is less suitable due to its larger atmospheric variability associated with its shorter lifetime (Junge, 1974) and the instrumental problems experienced for CO during the second half of the campaign. Therefore, a methane threshold was derived to signify monsoon influenced air masses from the NH background profile. To avoid boundary layer effects and the above mentioned slight increase in the NH background profile above 11 km due to a small contribution of monsoon influenced air above the eastern Mediterranean, only data between 4 km and 10 km were used, yielding an average CH$_4$ mixing ratio for the background of 1859.4±10.2 ppbv, which is slightly lower than the above mentioned mixing ratio covering the whole altitude range. The CH$_4$ threshold is then defined as this average plus twice the standard deviation:

$$CH_{4\,threshold} = CH_{4\,average} + 2\,\sigma = 1859.4 \text{ ppbv} + 2 * 10.2 \text{ ppbv} = 1879.8 \text{ ppbv}, \tag{1}$$

In situ CH$_4$ mixing ratios that exceed this threshold are assumed to be influenced by the South Asian monsoon and are therefore representative of the AMA, in the following denoted as being AMA-influenced. While in the NH background profile, CH$_4$ mixing ratios in the upper troposphere are generally smaller than this threshold CH$_4$ mixing ratios in the AMA profile significantly exceed this threshold above 9 km (Figure 2). Further evaluation depends on the CH$_4$ threshold and thus the results are sensitive to it. Nevertheless, also other compounds measured during OMO showed the isolation of the anticyclone in the upper troposphere (Lelieveld et al., 2018) which confirms the possibility of this threshold to divide the air mass origin between inside and outside the AMA. With a change in the absolute value the region which is supposed to be AMA-influenced will be either larger or smaller, thus the edge of the anticyclone would be differently defined but the whole dynamical process is not significantly changing.

## 3.2 Case study flight 19

To illustrate the connection between enhanced $CH_4$ mixing ratios, monsoon convection, and South Asian pollution sources at the surface we performed a case study on flight 19 data (August 13, 2015). The flight took place over the Arabian Peninsula. After take-off from Paphos HALO headed towards Oman before returning back to Paphos. Enhanced mixing ratios for CO and $CH_4$ were measured between 10-11UTC (Figure 3). Over Oman at a pressure level of 175 hPa mixing ratios for CO and $CH_4$ increased from background levels of $74.3\pm10.6$ ppbv and $1846.7\pm16.1$ ppbv to $99.5\pm14.3$ ppbv and $1905.2\pm13.9$ ppbv, respectively. According to the classification defined in the previous chapter $CH_4$ mixing ratios reached values well above the threshold indicating that air masses influenced by the monsoon were probed. Elevated mixing ratios were still observed after a flight level change (200 hPa). Accordingly, both flight levels were within the altitude range of the AMA confinement of 200-100 hPa reported by Randel and Park (2006). Within the AMA the average increase relative to background for CO is around 25 ppbv and for $CH_4$ around 58 ppbv. The increase in $CH_4$ is rather sharp, indicating a rather well defined edge of the AMA, as has been reported in previous studies (e.g. Park et al., 2008).

For the MINOS campaign over the eastern Mediterranean, Scheeren et al. (2003) distinguished between air masses that originated over South Asia and those over North America/North Atlantic, corresponding to our classification of AMA air masses and NH background, respectively. Note that Scheeren's South Asia air mass would be incorporated in our background, due to its location over the eastern Mediterranean. Scheeren et al. (2003) reported in situ trace gas measurements for the 6-13 km altitude range. Mean CO mixing ratios were $74\pm12$ ppbv for North American/North Atlantic origin and $102\pm4$ ppbv for air masses with a South Asian origin, resulting in a difference of 28 ppbv. The relative difference in $CH_4$ observed by Scheeren is 63 ppbv (North America/North Atlantic: $1819\pm26$ ppbv, South Asia: $1882\pm21$ ppbv). The enhancements observed during MINOS are similar to those observed during OMO flight 19, although absolute mixing ratios in particular for $CH_4$ are higher, since global $CH_4$ concentrations have been increasing since summer 2001 (Zimmermann et al., 2018). Furthermore, similar increases in CO and $CH_4$ mixing ratios, caused by an outflow event of the AMA, were documented over Northern Europe during a TACTS flight by Vogel et al. (2014). They reported enhancements of approximately 25 ppbv for CO (background: 15-25 ppbv, outflow air: 40-50 ppbv) and 65 ppbv for $CH_4$ (background: 1700-1750 ppbv, outflow air: 1770-1810 ppbv). The absolute values for CO and $CH_4$ are lower in comparison to the present study. The transport time was about five weeks so the air mass of the outflow event could be mixed with background air. The background itself has a different characteristic as the flight during TACTS took place in the lower stratosphere and flight 19 took place in the upper troposphere.

Using FLEXPART, 10-day centroid back trajectories along the flight track were calculated. An analysis indicates that in general enhanced $CH_4$ mixing ratios are associated with an air mass origin inside the AMA, while lower $CH_4$ mixing ratios are associated with background air (Figure 4). In particular the back trajectories starting at release points with the highest $CH_4$ mixing ratios measured along the flight track (Figure 7) have been confined in the AMA for several days with their origin over Northern India and Bangladesh. Between five to ten days before observations the back-trajectories are found in the boundary

layer or the lower troposphere, before they are uplifted into the upper troposphere by deep convection (> 200 hPa) (Figure 5). This finding is in good agreement with Bergman et al. (2013), who calculated trajectory transit times of 2-22 days from the surface to the 200 hPa level in the region of the Tibetan Plateau and India/SE Asia. After the convective injection into the upper troposphere the air masses in this case study are advected at the southern edge of the AMA, following the tropical easterly jet towards the measurement region over the Arabian Peninsula. The transport distance of an air parcel in the upper troposphere depends on its origin and takes 1-6 days in the present study depending on the area of convective transport (Northern India, Bangladesh, Bay of Bengal). Scheeren et al. (2003) during MINOS found a longer transport time for polluted air masses ranging from 7 to 10 days from the South Asian source region towards the eastern Mediterranean, which also represents a longer transport pathway. During ESMVal the long range transport in the upper troposphere from the eastern part of the AMA towards the Arabian Peninsula along the southern fringe of the AMA took 2-4 days and the majority of the trajectories were circulating around the AMA within 10 days prior to the observation (Gottschaldt et al., 2017).

By comparing the back trajectories with satellite images of daily mean cloud top pressure it was possible to identify regions with strong convection (here with cloud top pressure below 200 hPa) that were intercepted by the trajectories during their uplifting phase (illustrated in Figure 6). Matches were generally found over the Bay of Bengal, the Indo-Gangetic Plain, Bangladesh, the north eastern region of India, and Myanmar. During the days when the back trajectories passed over central India, convection occurred also in this area, but the cloud top pressure was at a lower altitude than the height of the trajectories. Thus the influence from central India seems to be negligible for this particular flight. Convective cloud top information for estimating the influence of convection on the transport pathways of trajectories were already applied in previous studies, like Scheeren et al. (2003) and Gottschaldt et al. (2017). They found the strongest convection occurring in the same area as mentioned above.

To compare observations with model simulations, Figure 3 shows time series for CO and $CH_4$. For flight 19, EMAC model simulation results for CO and $CH_4$ agree well with the observations (CO: 2.1±8.7 ppbv, $CH_4$: 11.9±21.7 ppbv), reproducing observed trends in mixing ratios for both species, although the model has a rather course resolution of 2.8°x2.8°. In general, the model tends to underestimate the enhanced $CH_4$ mixing ratios in particular for AMA-influenced regions and overestimates CO. A comparison for all flights during the OMO mission yields a model overestimation of 4.6±11.8 ppbv for CO and an underestimation of 7.0±32.8 ppbv for $CH_4$ (see section 3.4.). A comparison between in situ CO and EMAC simulations for the ESMVal flight showed good agreement with a negative bias of the simulated CO, which was in regions of strong CO gradients about 10 ppbv, otherwise smaller (Gottschaldt et al., 2017). Since the trace gases mixing ratios and trends are in general well reproduced by the EMAC model, it will be extensively used for further interpretation of the measurements in the remainder of the manuscript.

The position of the AMA can be determined from horizontal transects of EMAC daily means of trace gas distributions and meteorological data on a pressure level of 204 hPa (Figure 7, 8). Here the anticyclone is identified by the wind field and

corresponding CO and $CH_4$ fields. Enhanced trace gas mixing ratios are found to be confined within the anticyclone due to the strong isolation caused by the anticyclonic circulation (Park et al., 2008). On August 13, 2015, the AMA extended from the eastern part of the Arabian Peninsula to the eastern part of China and from the northern part of the Bay of Bengal to the Gobi Desert. Its centre was located over the Tibetan Plateau, which is consistent with the climatological mean position of the AMA

as documented by Nützel et al. (2016). The simulated mixing ratios inside the anticyclone increase from around 90 ppbv (CO) and 1860 ppbv ($CH_4$) at the fringes of the anticyclone to values inside the AMA above 150 ppbv and 1920 ppbv for CO and $CH_4$, respectively. Mixing ratios in background air over the Mediterranean are around 65 ppbv for CO and 1840 ppbv for $CH_4$ and thus below those simulated inside the AMA. Enhanced mixing ratios in the AMA were reported also in previous studies, e.g. increased $CH_4$ mixing ratios in satellite data over the summer monsoon region by Park et al. (2004) and Xiong et al. (2009),

or enhanced CO values from satellite measurements in the monsoon region by Li et al. (2005). The latter study reported CO mixing ratios of up to 133 ppbv at a pressure level of 147 hPa over the monsoon region, which fits with our slightly higher simulated CO values at a lower pressure level, if we assume that CO mixing ratios decrease with height (Park et al., 2008). Since the 200 hPa level is representative for the dominant flight altitude, the in situ observations along the flight track can also be compared to the simulated 2D fields. Further, we assume that the 200 hPa level is where most of the convective outflow

takes place, and therefore pollution levels are expected to be highest (Park et al., 2009). Figure 7 and 8 show that OMO flight 19 only scratches the western edge of the AMA. Measured $CH_4$ mixing ratios are higher inside the AMA than the simulated ones. This is in line with the above mentioned $CH_4$ underestimation by the model. In the model simulations the anticyclone shows a more distinct signal in $CH_4$ compared to CO, since the edge in $CH_4$ is well-defined, with mixing ratios dropping off significantly outside the anticyclone. In contrast, the CO pattern is more diffuse. The simulated CO pattern, especially the

enhanced values over Oman, fits well to the observed CO mixing ratios along the flight track. The EMAC model underestimates $CH_4$ and CO in the upper troposphere. As shown by Krol et al. (2018), EMAC seems to have a weaker transport of surface tracers than other models. There are two potential reasons for that, but it is difficult to distinguish them. First, a too slow vertical velocity, thus the convective updraft is too ineffective, or second, the numerical diffusion implied by the coarse resolution restricts the updraft too strong. Nevertheless we would like to notice that for the comparison of CO with the model,

the results are in line with other literature studies at such resolution (e.g. Baret et al., 2016). Additional vertical transects along 23.7°N latitude and 56.2°E longitude on August 13, 2015 complete the picture of the AMA with respect to its extension. In a vertical $CH_4$ transect along 23.7°N (Figure 9) it is obvious that the flight touches only the western edge of the anticyclone in the upper troposphere and that the majority of the flight took place outside the anticyclone. According to the model simulation, convective uplift of $CH_4$ takes place between 75°E and 95°E, which corresponds to India and the Bay of Bengal. Moreover,

the upward transport of polluted air masses is only simulated in a rather restricted area, analogous to a chimney, as reported by Bergman et al. (2013). This area was also the preferred location for convection 10 days prior to the flight as reported above. Rauthe-Schöch et al. (2016) reported a similar longitudinal position for convection between 80°E and 100°E in summer 2008

for CARIBIC flights over India. In the vertical transect along 56.2°E (Figure 10) $CH_4$ mixing ratios show an increase at the surface from the equator towards higher northern latitudes. In the upper troposphere the AMA is located approximately between 15°N and 30°N, which fits well with the location of the AMA in summer 2008 identified by enhanced $CH_4$ mixing ratios observed on a CARIBIC flight between 10-40°N (Baker et al., 2012). In vertical transects at longitudes between 75-95°E (not shown) the convection can be determined to occur between 20°N to 35°N, which reflects the area Indo-Gangetic Plain, Tibetan Plateau, Bangladesh, and the north eastern part of India. In the vertical transects for CO (not shown) the same patterns are found, although less pronounced compared to $CH_4$. In a CO latitudinal transect along 23.7°N  the enhanced mixing ratios range from around 90 ppbv inside the anticyclone to over 400 ppbv at the surface, while $CH_4$ mixing ratios scale from around 1850 ppbv inside the anticyclone to surface values above 2250 ppbv (Figure 9).

## 3.3 Emission source region

Emission source regions are identified by combining the FLEXPART single particle information with EMAC daily means for CO and $CH_4$ mixing ratios in the planetary boundary layer. With FLEXPART the last boundary contact can be determined as a footprint of parcels, which started from release points with enhanced methane, i.e. $CH_4$ mixing ratios > threshold. The last boundary layer contact of a parcel is determined if its height position is within ± 5% of the PBL height (in m above ground level). These parcels are sorted depending on their geographical position and then summed-up in each grid cell (1°x1°), yielding a number of parcels per grid cell in the PBL. For flight 19 (August 13, 2015) the footprint of the last PBL contact approximately 10 days prior to the parcel release shows highest values over Bangladesh and the north eastern part of India (Figure 11). The last boundary layer contact is assumed to be a useful indicator for the area where parcels are uplifted by convection and subsequently injected into the AMA.

To identify emission sources, EMAC daily mean trace gas mixing ratios at a pressure level of 1008 hPa (assumed to be the surface) are presented. The $CH_4$ mixing ratios in Figure 12 exhibit the highest values in the Indo-Gangetic Plain and Bangladesh, spreading southward along the east coast of India and towards Myanmar. Peak values reach up to 2600 ppbv. Figure 13 shows that CO has the highest mixing ratios in the same region as $CH_4$, with peak values up to 400 ppbv. A comparison to Figure 11 indicates that high CO and $CH_4$ mixing ratios are co-located with the areas of convection, leading to an updraft of polluted air masses to the upper troposphere. In the footprint (Figure 11) the east coast of Africa, the Indian Ocean, and the Pacific indicate a minor influence, but they do not correspond to enhanced mixing ratios in EMAC daily mean $CH_4$ or CO. Thus, they are assumed not to contribute to the pollution of the air masses in the AMA. The EMAC daily mean data also indicate polluted areas along the east coast of China. This area is not reflected in the parcel footprint, which indicates that the south eastern part of China is only a minor contributor to the pollution of the AMA on this particular flight. Note that

some convection in the daily mean cloud top pressure maps occur approximately 10 days prior to the observation over eastern China, confirming that Chinese emissions could have a minor contribution on the composition of the AMA.

The above mentioned peak values at the surface for CO and $CH_4$ over Southern Asia can be related to different emission sources. The regions including the Indo-Gangetic Plain and Bangladesh are densely populated areas with an increasing population trend in combination with strong economic development (Rauthe-Schöch et al., 2016, Ohara et al., 2007). Potential CO sources are biomass, fossil, and domestic fuel combustion as well as oxidation of $CH_4$ and volatile organic compounds (Pandis and Seinfeld, 2006). $CH_4$ is emitted by rice cultivation, wetlands, domestic ruminants, biomass burning, fossil fuels, waste decomposition (Khalil, 2000), and flood plains, especially if they are polluted by urban waste and sewage (Baker et al., 2012). The latter has its maximum emission during the monsoon due to the influence of rain. At the same time rice has its primary growing period in the wet season and hence $CH_4$ emissions have a seasonal cycle and are strongest during the monsoon, contributing significantly to the $CH_4$ emissions (Baker et al., 2012). The simultaneous appearance of a $CH_4$ emission maximum and the strong convection leads to a more pronounced chemical signature in the AMA for $CH_4$ than for CO. The CO emission sources do not experience such a strong seasonal dependency. This emphasizes again the use of the $CH_4$ threshold to differentiate between AMA-influenced and background air masses.

Bangladesh and the north eastern part of India are also mentioned in the study of Pan et al. (2016) as the preferred uplifting regions, which agrees with the main areas of the footprint determined in this study. The marked source regions were also identified by Park et al. (2009), as they reported the origin of upper tropospheric CO to be mainly from India and Southeast Asia, and by Pan et al. (2016) who identified Northeast India, the southern flank of the Tibetan Plateau, Nepal, and north of the Bay of Bengal to be the most preferred spots for CO uplifting. Rauthe-Schöch et al. (2016) identified similar source regions using FLEXPART 10 day back trajectories for CARIBIC flight tracks over the South Asian monsoon region. They used geographical positions of back trajectory points below 5 km, thus not only the boundary layer as used here, and documented air originating from India, Indo-Gangetic Plain, Bay of Bengal, mainland of Southeast Asia, and the western part of the Arabian Sea. Consequently, their area of source regions covers a larger part in comparison to the source region identified for flight 19. Bergman et al. (2013) demonstrated that trajectories from eastern China have only a minor influence to the AMA at the 200 hPa level, which supports our footprint analysis as it does not show any boundary layer influence from the eastern coast of China.

### 3.4 The AMA during OMO

Similar analysis as for flight 19 is discussed in detail for the other OMO flights including time series, back trajectories, and model results in the supplement. Here we extend the analysis of OMO results by analysing CO and $CH_4$ mixing ratios between 300 and 140 hPa from all research flights. In Table 1 observations and model data are separated by the $CH_4$ threshold into monsoon-affected and background periods. Observed CO mixing ratios increased under monsoon influence by about 20.1 ppbv, while $CH_4$ mixing ratios were enhanced by about 72.1 ppbv. EMAC SD4 CO and $CH_4$ also indicate increased

mixing ratios along the flight tracks within the AMA, but not as strong as for the observations (14.7 ppbv for CO and 24.0 ppbv for $CH_4$).

The observed increase in $CH_4$ inside the AMA by ~ 70 ppbv is in good agreement with the reported increase of 30-80 ppbv (at 8-12.5 km) between pre-monsoon and monsoon season reported by Schuck et al. (2012). Xiong et al. (2009) reported a $CH_4$ increase of up to 100 ppbv from June to September (for 2003-2007) at 300 hPa. Both studies are based on data observed over India, thus much closer to the AMA centre, which might explain the difference in absolute values. Moreover, they mirror seasonal variations by comparing pre-monsoon with monsoon conditions and thus are not necessarily representative for background conditions as determined during OMO.

In Figure 14 and 15 histograms for $CH_4$ and CO mixing ratios, respectively, are shown for observations and model data at altitudes between 300 and 140 hPa separated into values above (AMA) and below (background) the $CH_4$ threshold. The CO observations and model data have similar distributions for background as well as AMA conditions. Average CO mixing ratios for observations and model data for AMA and background conditions agree within their $1\sigma$-standard deviations (Table 1). The $CH_4$ mixing ratio average from the model for background conditions also agrees with the observation within their combined standard deviations, while the difference of the AMA $CH_4$ averages between the model and the observations is more pronounced. Observed $CH_4$ averages for background and AMA can be clearly distinguished, which is mostly due to the $CH_4$ threshold itself, which subdivides the data into two regimes. The distribution of the in situ $CH_4$ background data consists of two modes. The one with the low mixing ratios is associated to the SH background, and consists of observations over the Indian Ocean from or towards Gan. In the southern hemisphere the $CH_4$ mixing ratios range from 1760 ppbv to 1820 ppbv. The second mode represents the NH background with mixing ratios in the range of 1820 to 1880 ppbv, comprised of observations over the Mediterranean and partly the Arabian Peninsula. In comparison, the simulated $CH_4$ background distribution lacks the SH mode. EMAC also underestimates the AMA $CH_4$ mixing ratios. Accordingly, the $CH_4$ enhancement of the model is smaller than in the observations. AMA-influenced air masses show a broad distribution of $CH_4$ mixing ratios with values ranging from 1880 ppbv to 1980 ppbv. During the OMO campaign the position of the anticyclone changed repeatedly and thus observations were made at varying distance between the aircraft and the centre of the anticyclone (see next chapter). As expected, this leads to variations in the observed $CH_4$ mixing ratios for air masses influenced by the AMA. Furthermore, changes in the location of deep convection, the strength of the updraft or differences in emission sources lead also to variability in the observed $CH_4$ mixing ratios (see section 3.5). While the observed background conditions for both NH and SH each cover a range of approximately 60 ppbv, the AMA mixing ratios vary by about 100 ppbv.

**3.5 AMA modes**

Over the course of the OMO campaign the occurrence, position and extent of the AMA varied. The AMA pattern can be subdivided into four meteorological situations, which can be identified in the EMAC daily means for $CH_4$ and the wind pattern

on a pressure level of 204 hPa. The first mode (Figure 16) is composed of 2 distinct anticyclones, which slowly move eastward between July 21 and August 1, 2015. On July 21 a western anticyclone is positioned over the Eastern Mediterranean. Its centre subsequently shifts towards the east. The eastern anticyclone with its centre around 70°E shifts eastward towards the Tibetan Plateau. The second mode (Figure 17) which consists of a single anticyclone is found during the period from August 6, 2015 to August 10, 2015. This mode has its centre over the Kashmir region (~70-80°E), which corresponds to the climatic mean location of the AMA centre (Zhang et al., 2002) and is called the central mode. The third single anticyclone mode (Figure 18) has its centre over the Tibetan Plateau (~82.5°-92.5°E) and was observed between August 11, 2015 and August 18, 2015 (Tibetan mode; Zhang et al., 2002). In the following (August 20, 2015 to August 27, 2015) the anticyclone of the Tibetan mode moves westward and splits-up into 2 anticyclones (Figure 19). Here the westward movement of the AMA leads to instabilities in the circulation (Popovic and Plumb, 2001) so that a second anticyclone can break-off the main feature (Hsu and Plumb, 2000). The western anticyclone now has its centre above the Middle East and the eastern part over the Tibetan Plateau.

During OMO the time period between the reoccurrence of the 2 distinct anticyclones is around 20 days. A 10-20 day cycle of westward propagation of the anticyclone including splitting into two anticyclones has been reported by Krishnamurti and Ardanuy (1980) and shown to lead to a succession of rainy and dry periods in India during the monsoon season. Zhang et al. (2002) presented a bimodality of the AMA with a centre position of the anticyclone over the Iranian or the Tibetan Plateau. During OMO we found both positions, which is in line with the bimodality assumption. In contrast, Nützel et al. (2016) reported different centre positions of the AMA in several models, but most of them did not simulate a preferred bimodality. Regarding the eastern anticyclones during the double anticyclones modes, the positions were in between the Iranian and Tibetan Plateau (first mode) and in the fourth mode over the Tibetan Plateau. Consequently, they do not support a preferred bimodality. In Zhang et al. (2002) and Nützel et al. (2016) the Iranian and the Tibetan mode are further distinguished by parameters, like diabatic heating, rain patterns or areas of convection, which are out of scope in the present study. Here the focus is on the dynamics with respect to the trace gas distributions. The subdivision into four modes represents the dynamics of the AMA over the course of the campaign.

Different trace gas levels between these four modes can also be identified in the $CH_4$ and CO observations by subdividing the observations into AMA-influenced and background values at altitudes between 300 hPa and 140 hPa (Table 2). The lowest $CH_4$ mixing ratios in background air are observed during the central mode, since during this time HALO was flying mostly over the Indian Ocean at low latitudes. These flights were influenced by southern hemisphere air masses due to the ITCZ shift to the north during boreal summer. Especially over India, the position of the ITCZ is in general between 5°-30°N in summer (Lawrence and Lelieveld, 2010). For the three other AMA modes the $CH_4$ background values are comparable, representing

northern hemispheric background conditions. Their mixing ratios reflect also the $CH_4$ mixing ratio derived for the NH background profile (1859.4 ppbv) in section 3.1.

CO mixing ratios in background air masses show hardly any difference between the northern and the southern hemisphere in the upper troposphere. This is consistent with airborne observations during the Indian winter monsoon (January 1999) over the northern part of the Indian Ocean during the INDOEX campaign (Gouw et al., 2001). While no significant CO gradient was observed in the upper troposphere, a pronounced north-south gradient of $3.9 \pm 1.9$ ppbv deg$^{-1}$ latitude was observed below 3 km (Gouw et al., 2001)

Differences in $CH_4$ and CO mixing ratios for AMA-influenced air masses during OMO can be explained by the relative distance between the position of HALO and the position of the centre of the AMA. With increasing distance from the centre of the anticyclone, mixing ratios tend to decrease. The distribution of the trace gases in the upper troposphere depends on the dynamics of the monsoon, since position and strength of the convection change during the wet season (Randel and Park, 2006). For example, if convection takes place over the Bay of Bengal, less polluted air is uplifted. In contrast, if the convection is directly over densely populated regions, e.g. Bangladesh, which are more polluted, convection transports the pollutants to the upper troposphere. At the same time, the strength of the convection changes within the monsoon due to differences in thermal heating between the relative cold ocean compared to the hot land (Dethof et al., 1999).

During OMO, the $CH_4$ mixing ratios for AMA influenced air masses varied temporally from mode to mode. The different mixing ratios do not follow a simple systematic (Table 2), and cannot be fully explained by the relative distance towards the AMA centre. Although the distances to the respective anticyclone centres were shortest during the two anticyclones modes, the observed $CH_4$ mixing ratios were both lowest and highest. In these cases, only the western AMA was probed above the eastern Mediterranean or the Middle East, because the flight tracks were in the same region. In contrast, the $CH_4$ mixing ratios for the two single AMA modes represented levels in between the minimum and the maximum $CH_4$ mixing ratios. In fact, $CH_4$ mixing ratios influenced by the monsoon increased from the first mode towards the fourth. The increase in average $CH_4$ is even more pronounced if we only include flights 22 and 23 in the fourth mode, as the flight track of flight 24 over the Mediterranean was not at all impacted by the AMA. Thus, the average $CH_4$ mixing ratio for the last mode becomes 1927.1 ppbv, which indicates an even stronger $CH_4$ increase over the course of the OMO campaign. The $CH_4$ enhancement can be most likely explained with a combination of the position of the convection, the temporal development of the AMA, and the accumulation of emission in time. A strengthening of the convection and/or a shift of the convection towards areas with larger emission sources would lead to higher $CH_4$ mixing ratios in the upper troposphere. A temporal increase in emissions, as described for $CH_4$ in section 3.3, induces also an increase of $CH_4$ in the AMA as reported by Xiong et al. (2009). They observed an enhancement of $CH_4$ in the upper troposphere during the monsoon season, starting with pre-monsoon conditions around

June, increasing toward the end of the monsoon season (September). Accordingly, the strong AMA circulation traps the polluted air masses and the degree of pollution increases during the monsoon season.

The $CH_4$ mixing ratios in the EMAC data at 204 hPa show similar distributions to the observations along the flight tracks, indicating that the simulations of the AMA position agrees with the observations. Note that the model may not resolve small scale features due to its coarse resolution, which will lead to deviations between simulated and observed trace gas distribution, in particular close to the fringe of the AMA. In Section 3.6 we will describe a case study were the model successfully captures the weakening of the transport barrier and subsequent outflow of air masses from the AMA.

### 3.6 Outflow event from the AMA

Besides the different modes mentioned above, an outflow event was detected during OMO. In the EMAC data on 204 hPa an air mass with enhanced $CH_4$ mixing ratios started to move from Bangladesh westwards on August 2, 2015. In the previous days, convection took place over Bangladesh and injected polluted air masses into the upper troposphere. The convection can also be identified in the cloud top pressure data showing high clouds over Bangladesh. The air mass was further transported at the southern edge of the AMA following the tropical jet towards the west.

On August 3/4, a disturbance in the subtropical jet caused an instability in the AMA circulation which led to a weakening of the wind field especially in the south western part of the AMA. The slow AMA circulation was associated with a weakened transport barrier, thus offering the possibility that air masses inside the AMA were split off. As the observed air mass was already at the fringe of the AMA circulation and thus transported into the region of the weak wind field, it left the AMA on August 5. Afterwards the circulation of the AMA strengthened again. The air mass that left the AMA was probed by HALO over Oman on August 6 (Figure 20). In the following days the westward transport of the air mass continued over the Arabian Peninsula following a south westerly flow on August 8. During its advection the originally compact air mass was stretched and transported north eastwards. On August 10 (Figure 21) it was located above the Red Sea and next moved further northeastward, to be reintegrated into the AMA circulation at the north western edge of the AMA in close proximity to the subtropical jet.

We measured the expelled air mass outside the AMA on two consecutive flights in a quasi-Lagrangian experiment. On August 6 (flight 12/13, Figure 20) HALO probed the expulsion with enhanced CO and $CH_4$ over Oman. In the air mass CO and $CH_4$ mixing ratios increased to $117.3\pm22.2$ ppbv and $1893.5\pm9.8$ ppbv, respectively (background: CO=$78.6\pm33.3$ ppbv and $CH_4$=$1827.4\pm26.8$ ppbv), which can be seen in Figure 22. The second probing of this air mass took place at August 10 (flight 17/18, Figure 21) over the Red Sea yielding mixing ratios of $94.2\pm6.8$ ppbv and $1903.7\pm19.2$ ppbv. This corresponds to the increase at around 12-13 UTC in Figure 23. The CO and $CH_4$ mixing ratios observed on both flights agree well within their standard deviations, with the small (but insignificant) differences probably caused by different flight levels (flight 12/13:

11.9 km and flight 17/18: 12.4 km). Note that the standard deviation for CO during flight 12/13 is larger than for flight 17/18 due to technical problems with the CO. Comparing the EMAC simulations with the in situ data along the flight tracks (Figure 22 and 23), the trends for the outflow agree. The EMAC average mixing ratios for CO and $CH_4$ are 112.2±1.2 ppbv and 1891.7±1.2 ppbv, respectively, for flight 12/13 and 90.8±3.1 ppbv and 1864.6±5.9 ppbv for flight 17/18. Thus also the values

agree within their standard deviation beside $CH_4$ in flight 17/18, where the outflow is underestimated by the model.

To check if the expelled air masses probed in the two flights were connected in a Lagrangian sense, we use centroid back trajectories. In Figure 24 the 10-day back trajectories for the enhanced $CH_4$ values, associated with the locations of the outflow event mentioned before (Figure 20 and 21), are shown for the flights 12/13 and 17/18. The trajectories have their origin in the lower troposphere (below ~550 hPa) over the Arabian Sea and the Indian subcontinent. In the area of Bangladesh, the

trajectories are convectively uplifted to the upper troposphere. From there they follow the tropical jet towards the west, towards Oman and the Red Sea. The trajectories fit well to the simulated movement of the air mass with the enhanced $CH_4$ values. Besides the similar geographical and altitude position of the trajectories of both flights, the positions also agree in time, which means that the back trajectories of flight 17/18, released on August 10, needed four days between their release points and the crossing with the release points of flight 12/13. This travel duration is exactly the time between the two flights. Therefore, in

Figure 25 the trajectories are colour-coded with time, starting from August 10 (day zero) counting backward in time. The trajectories are uplifted in the same period and exceed the 300 hPa level around -9 days. Subsequently, the trajectories of both flights travel together westwards in the same latitudinal band. On August 6 (-4 days), when the trajectories of flight 12/13 are released, they reach Oman and accordingly the release points of flight 12/13. Thus, the trajectories for both flights coincide in time and space. Therefore, they complete the picture of the outflow observed in the simulations and confirm the in situ

measurement analysis.

Beside outflow events at the western edge of the AMA, as documented here, they are also possible at the eastern edge of the AMA as described in Vogel et al. (2014). During TACTS they probed an air mass with enhanced CO and $CH_4$ mixing ratios over Northern Europe in the lower stratosphere. They used back trajectories to analyse the transport pathways of the expelled air mass. The air mass was injected at the south eastern edge of the AMA, streamed clockwise around the AMA, and flowed

out of the AMA at the north eastern part by eastward eddy shedding. Afterwards the air was transported eastward with the subtropical jet. After ca. five weeks the expelled air mass reached the flight track over Northern Europe.

## 4 Summary and Conclusion

The AMA is a circulation system in the upper troposphere and lower stratosphere, which appears over Asia during boreal summer. The anticyclone is coupled to deep convection (Hoskins and Rodwell, 1995), which pumps up polluted air masses.

The relatively strong anticyclonic circulation traps the pollutants inside the AMA and constitutes a clear chemical signature. The emissions have predominant sources in South Asia, and are growing owing to population increase and economic

development. It is not fully understood how the pollutants influence the chemical composition in the AMA, nor in the upper troposphere on a global scale. We especially lack information about the transport pathways of the boundary layer air masses into the AMA and how they escape the anticyclone. In the present work, we address the transport pathways, including the convective transport from the boundary layer into the upper troposphere, the circulation in the AMA, the transport at and

across the edges of the AMA, associated with outflow events and further transport in the upper troposphere partly in connection with the jet streams. The aircraft campaign OMO took place in July/August 2015 in the upper troposphere over the Mediterranean, the Arabian Peninsula, and the Arabian Sea to investigate the AMA and regions west and south of the AMA. On board HALO the trace gases CO and $CH_4$ were measured with the IR-absorption spectrometer TRISTAR. Both trace gases exhibit enhancements in the mixing ratios when influenced by the monsoon. To support our analysis, FLEXPART

back trajectories and EMAC model simulations were used. In this study we focused on the dynamics with respect to trace constituents and their transport pathways into and in the AMA and their origin. In view of the flight tracks in the west of the AMA, the focus is on westward transport pathways. To investigate the long range transport of relatively long-lived species, CO and $CH_4$ are suitable.

The AMA extends vertically from the upper troposphere into the stratosphere. It could be clearly distinguished in the observed

$CH_4$ AMA profile at altitudes of 9-12.5 km, while the NH and SH background profiles show no or minor influence. In the observed CO profiles and the simulated $CH_4$ and CO profiles the signature is not as clear as in the observed $CH_4$ profiles. With the help of the observed $CH_4$ background profile, we calculated a $CH_4$ threshold of 1879.8 ppbv to distinguish between background and AMA-influenced air masses. Over the course of OMO the mixing ratios of CO grew by about 20.1 ppbv due to the influence of the monsoon. The increase in the $CH_4$ mixing ratios was about 72.1 ppbv between background and AMA-

influenced air. Furthermore, $CH_4$ had less background variability than CO, and the $CH_4$ emissions exhibit a seasonality with a maximum during the monsoon season. Consequently, $CH_4$ is an ideal monsoon tracer. In a case study of flight 19 the increase in the trace gas mixing ratios of CO and $CH_4$ can be unambiguously associated with the AMA. Back trajectories indicate transport pathways from the source regions in South Asia towards the measurement region for release points with enhanced $CH_4$ values. Source regions include the Indo-Gangetic Plain, Northeast India, Bangladesh, and the Bay of Bengal. These

regions are densely populated with agricultural, urban, and industrial emissions. Due to convection mainly over the same region the polluted air was uplifted and then transported with the easterly jet towards the measurement region. The transport time was approximately 10 days. The mode of the anticyclone changed during OMO according to Zhang et al. (2002): starting with a double anticyclone mode, transforming to a central mode, shifting to a Tibetan mode and then splitting again into two anticyclones. The position of the anticyclone is visible in the EMAC simulations, but can also be identified in the observations.

Moreover, the observed $CH_4$ mixing ratios influenced by the AMA indicate strengthening in the convection or in the emissions

or a combination of both as well as an accumulation in time over the course of OMO by a continuous increase. Additionally, the transport of an air mass with enhanced trace gas mixing ratios out of the AMA towards the west was observed.

In conclusion, the AMA has a distinct fingerprint in the upper troposphere, which was most prominent in observed $CH_4$, with enhanced mixing ratios inside the AMA circulation owing to the strong $CH_4$ emissions during the monsoon season. The AMA influences the region between the Eastern Mediterranean and East Asia through its extent, position, and by outflow events. We demonstrated the pathways of trace gas from the source regions into and within the AMA. The outflow of polluted air masses from the AMA, by overcoming the transport barrier during weakening circulation, represents how emissions can be further distributed in the upper troposphere and therefore may influence the upper troposphere at a global scale. Consequently, surface emissions alter the chemical composition of the upper troposphere, leading to changes in the atmospheric chemistry, which can influence radiative heating and cooling rates of different trace gases in the upper troposphere. Further investigations will be needed concerning the composition and trace gas chemistry and aerosols in the AMA. In the present study the focus was on long-range transport, while with shorter lived chemical constituents further understanding of the chemical composition can be gained, as in Bourtsoukidis et al. (2017). It will be helpful to extend the measurements further to the east, e.g. over India and the Bay of Bengal region.

**Data availability**

The data are uploaded to the HALO database (https://halo-db.pa.op.dlr.de/) and are available on request.

**Author contribution**

LT has written the original draft of the manuscript and evaluated the data in close cooperation with HF. AP and NO were responsible for the EMAC model simulations. JL was one of the principal investigators of the OMO mission. UP and LT performed the experimental work presented. All authors were involved in the review and editing during the writing.

**Acknowledgements**

We would like to thank all the participants of the OMO mission, the German Aerospace Center (DLR), and EDT Offshore Ltd in Cyprus for the great cooperation during the mission. We also thank Angelika Heil for help with the ECMWF data for the FLEXPART calculations.

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

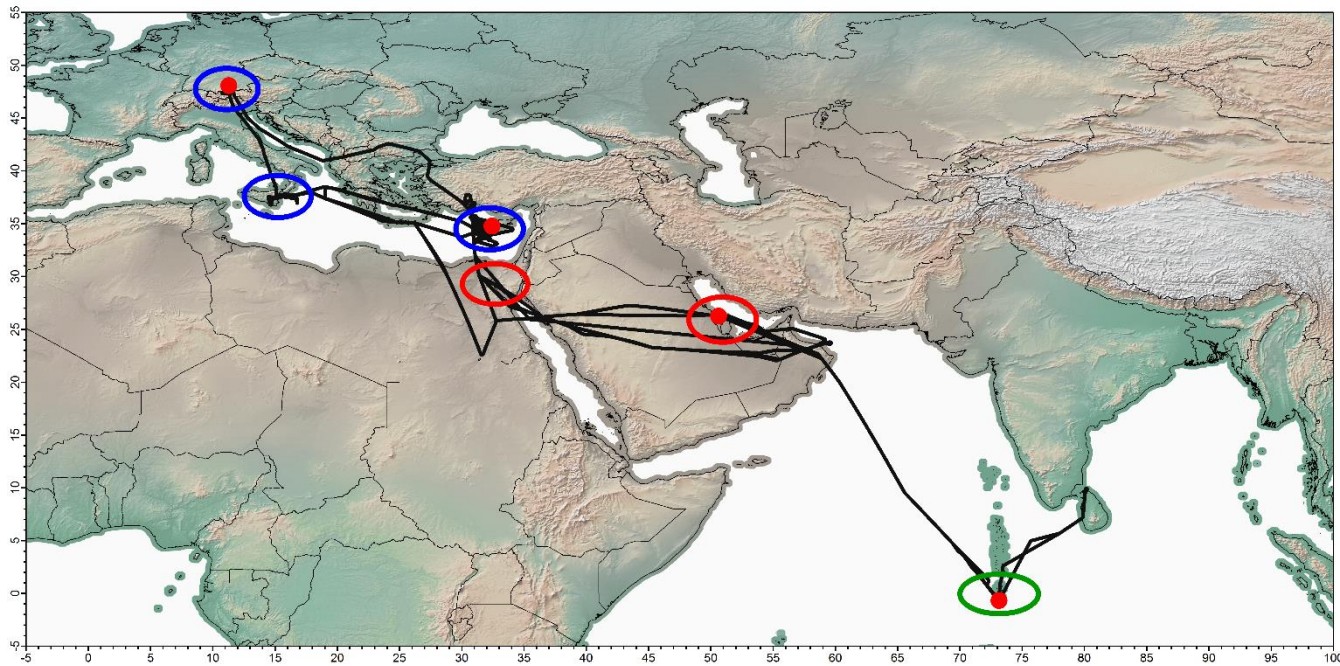

10    **Figure 1: Overview of the flight tracks during OMO with all four airports (red dots): Oberpfaffenhofen (Germany), Paphos (Cyprus), Bahrain and Gan (Maldives). Additionally, the regions of the profiles for northern hemispheric background (blue), AMA (red), and southern hemispheric background (green) are marked.**

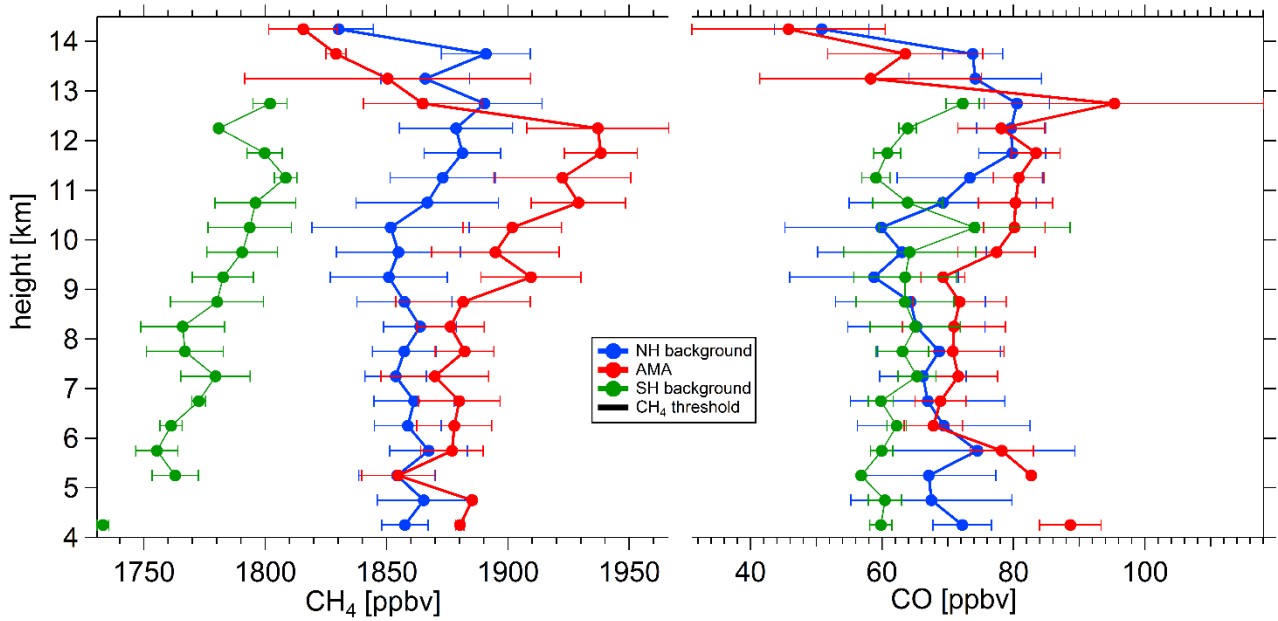

**Figure 2: Average profiles for northern hemispheric (NH) background, AMA and southern hemisphere (SH) background for CH₄ (left) and CO (right); profile locations are presented in Figure 1. The CH₄ threshold (1879.8 ppbv) is indicated by the black line.**

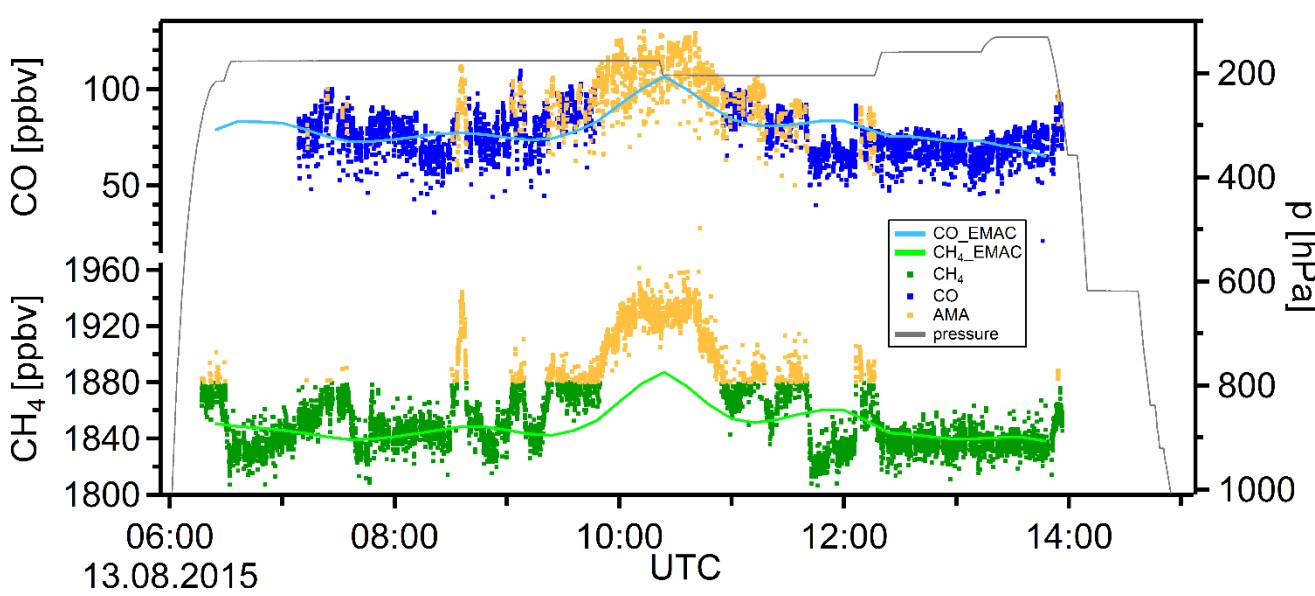

**Figure 3: Flight 19 (August 13, 2015) in situ CH₄ and CO data and EMAC results along the flight track, as well as the flight altitude. The AMA is colour coded by CH₄>1879.8 ppbv.**

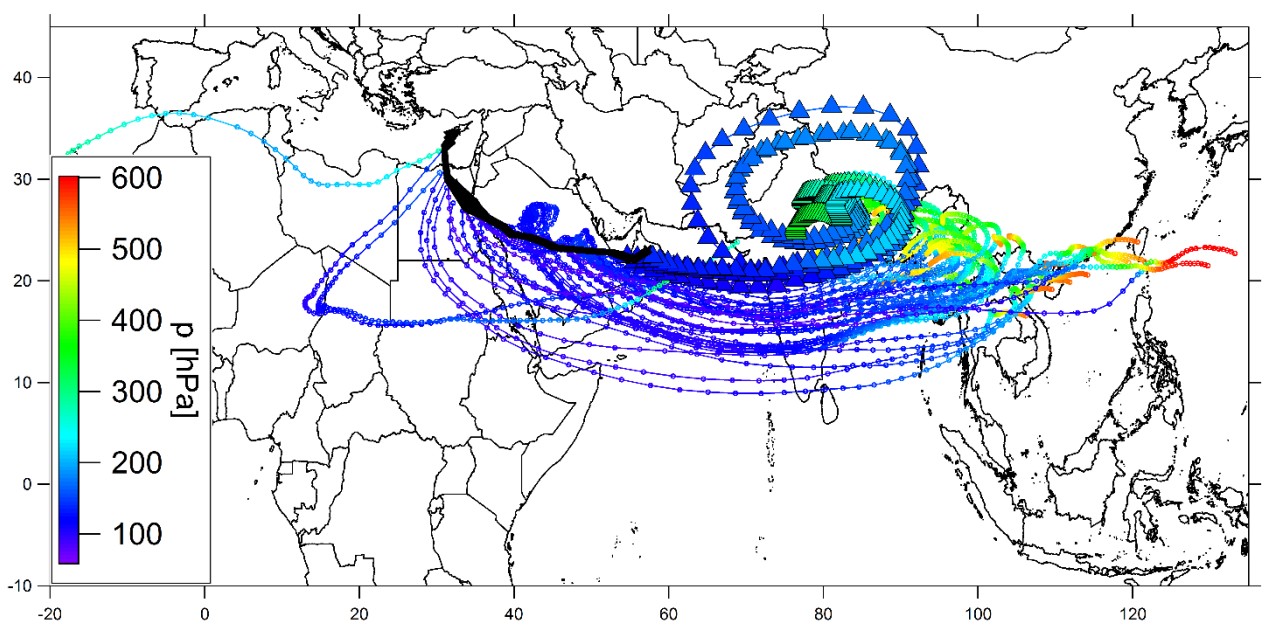

**Figure 4: Centroid trajectories for flight 19 (August 13, 2015) with colour coded altitude. Triangles are back trajectories for CH$_4$ mixing ratios above the CH$_4$ threshold and circles for below the CH$_4$ threshold.**

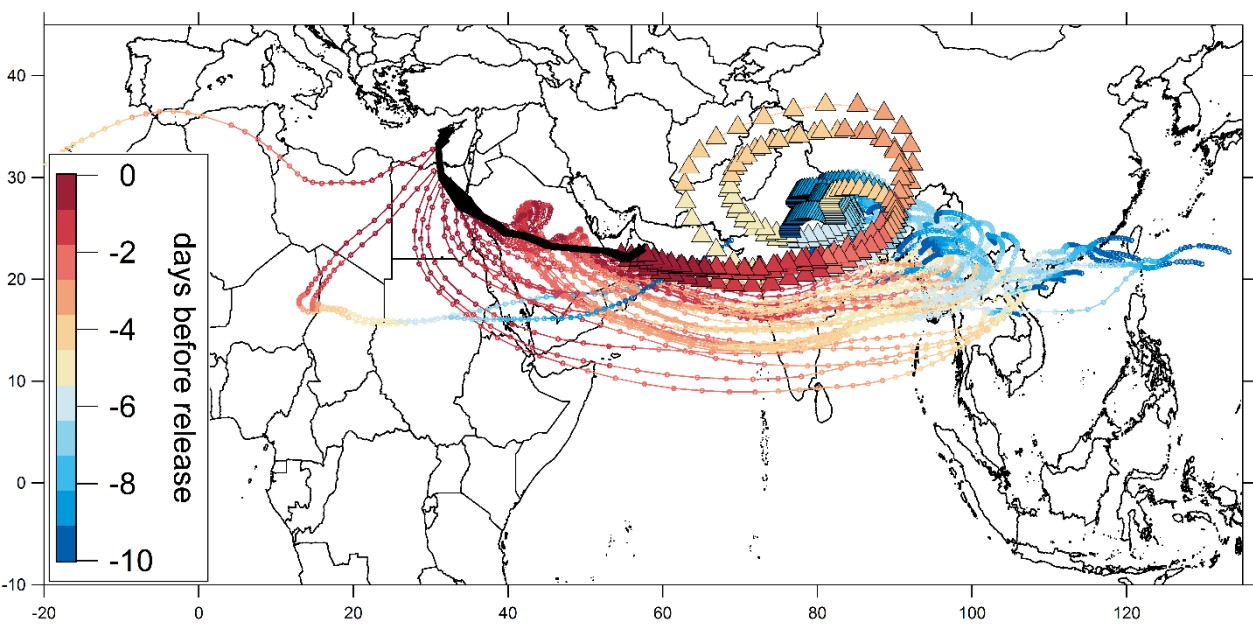

5    **Figure 5: Centroid trajectories for flight 19 (August 13, 2015) with colour coded transport time. Triangles are back trajectories for CH$_4$ mixing ratios above the CH$_4$ threshold and circles for below the CH$_4$ threshold.**

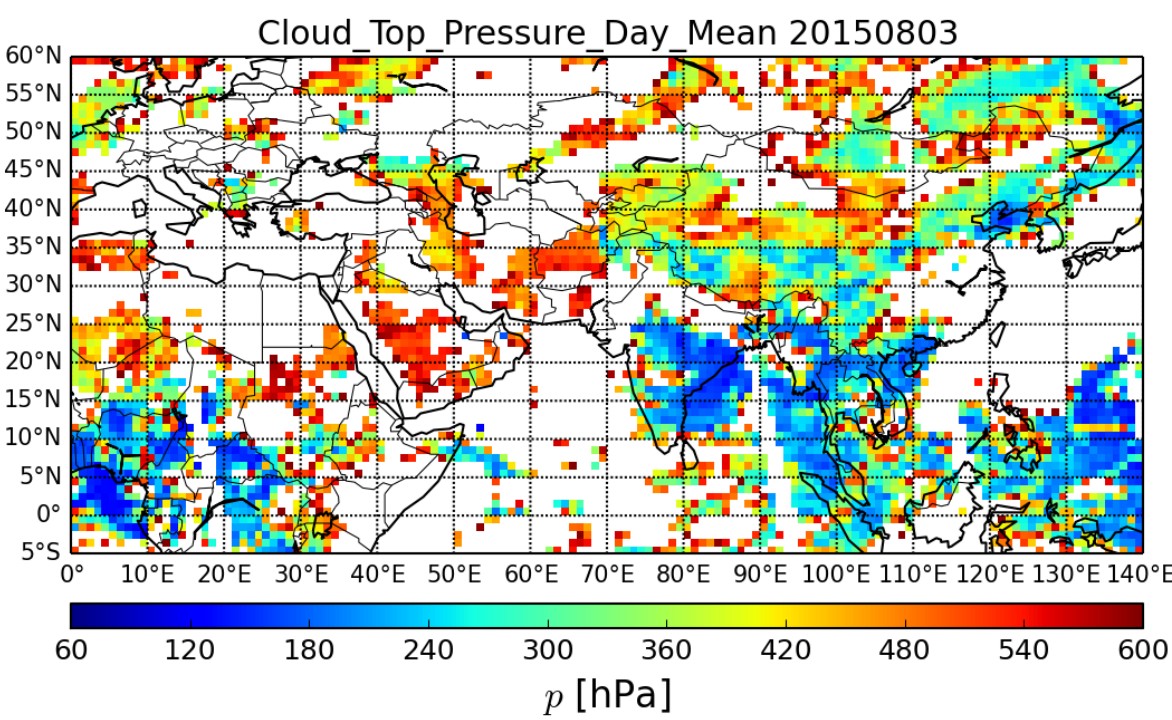

**Figure 6: Satellite-derived cloud top pressure 10 days prior (August 03, 2015) to flight 19. Pressure below 250 hPa represents strong convection.**

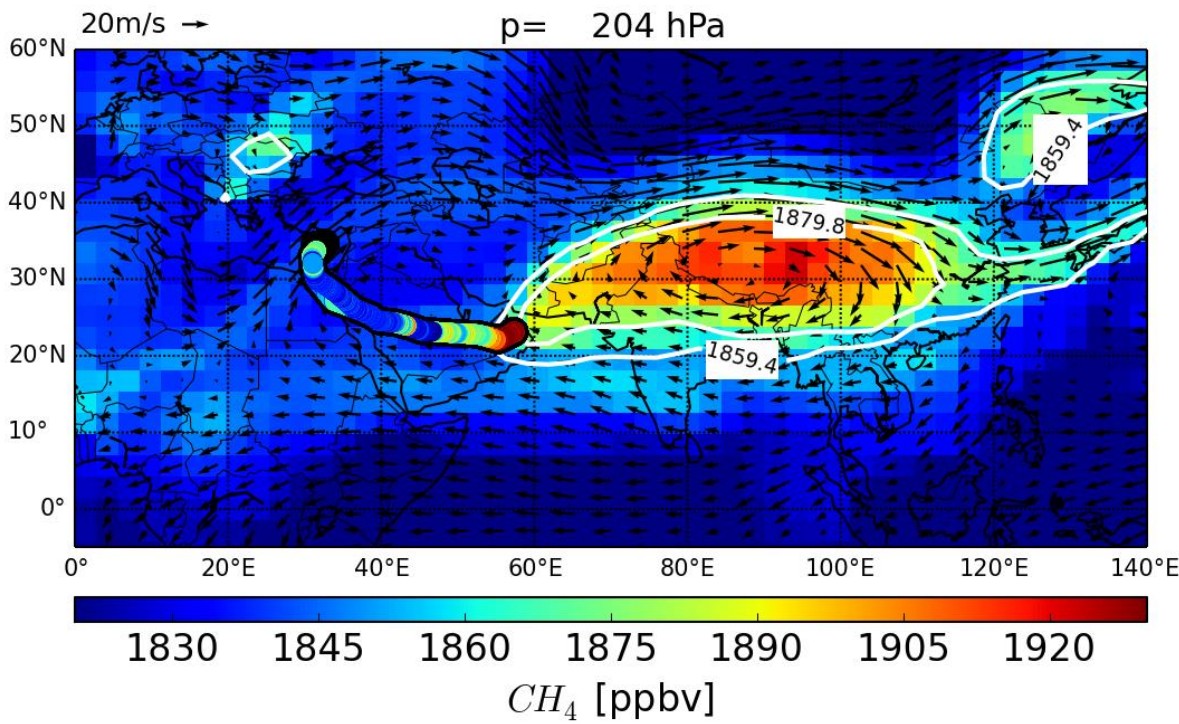

**Figure 7: EMAC modelled CH₄ and wind field; daily means at 204 hPa, and in situ CH₄ (above 300 hPa) along the flight track for flight 19 (August 13, 2015).White contours represent CH₄ threshold and background values according to section 3.1.**

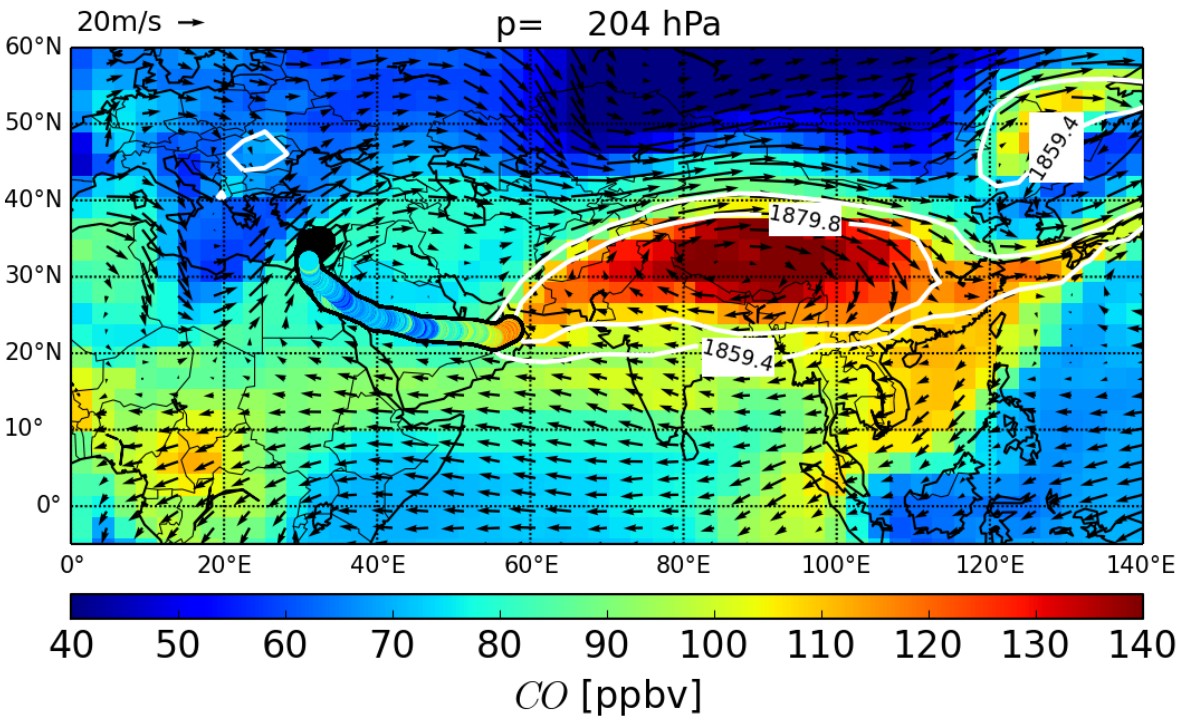

**Figure 8: EMAC modelled CO and wind field; daily mean at 204 hPa, and in situ CO (above 300 hPa) along the flight track for flight 19 (August 13, 2015).White contours represent CH₄ threshold and background values according to section 3.1.**

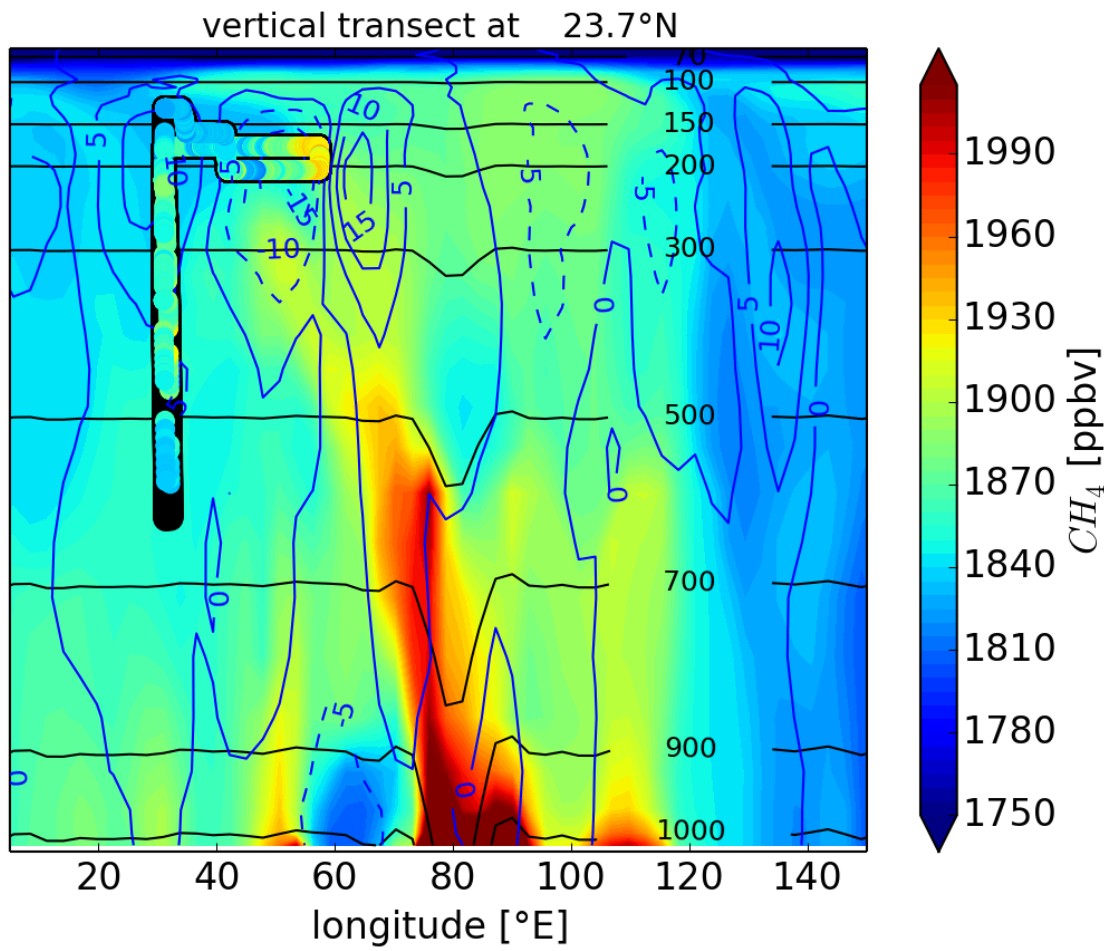

**Figure 9: EMAC modelled CH₄; daily mean transect along 23.7°N, and measured CH₄ along the aircraft track for flight 19. Additional EMAC pressure (black lines in hPa) and EMAC northward wind component (blue lines in m/s; southward wind dashed lines).**

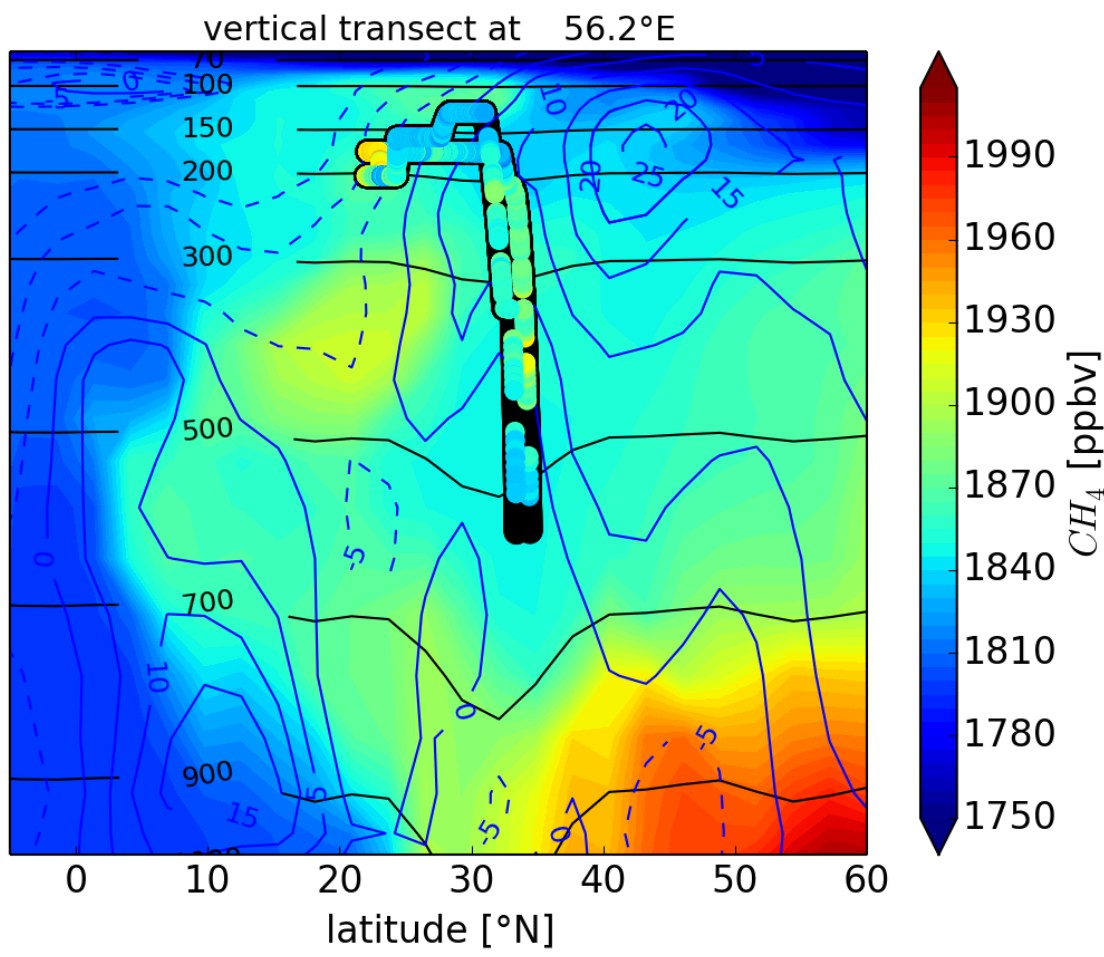

**Figure 10: EMAC calculated CH₄; daily mean transect along 56.2°E, and measured CH₄ along the aircraft track for flight 19. Additional EMAC pressure (black lines in hPa) and EMAC eastward wind component (blue lines in m/s; westward wind dashed lines).**

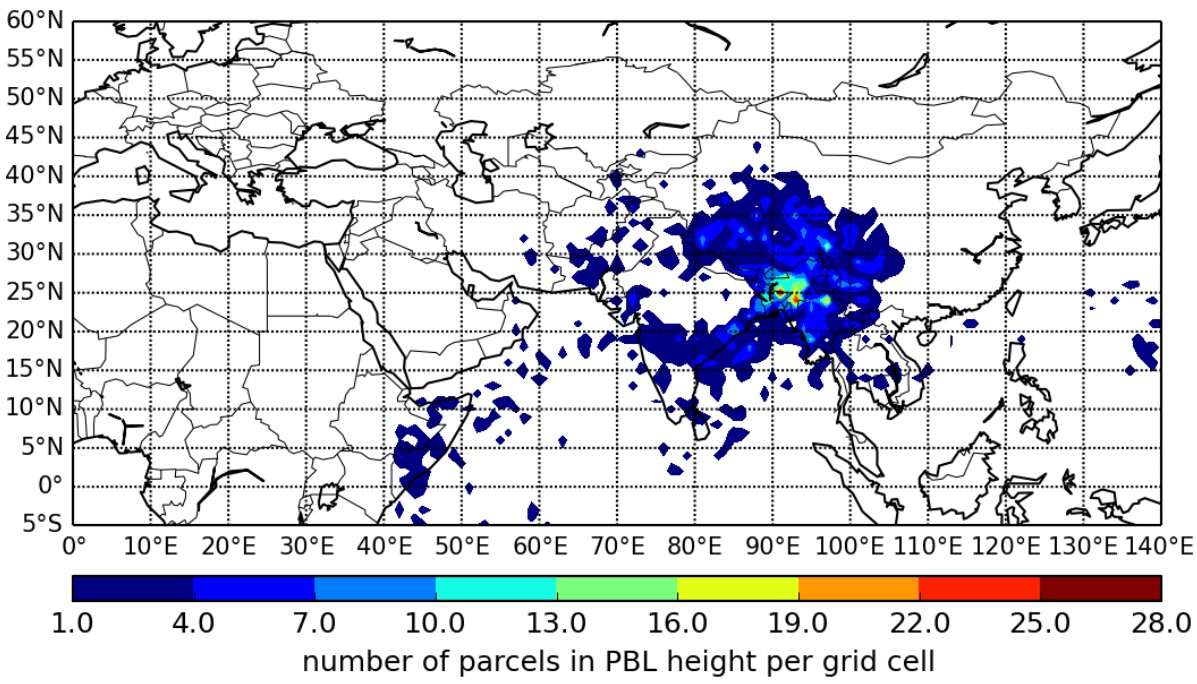

**Figure 11: Last boundary layer contact of parcels from trajectories, which start along the flight track at locations with CH$_4$ mixing ratios above the threshold, before they were transported to the track of flight 19 (10 days prior to flight).**

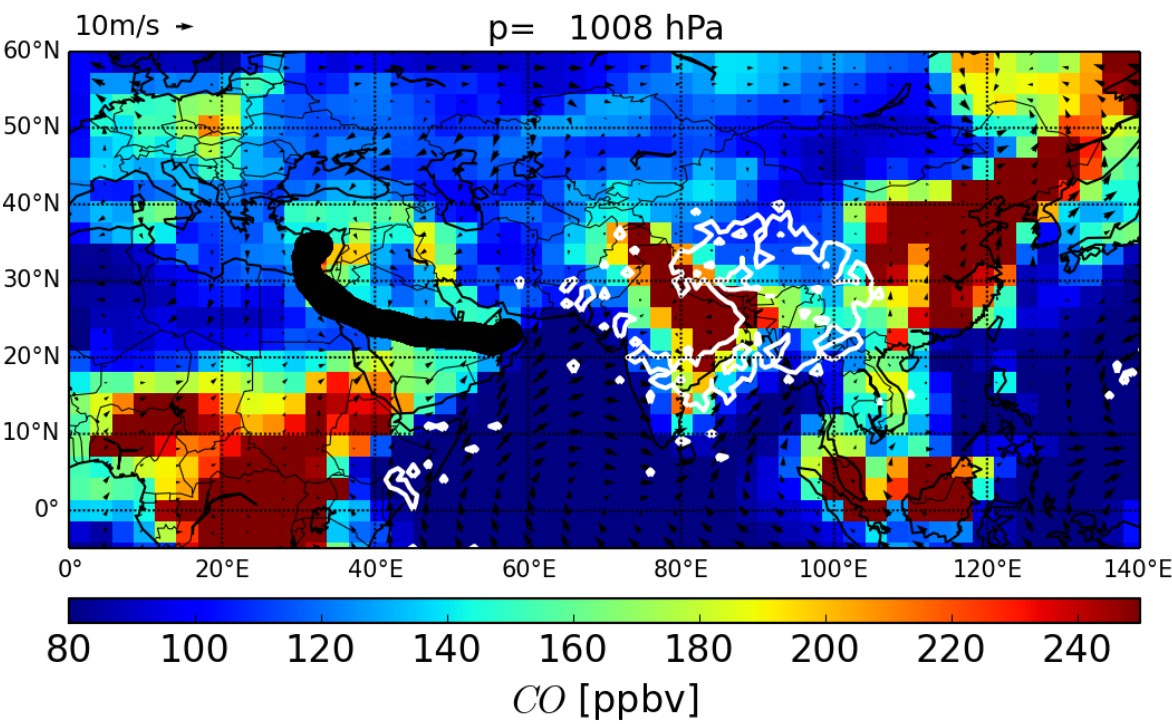

**Figure 12: EMAC calculated CO; daily mean at the surface (1008 hPa, August 03, 2015) as an indicator for surface emissions (10 days prior to measurement) and the flight track of flight 19 (black). Additionally, the footprint of the last boundary layer contact as white contour line from Figure 11.**

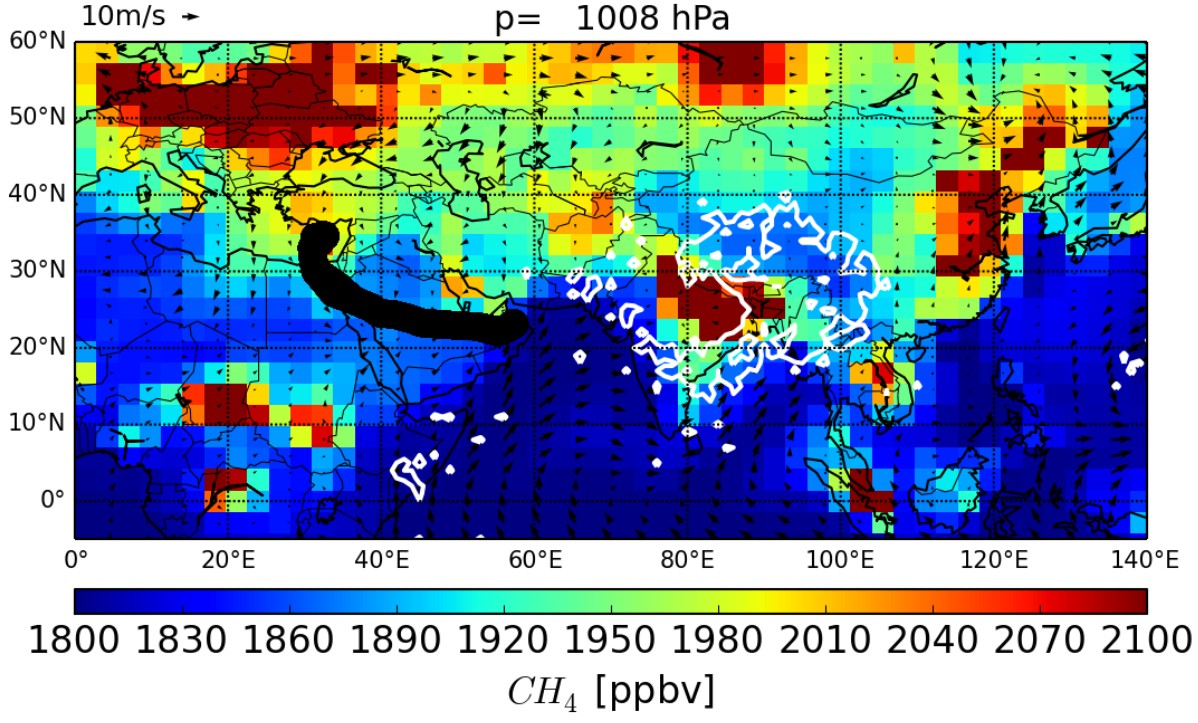

**Figure 13: EMAC calculated CH₄; daily mean at the surface (1008 hPa, August 03, 2015) as an indicator for surface emissions (10 days prior to measurements), and the track of flight 19 (black). Additionally, the footprint of the last boundary layer contact as white contour line from Figure 11.**

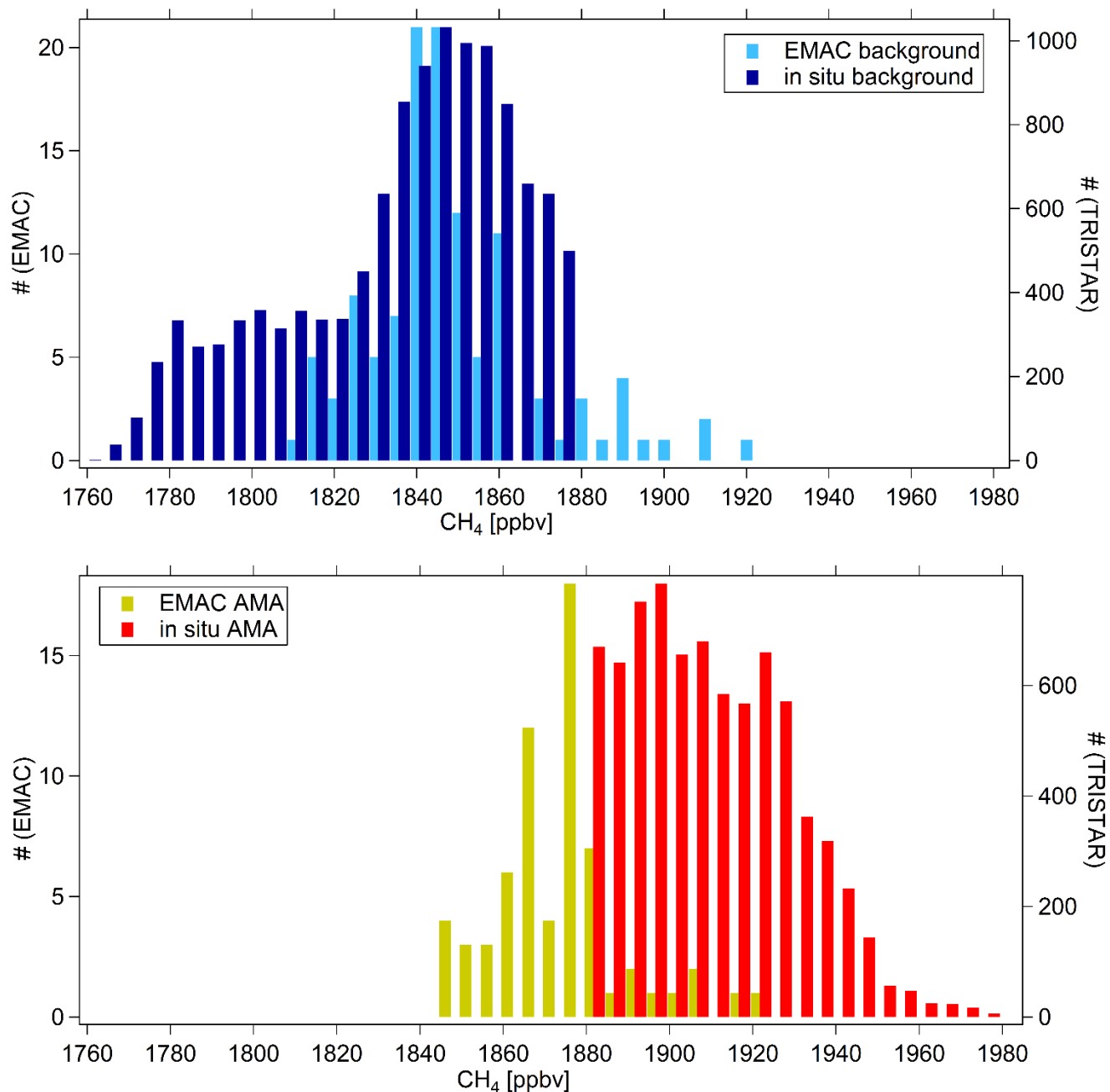

**Figure 14: Histogram for in situ measured and EMAC modelled CH₄ within the altitude range 300-140 hPa, both for background and AMA air.**

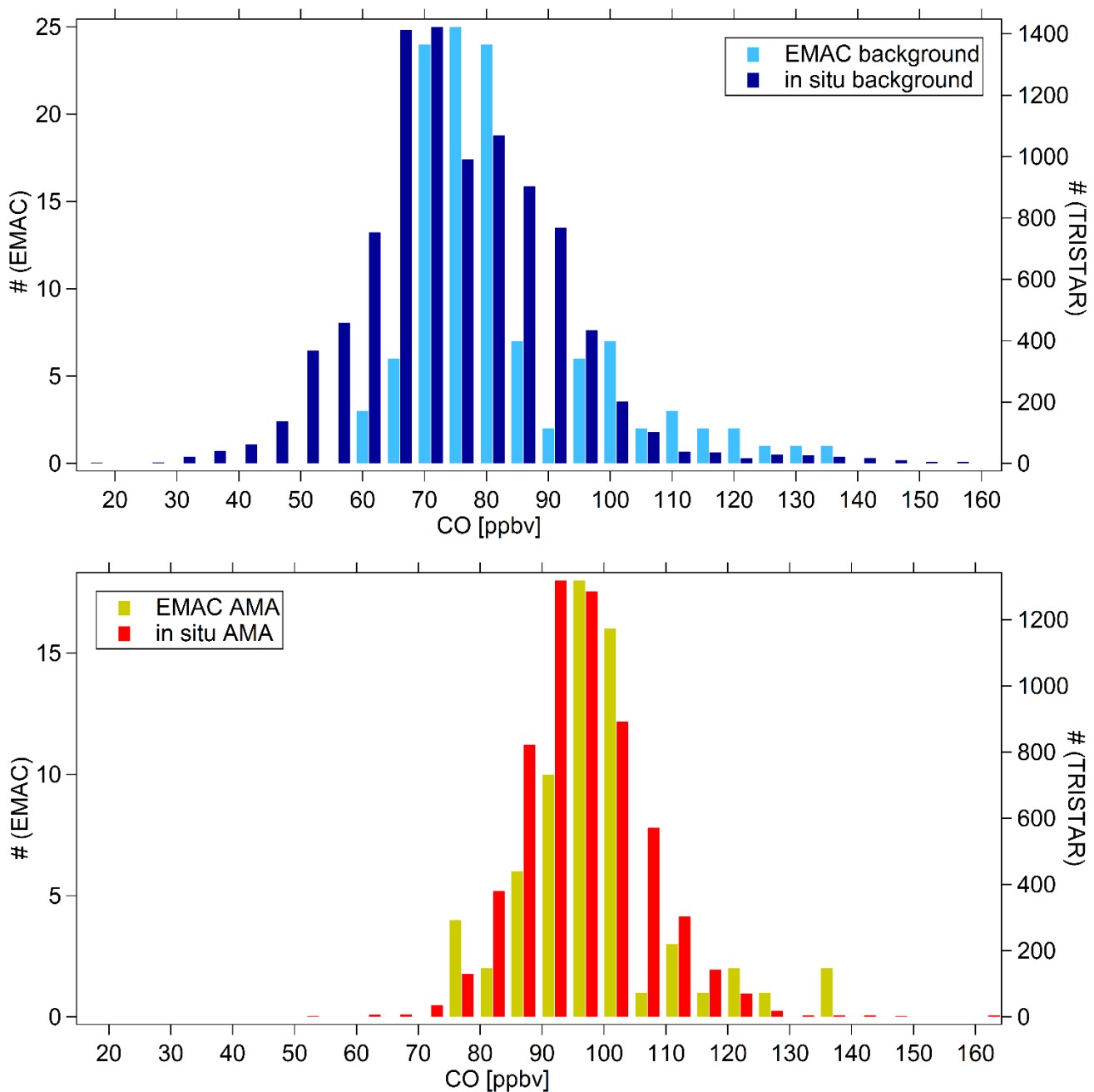

**Figure 15: Histogram for in situ measured and EMAC modelled CO within the altitude range 300-140 hPa, both for background and AMA air.**

**Table 1: CH4 and CO averages and standard deviations for in situ measured and EMAC data, both for background and monsoon influenced air masses according to the CH4 threshold for altitudes between 300-140 hPa.**

| p=[300-140] hPa | | CH4 [ppbv] | | CO [ppbv] | |
|---|---|---|---|---|---|
| | | avg | std | avg | std |
| monsoon | in situ | 1910.0 | 19.2 | 96.9 | 10.0 |
| | EMAC | 1874.4 | 15.3 | 99.0 | 11.9 |
| background | in situ | 1837.9 | 27.6 | 76.8 | 15.7 |
| | EMAC | 1850.5 | 21.2 | 84.3 | 15.1 |

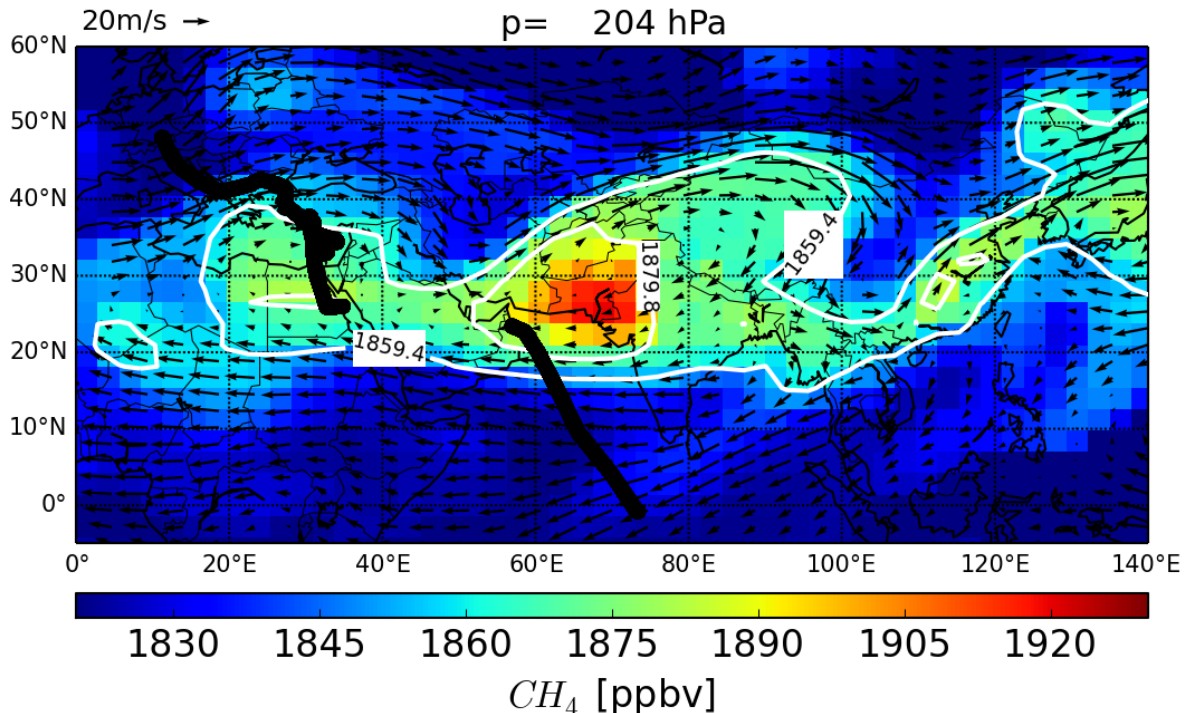

**Figure 16: Double anticyclone mode illustrated with wind field and CH4 EMAC daily means at 204 hPa (July 25, 2015) and the associated flight tracks. White contours represent CH4 threshold and background values according to section 3.1.**

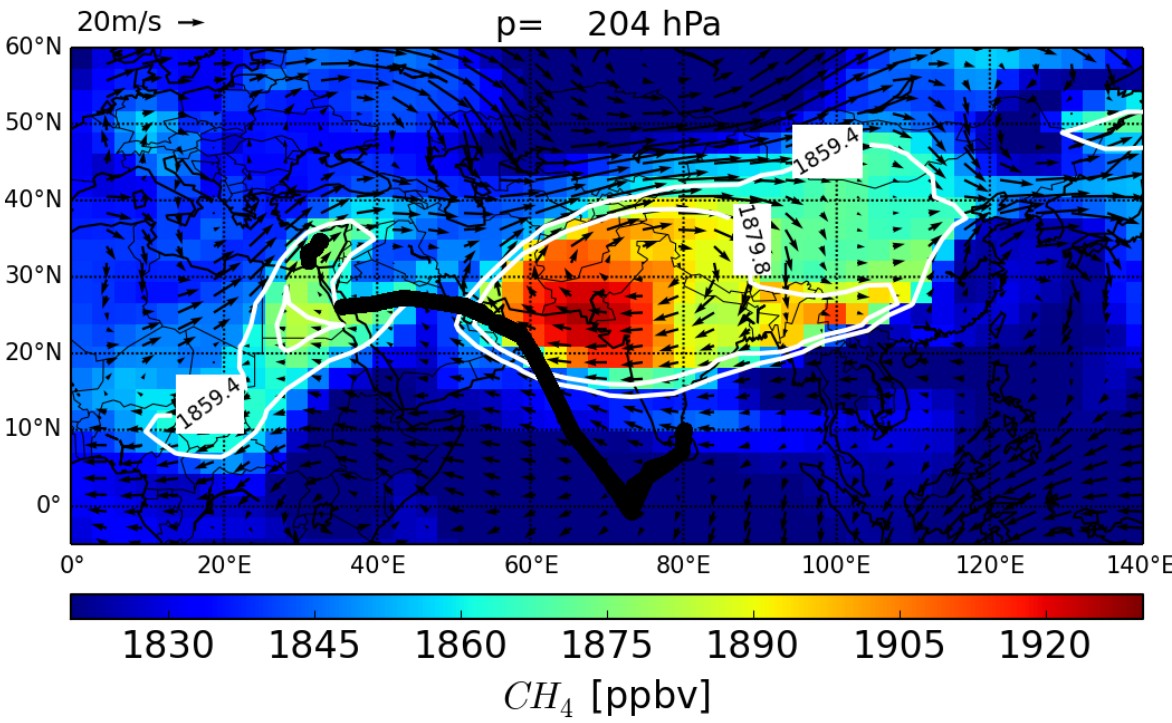

**Figure 17: Central anticyclone mode illustrated with wind field and CH₄ EMAC daily means at 204 hPa (August 09, 2015) and the associated flight tracks. White contours represent CH₄ threshold and background values according to section 3.1.**

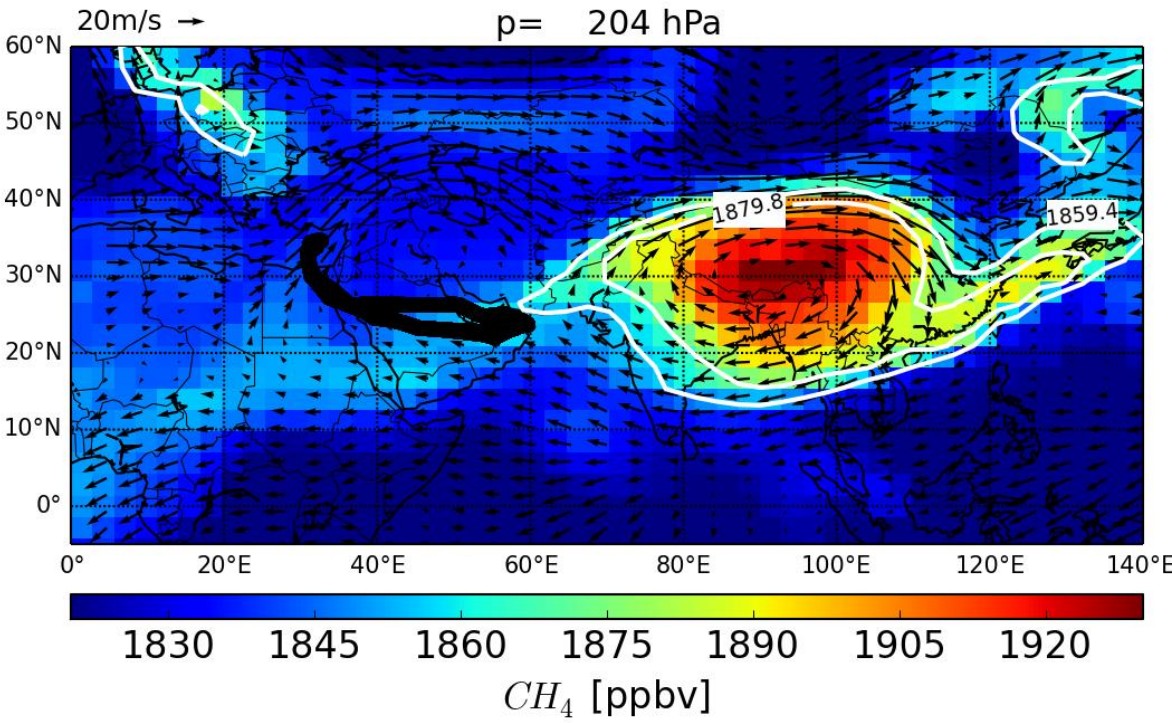

**Figure 18: Tibetan anticyclone mode illustrated with wind field and CH₄ EMAC daily means at 204 hPa (August 15, 2015) and the associated flight tracks. White contours represent CH₄ threshold and background values according to section 3.1.**

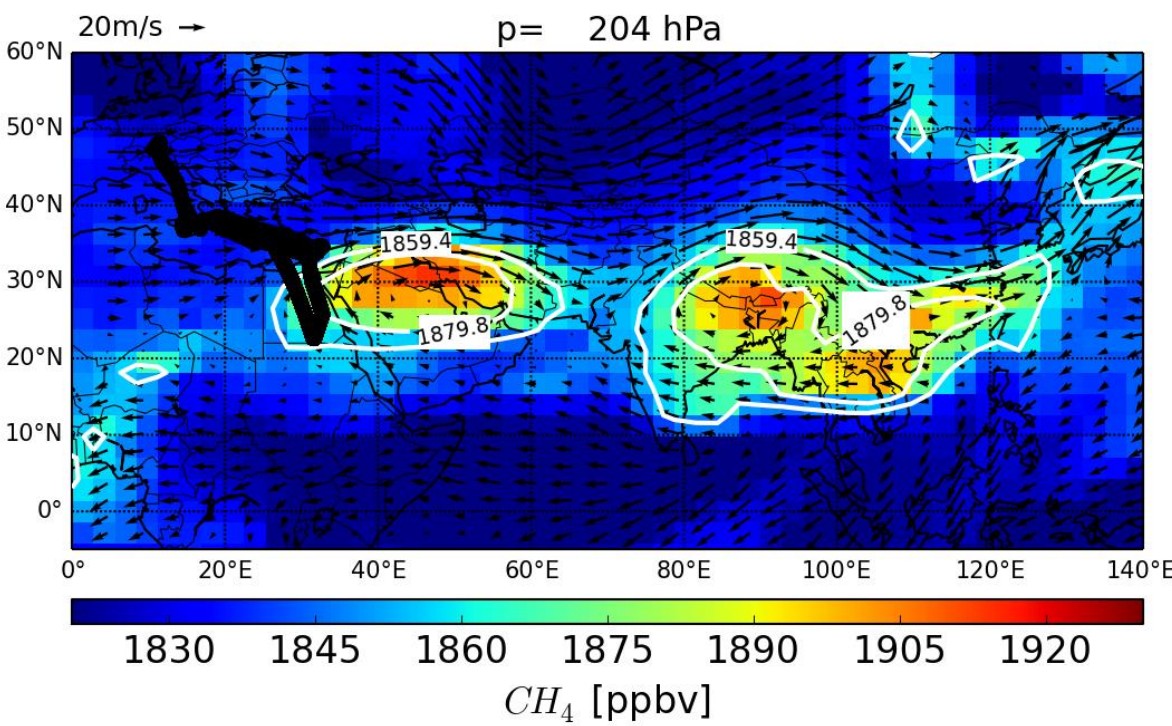

**Figure 19: Double anticyclone mode illustrated with wind field and CH₄ EMAC daily means at 204 hPa (August 25, 2015) and the associated flight tracks. White contours represent CH₄ threshold and background values according to section 3.1.**

**Table 2: In situ CO and CH₄ for the four different anticyclone situations. Differentiation between AMA and background for each flight between 300-140 hPa.**

| meteorological situation | flight no. | date | position relative to AMA | in situ at 300-140 hPa | | | | | | | |
| --- | --- | --- | --- | --- | --- | --- | --- | --- | --- | --- | --- |
| | | | | CO [ppbv] | | | | CH₄ [ppbv] | | | |
| | | | | background | sigma | monsoon | sigma | background | sigma | monsoon | sigma |
| double anticyclone | #8 | 21.07.2015 | partly in the western AMA | 67.8 | 8.7 | 89.8 | 7.4 | 1847.1 | 12.3 | 1898.6 | 7.8 |
| | #9 | 25.07.2015 | in the western AMA | 83.1 | 9.4 | 94.5 | 6.1 | 1870.0 | 11.4 | 1913.7 | 16.7 |
| | #10 | 28.07.2015 | in the western AMA | 76.1 | 16.4 | 91.4 | 5.1 | 1856.4 | 24.8 | 1896.4 | 12.4 |
| | #11 | 01.08.2015 | partly in residuals of the AMA | 92.8 | 6.8 | 108.6 | 4.5 | 1823.5 | 21.0 | 1889.0 | 4.8 |
| | | | | 80.0 | 10.3 | 96.1 | 5.8 | 1849.3 | 17.4 | 1899.4 | 10.4 |
| central mode | #12/13 | 06.08.2015 | in outflow region | 78.6 | 33.3 | 117.3 | 22.2 | 1827.4 | 26.8 | 1893.5 | 9.8 |
| | #14 | 08.08.2015 | in background south of the AMA | 76.3 | 8.0 | | | 1788.2 | 9.2 | | |
| | #15/16 | 09.08.2015 | at the south western edge | 77.5 | 12.0 | | | 1812.6 | 34.3 | | |
| | #17/18 | 10.08.2015 | at the south eastern edge and in outflow region | 76.5 | 7.9 | 98.3 | 7.8 | 1832.0 | 19.5 | 1909.3 | 15.0 |
| | | | | 77.2 | 15.3 | 107.8 | 15.0 | 1815.1 | 22.5 | 1901.4 | 12.4 |
| Tibetan mode | #19 | 13.08.2015 | at the western edge of the AMA | 74.7 | 10.4 | 99.4 | 13.8 | 1848.0 | 16.3 | 1907.3 | 20.8 |
| | #20 | 15.08.2015 | at the western edge of the AMA | | | | | 1855.2 | 11.6 | 1905.2 | 13.9 |
| | #21 | 18.08.2015 | in and outside the AMA | 87.9 | 16.3 | 104.8 | 9.8 | 1853.0 | 12.9 | 1917.1 | 20.6 |
| | | | | 81.3 | 13.4 | 102.1 | 11.8 | 1852.1 | 13.6 | 1909.9 | 18.4 |
| double anticyclone | #22 | 23.08.2015 | at the western edge of the western AMA | | | | | 1857.0 | 8.2 | 1927.9 | 22.6 |
| | #23 | 25.08.2015 | at the western edge of the western AMA | 65.7 | 12.4 | 93.8 | 7.6 | 1855.9 | 8.5 | 1926.4 | 21.0 |
| | #24 | 27.08.2015 | outside the AMA | | | | | 1853.7 | 14.6 | 1889.1 | 8.8 |
| | | | | 65.7 | 12.4 | 93.8 | 7.6 | 1855.5 | 10.4 | 1914.4 | 17.5 |

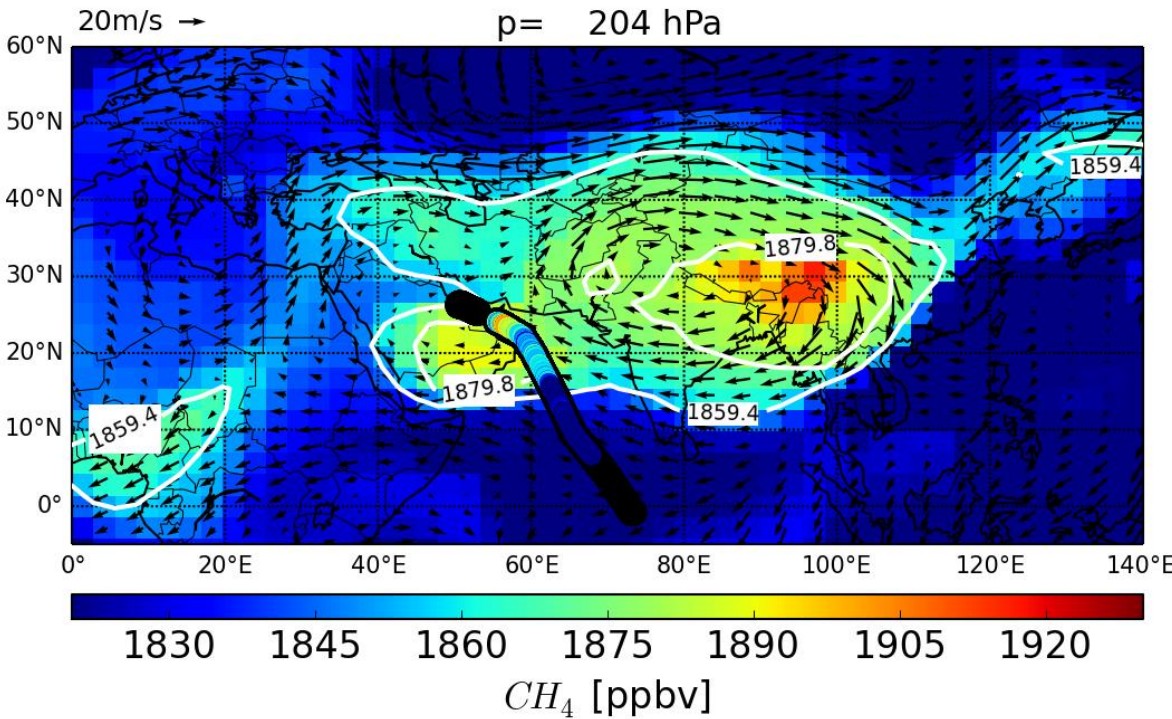

**Figure 20: EMAC calculated CH₄ and wind field; daily means at 204 hPa, and in situ CH₄ (above 300 hPa) along the aircraft track for flight 12/13 (August 06, 2015). White contours represent CH₄ threshold and background values according to section 3.1. The enhanced CH₄ values over Oman represent the outflow.**

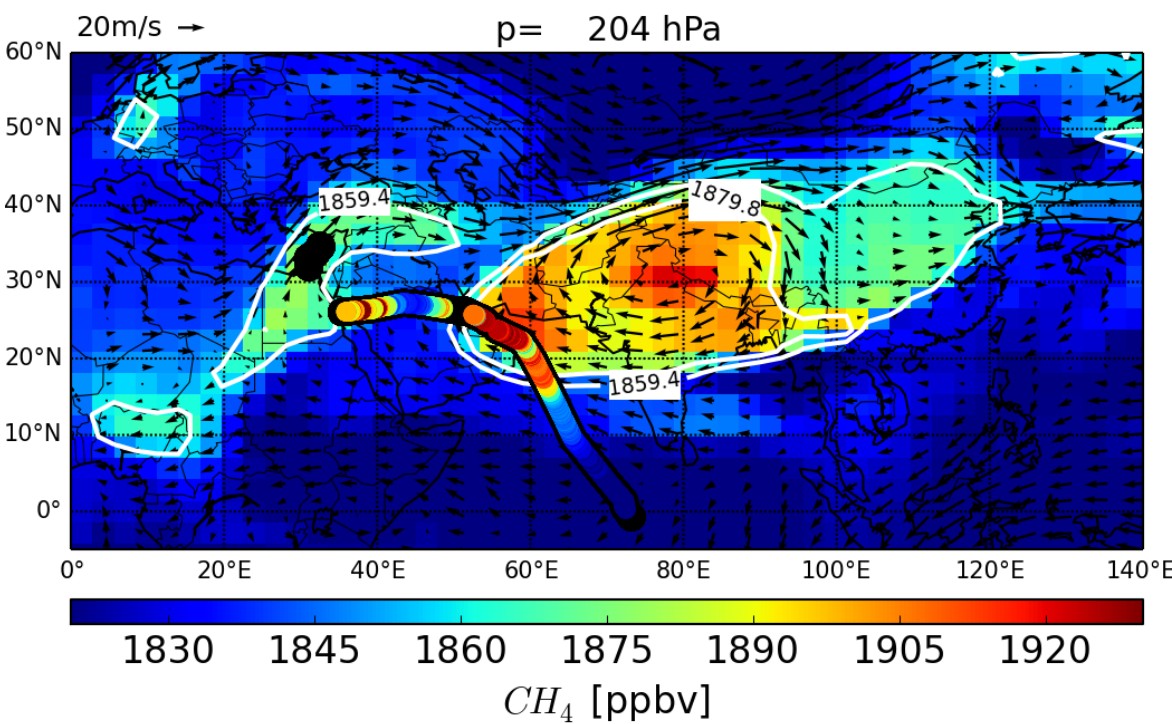

**Figure 21: EMAC calculated CH₄ and wind field; daily means at 204 hPa, and in situ measured CH₄ (above 300 hPa) along the aircraft track for flight 17/18 (August 10, 2015). White contours represent CH₄ threshold and background values according to section 3.1. The enhanced CH₄ values over the Read Sea represent the outflow.**

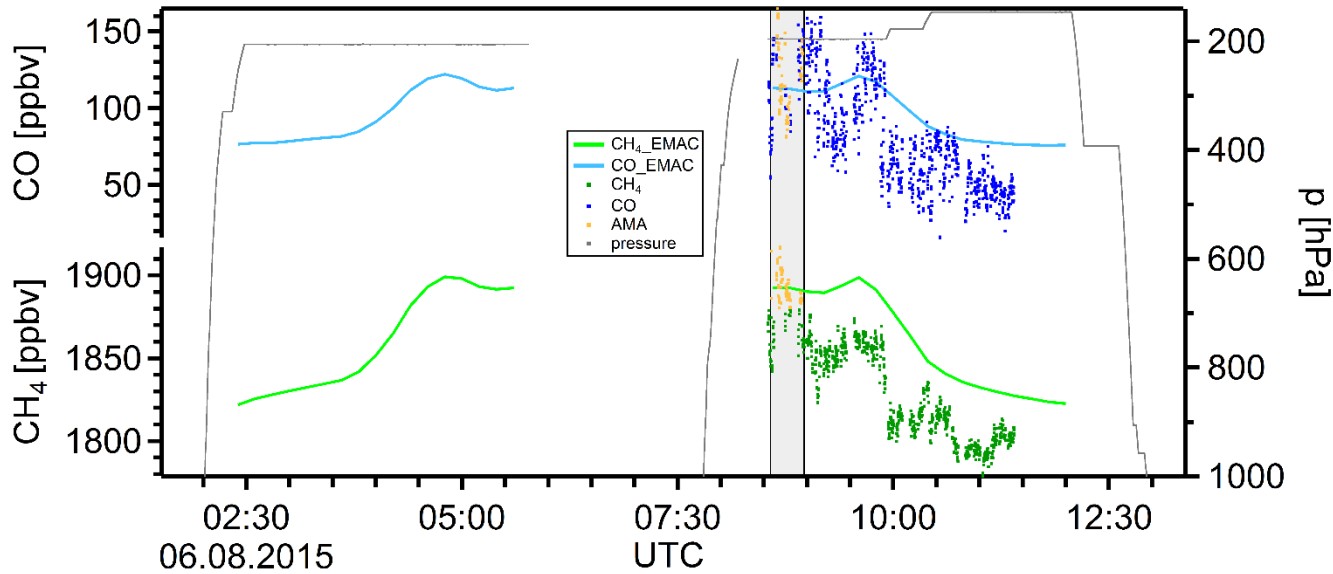

Figure 22: Flight 12/13 (August 6, 2015) in situ $CH_4$ and CO data and EMAC results along the flight track, as well as the flight altitude. The AMA is colour coded by $CH_4$>1879.8 ppbv.

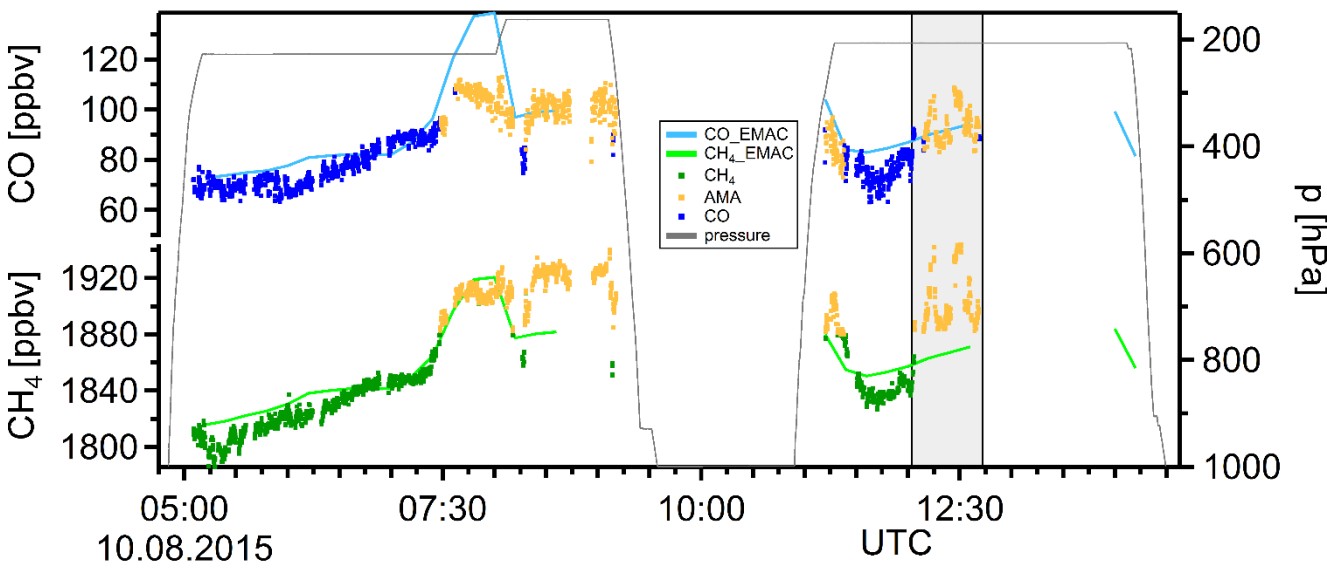

Figure 23: Flight 17/18 (August 10, 2015) in situ $CH_4$ and CO data and EMAC results along the flight track, as well as the flight altitude. The AMA is colour coded by $CH_4$>1879.8 ppbv.

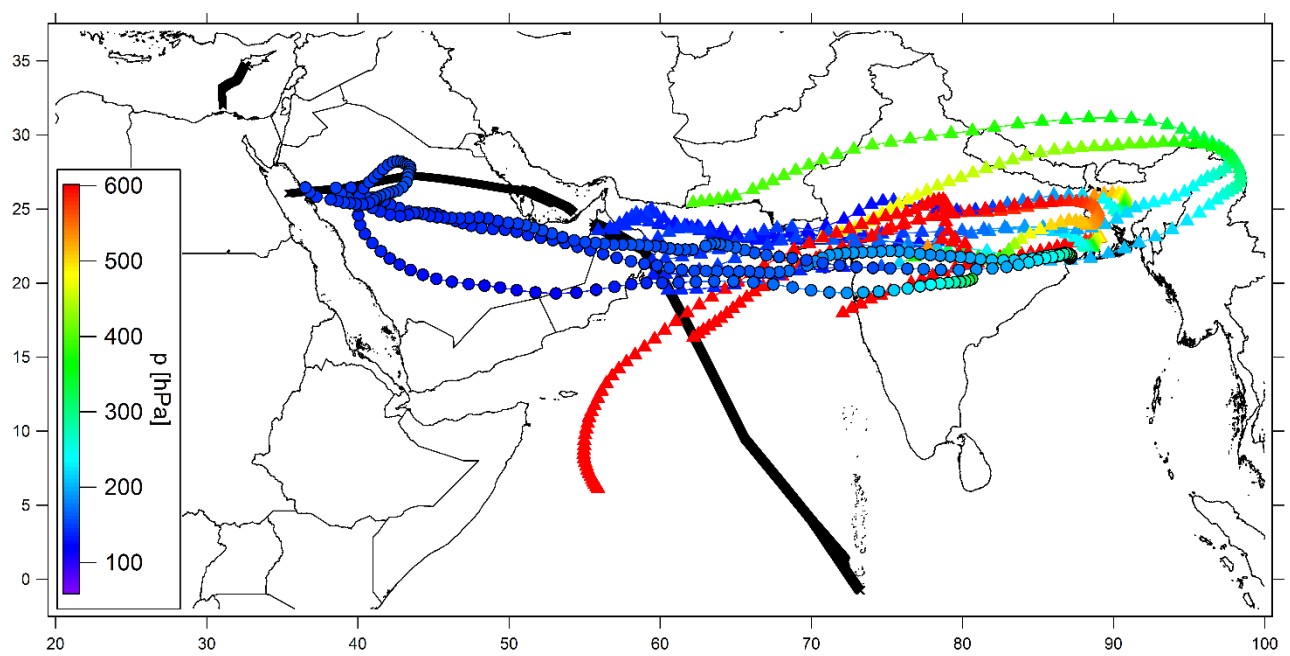

**Figure 24: Centroid back trajectories for enhanced CH₄ mixing ratios during flight 12/13 (triangles) and flight 17/18 (circles) with colour coded height.**

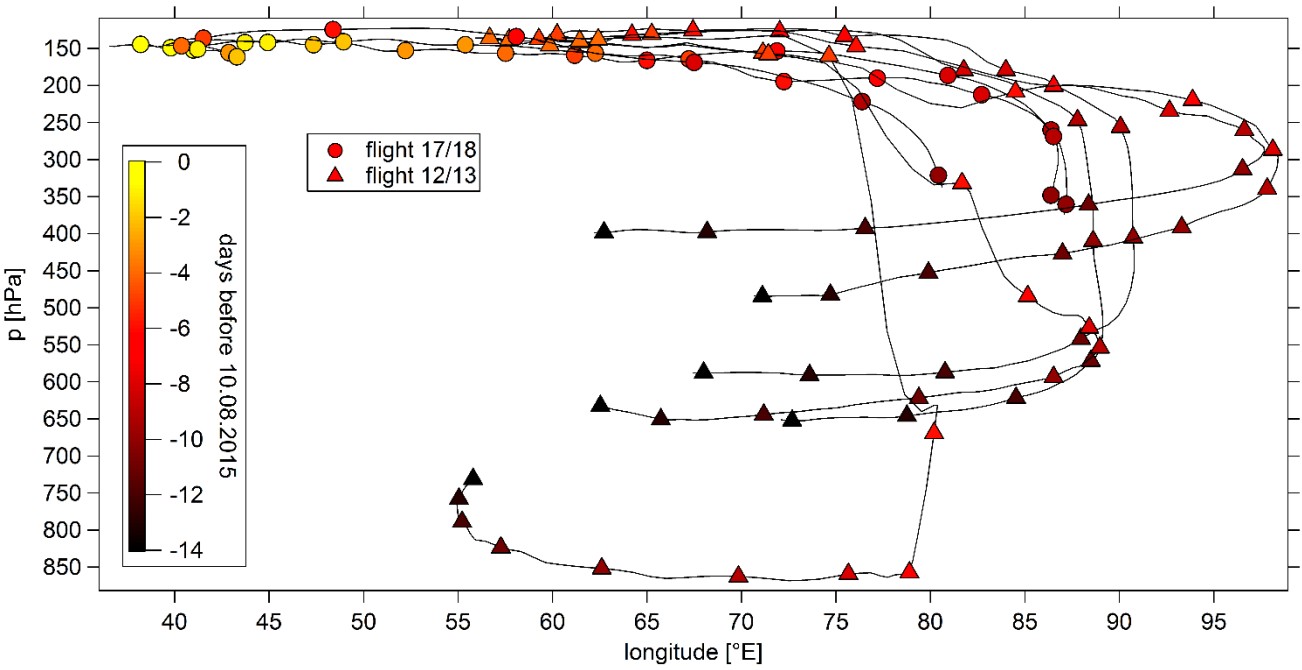

**Figure 25: Centroid back trajectories for enhanced CH₄ mixing ratios during flight 12/13 (triangles) and flight 17/18 (circles) with colour coded days before the release on August 10, 2015.**