# Peer review of "Upper tropospheric CH4 and CO affected by the South Asian summer monsoon during OMO"

_Atmospheric Chemistry and Physics, 2018_

## Referee Comment (RC1) · Anonymous Referee #1 · 29 Oct 2018

Summary

The manuscript by Tomsche et al. presents observations of CH4 and CO obtained during the OMO campaign in 2015 with the TRISTAR instrument on board the German High-Altitude and Long Range Research Aircraft HALO. Transport pathways and origin of trace gases are identified using FLEXPART, an Lagrangian dispersion model. Additionally the in situ data are compared with simulations carried out with the CCM EMAC.

The vertical profiles taken during the campaign show different altitude distributions depending on geographic locations. The authors classify different profiles as NH and SH background and AMA influenced and define an observation-based threshold value for CH4 and CO to distinguish between air masses from in and outside the AMA.

[Figure]

A case study for one flight during OMO (leading from Greece to Oman and back) is presented to demonstrate the interplay between surface emissions, deep convection and transport for the observed CH4 values. The observations showed enhanced values of CH4 and CO in air masses, which could be traced back with trajectory calculations to areas of strong emission in southeastern and eastern Asia. The trajectories showed upward transport via deep convection inside and at the edge of the Asian Monsoon Anticyclone (AMA), indicating strong impact of the dynamics of the AMA, thus making especially CH4 a good AMA tracer. Comparisons with results from EMAC show only a mediocre agreement, in particular the model underestimates the CH4 values and overestimates CO for air from within the AMA, while it overestimates both species in background air. Nevertheless, the large-scale dynamical situation seem to be represented quite good by EMAC (Table 1). Therefor the model is used in the following to identify different AMA modes and outflow events. Various measurement flights are then analyzed with respect to the dynamical situation and the relative position with respect to the AMA.

The paper shows the AMA as a distinct and persistent, although dynamically active feature observable in CH4 and CO. The dependency of the distributions of these tracers from the emission regions and the convective centres, as well as the influence of the relative position of the observations with respect to the AMA is clearly emphasized.

General

The present manuscript is well written and organized, but very extensive and sometimes too descriptive. Nevertheless the content presented is structured methodically and scientifically sound. The paper focuses on the analysis of observations of atmospheric tracers (CH4, CO) obtained during the OMO campaign carried out in 2015.

The tools used in addition to the statistical analysis are primarily two numerical models, namely FLEXPART, a Lagrangian dispersion model, and EMAC, a CCM widely used in the german atmospheric community. The FLEXPART trajectories are driven by

ECMWF operational data used to gain information about the transport pathways of the observed air masses. These data is used in a 1x1 degree horizontal resolution with the full vertical resolution of 137 levels. Vertical motion is calculated using a stochastic approach. Additionally moist convection is parameterized.

The EMAC model, on the other hand, has a much coarser resolution of 2.8x2.8 degrees and 90 levels. Unfortunately the EMAC simulation used is not described in detail, leaving open some important questions: Is EMAC used in an offline CTM mode? If this is the case, what is the model then driven by? Or, in other words: Do both models, the Lagrangian as well as the Eulerian model "see" the same background atmosphere? What kind of vertical velocity was used for the EMAC simulation. Another important point of course would be the initialization of the model, the length of the simulation and whether a certain spin-up time was necessary.

Since during the analysis of the data results from both models were used simultaneously (e.g. footprints and emission data) or observations of tracers obviously transported upward by convection are compared to distributions modified by vertical transport in EMAC, a more detailed description of the model setup would be very helpful. A very interesting diagnostic in this context would e.g. be the vertical transport time of tracers emitted from the surface to reach the 200 hPa level in EMAC.

The derivation of threshold values for CO and CH4 to distinguish between the inside of the monsoon anticyclone and the outside by using vertical profiles for NH and SH background and AMA leads to the question, why profiles over Egypt are considered as influenced by AMA and profiles over Cyprus are not. At least a look at the figures showing the different AMA modes (figures 18 to 21) would lead to a different expectation. But this is just judged by visual measure (and only on 204 hPa), so if there are distinct differences between profiles at these locations, the authors would be well advised to please show them. Since the classification of profiles influences the threshold values, this question may be quite important for the further analyses.

The observations shown for the case study for flight 19 indicate a highly structured CO and CH4 distribution in the vicinity of the AMA boundary region. The distributions simulated by EMAC matches the observations only very roughly. In particular the CH4 values are underestimated significantly. By looking at the horizontal and vertical distributions one gets the impression, that the vertical transport of the model is probably to weak. This may have several reasons: First, the vertical velocity may be to slow, e. g. the processes leading to strong updraft (namely convection) are to weak or insufficiently parameterized, or second, the numerical horizontal diffusion implied by the coarse grid resolution dampens the strong updraft plumes (approximately above 500 K). Adding horizontal wind as contour lines to the cross sections could shed some light on this problem. The included lines of potential temperature already point into this direction.

However, although the EMAC distributions may be consistent within the model, these effects may lead to a too small AMA region, when defined by an observational-based CH4 threshold. A dynamical shape of the AMA could be gained by using geopotential height or stream function. In this context I would suggest to add some contour lines to the figures displaying the horizontal CO and CH4 distribution including the threshold values and lower values to give a better visual feedback of the AMA and its position relatively to the flight tracks. This is meant with reference to figures 7, 8, 18 – 21.

A comparison between footprints of last PBL contact derived from 10 day backward trajectories from FLEXPART and the surface emissions from EMAC could be much more efficient, when footprints would be graphically added to the surface emission charts.

The analysis with respect to the different AMA modes defined by the CH4 distribution of the EMAC simulation leads to very interesting results, which are almost impossible to interpret from the values of table 2 without the knowledge of the flight tracks and the position of the AMA. Probably one could use the distance not to the anticyclonic centers but to the boundaries of the anticyclones. Nevertheless the discussion of the

results remains complex, and the authors do a good job here.

The last case study focusing on an outflow event tracked with trajectories and probed twice within 4 days seem to give better agreement with EMAC results (again only judged by visual measure). Maybe an additional figure showing observed and simulated tracer distributions would complement this very interesting manuscript. These plots are already in the supplement.

Summarizing, the paper is well-written and presents an important contribution to our understanding of transport. I recommend to accept the paper after some minor revisions noted in the text above. The most important one would be a more detailed description of the EMAC simulation with respect to the questions raised above.

Specific comments

Important: Please describe the EMAC simulation with respect to the above mentioned questions.

Important: Please be more specific on the reasons for the distinction between Cyprus and Egypt profiles.

Suggestion: Add horizontal wind as contour lines to the cross sections. Refers to figures 9-12.

Suggestion: Add some contour lines to the figures displaying the horizontal CO and CH4 distribution including the threshold values and lower values to give a better visual feedback of the AMA and its position relatively to the flight tracks. This is meant with reference to figures 7, 8, 18 – 21.

Suggestion: Add footprints graphically to the surface emission charts.

Suggestion: Add figures with observed and modeled CO and CH4 along the flight track for the outflow event case study.

---

## Referee Comment (RC2) · Anonymous Referee #2 · 29 Oct 2018

This study presents analyses of a very unique set of in-situ measurements of CO and CH4 to investigate transport pathways from the surface to the upper troposphere in the Asian summer monsoon region. The FLEXPART model calculations and simulations from a global chemistry transport model, EMAC, were also used in conjunction with the in-situ measurements during the monsoon season. The results presented here contains vast information with numerous figures depicting transport processes. I think the overall quality of this paper can be improved even further by redefining the science goals and reorganizing the results for more simplicity and clarity. I have a number of comments and suggestions for the authors might take into consideration.

General Comments

1. The goal of this study is not clearly stated. Is it to explore transport pathways inside

the anticyclone or in the vicinity? For instance, flight 19 suggests the measurements took place outside the anticyclone based on the boundaries estimated from the model simulations (Fig. 7 & 8). I think it is important to clarify the goal of this study and explain different transport pathways separately.

2. The abstract and introduction can be reorganized and refined. In introduction, brief background of the Asian monsoon anticyclone and its role in chemical transport in the UTLS region should be mentioned first. Then why in-situ measurements are so valuable but challenging and limited should be mentioned along with pros and cons of other data sources, including, satellite measurements. The purpose of utilizing two separate models should be emphasized. The key factors of OMO field campaign should be included with proper citations as well. Additionally, the goal of this paper and why this paper is unique compared to previous work should be mentioned clearly. Abstract of this paper should be a summary of what is shown in this work without including general statements. In the current form, most of the information exists without clearly stating what the goal of this paper is.

3. Section 2 (methods) should include general information about OMO field campaign, including its science goal. What other species were measured during the campaign? What were the science questions? Are there any references?

4. Section 3.5 (AMA mode) should include discussions of bimodal mode of the monsoon anticyclone shown in Zhang et al. (2002) and Nützel et al. (2016). Also, it should be justified why it is necessary to have four modes instead of two. Is bimodal distribution of the anticyclone wrong?

Specific Comments

1. The abstract includes a few general statements, which makes abstract sound rather like introduction. For instance, L9-11 (However. . .expected) can be removed.

2. P1, L7 – It is connected to -> It is part of the South Asian summer monsoon system

3. P1, L17-19 – Are those based on the in-situ measurements?

4. P1, L21 – areas within the upper troposphere -> areas in the upper troposphere

5. P2, L3 – Park et al. (2008) might be relevant here.

6. P2, L4 – within the strong. . .monsoon -> by the strong monsoon convection

7. P2, L5 – Park et al. (2007) might be relevant here.

8. P2, L9 – physical -> physically

9. P2, L17-18 – Full name for CARIBIC and IAGOS-MIZAIC should be provided here as well.

10. P3, L1- It is also important to mention that there is a big uncertainty in source estimates of methane (Bloom et al., 2017 GMD and references there in).

11. P3, L8 – 'variability of the AMA' can be explained more detail here.

12. P4, L8 (section 2.2) – I assume the trajectory calculations are done backward. Where is the initialization location?

13. P5, L11 (section 2.4) – The reason why MODIS cloud top pressure is used is missing. Is this used as convective proxy?

14. P5, L28 – I would like to know if there are any in-situ measurements of methane and if so how the mixing ratios compare with them even over different regions in different season.

15. P5, L30 – I have tried to find CO observations from satellite in Randel and Park (2006) but they seem to have used only ozone and water vapor.

16. P5, L31 – to identified -> to identify

17. P6, L8 -12 – This paragraph is not convincing to me without supporting material or references.

18. P6, L18 – This is in consistent -> This is consistent

19. L6, L20-22 – Do the mixing ratios of CO in the upper troposphere agree as well?

20. P6, L30-31- Does this problem prevented from using the measurement or only degraded the data quality of CO measurements?

21. P7, Eq. (1) – I think this threshold is somewhat subjective. At least it should be mentioned that this might introduce uncertainty in the analyses and also how sensitive the results are depending on the threshold values.

22. P7, L28 – Does the difference between Scheeren et al. (2003) and this study agrees with the values in Zimmermann et al. (2018) quantitatively?

23. P8, L13 – cloud top height pressure -> cloud top pressure (also in P10, L23)

24. P8, L16-17 – This sentence should be revised for clarity.

25. P8, L34 – high pressure -> anticyclonic

26. P9, L13 (Figs. 7 & 8) – Here, it looks like the flight path is outside the anticyclone based on the model simulations. The high values from the flight almost should be at the center of the anticyclone. I am not sure how to understand those comparisons.

27. P15, L29 – Instead of 'these transport' describe specific transport processes here.

---

## Author Response (AR2)

**Author's response to reviewer comments for Tomsche et al. (2018)**

-Reviewer 1:

The EMAC model, on the other hand, has a much coarser resolution of 2.8x2.8 degrees and 90 levels. Unfortunately the EMAC simulation used is not described in detail, leaving open some important questions: Is EMAC used in an offline CTM mode? If this is the case, what is the model then driven by? Or, in other words: Do both models, the Lagrangian as well as the Eulerian model "see" the same background atmosphere?

What kind of vertical velocity was used for the EMAC simulation? Another important point of course would be the initialization of the model, the length of the simulation and whether a certain spin-up time was necessary.

Since during the analysis of the data results from both models were used simultaneously (e.g. footprints and emission data) or observations of tracers obviously transported upward by convection are compared to distributions modified by vertical transport in EMAC, a more detailed description of the model setup would be very helpful. A very interesting diagnostic in this context would e.g. be the vertical transport time of tracers emitted from the surface to reach the 200 hPa level in EMAC.

Author:

We thank the reviewer for pointing out the lack of information regarding the model simulation. Here additional details are given, also added to manuscript.

Authors changes in manuscript:

P5-6 L30-9: The EMAC model was not run in an offline CTM mode, as the radiation calculations were based on simulated GHGs concentrations. Nevertheless, the model was weakly nudged towards ECMWF ERA Interim data (Jeuken et al., 1996) and therefore reproduced very similar dynamics to the ECMWF model (although not binary identical). The simulation is an extension of simulation RC1SD-base-10 (Jöckel et al. 2016) so to cover the full OMO campaign. Few changes to the original simulation have been applied (i.e. increased South Asia $SO_2$ emissions and reduced lightning $NO_x$), as described in Lelieveld et al. (2018). Although the simulation is the continuation of a well evaluated experiment, the simulation was running from March 1st, 2015 so to give time to the $SO_2$ and $NO_x$ to balance to the new emissions (i.e. 4 months spin up time). Only the data from July and August 2015, which covers the field campaign is actually used. The EMAC model is a hydrostatic model and the convective transport is parameterized (Ouwersloot et al. 2015, Tost et al. 2006). Indication of the vertical transport time in EMAC can be found in Krol et al. (2018), where also a comparison with model of similar complexity is shown.

-Reviewer 1:

The derivation of threshold values for CO and $CH_4$ to distinguish between the inside of the monsoon anticyclone and the outside by using vertical profiles for NH and SH background and AMA leads to the question, why profiles over Egypt are considered as influenced by AMA and profiles over Cyprus are not. At least a look at the figures showing the different AMA modes (figures 18 to 21) would lead to a different expectation. But this is just judged by visual measure (and only on 204 hPa), so if there are distinct differences between profiles at these locations, the authors would be well advised to please show them. Since the classification of profiles influences the threshold values, this question may be quite important for the further analyses.

Author:

In the classification of the profiles used for the calculation of the Northern hemisphere background and AMA-influenced air masses, respectively, and not only the geographical location but also the meteorological context have been accounted for. The profiles over Egypt were sampled during the second double anticyclone mode, with the westerly part of the anticyclone extending over Egypt. Profiles over Paphos were obtained over a longer period, representing background conditions but partly also AMA-influenced air masses. We calculated the NH background with and without profiles over Paphos. For profiles only over Oberpaffenhofen and Etna the average $CH_4$ mixing ratio is $1871.2\pm9.2$ ppbv and for profiles over Oberpfafffenhofen, Etna, and Paphos the $CH_4$ average is $1863.4\pm14.0$ ppbv. Thus the profiles with and without Paphos profiles agree within their standard deviation. Due to a better statistics, we used the NH background profile including profiles over Paphos.,

Authors changes in manuscript:

P7 L5-7: As observed, the CO and $CH_4$ profiles measured during OMO indicate different altitude distributions depending on the geographical location and partly also on the meteorological situation, especially for Paphos and Egypt. Profiles over Egypt were measured when the AMA extended over this region. Profiles over Paphos were sampled during periods with and without the AMA being positioned over Cyprus.

-Reviewer 1:

The observations shown for the case study for flight 19 indicate a highly structured CO and $CH_4$ distribution in the vicinity of the AMA boundary region. The distributions simulated by EMAC matches the observations only very roughly. In particular the $CH_4$ values are underestimated significantly. By looking at the horizontal and vertical distributions one gets the

impression that the vertical transport of the model is probably too weak. This may have several reasons: First, the vertical velocity may be too slow, e. g. the processes leading to strong updraft (namely convection) are too weak or insufficiently parameterized, or second, the numerical horizontal diffusion implied by the coarse grid resolution dampens the strong updraft plumes (approximately above 500 K). Adding horizontal wind as contour lines to the cross sections could shed some light on this problem. The included lines of potential temperature already point into this direction.

Author:

Indeed, the referee is correct in mentioning a possible too low transport of methane and carbon monoxide as a reason for underestimation in the upper troposphere. As shown by Krol et al. (2018), EMAC seems to have a weaker transport of surface tracers than other models. Both reasons suggested by the referee are possible, and it is difficult (if not impossible) to really distinguish the real reason for the underestimation of the transport. Nevertheless we would like to notice that for the comparison of CO with the model, the results are in line with other literature studies at such resolution (e.g. Baret et al., 2016). Horizontal wind components are added to the cross sections in figures 9-12, in detail: eastward wind component in cross sections along a longitude and northward wind component in cross sections along a latitude.

Authors changes in manuscript:

P11 L13-18: The simulated CO pattern, especially the enhanced values over Oman, fits well to the observed CO mixing ratios along the flight track. The EMAC model underestimates $CH_4$ and CO in the upper troposphere. As shown by Krol et al. (2018), EMAC seems to have a weaker transport of surface tracers than other models. There are two potential reasons for that, but it is difficult to distinguish them. First, a too slow vertical velocity, thus the convective updraft is too ineffective, or second, the numerical diffusion implied by the coarse resolution restricts the updraft too strong. Nevertheless we would like to notice that for the comparison of CO with the model, the results are in line with other literature studies at such resolution (e.g. Baret et al., 2016).

Horizontal wind components are added to the cross sections in Figures 9-12 (P35-41)

-Reviewer 1:

However, although the EMAC distributions may be consistent within the model, these effects may lead to a too small AMA region, when defined by an observational-based $CH_4$ threshold. A dynamical shape of the AMA could be gained by using

geopotential height or stream function. In this context I would suggest to add some contour lines to the figures displaying the horizontal CO and CH$_4$ distribution including the threshold values and lower values to give a better visual feedback of the AMA and its position relatively to the flight tracks.

Author:

5    In Figures 7,8, 18-21 and also in Figures 22 and 23 now contour lines are added for the CH$_4$ threshold (1879.8 ppbv) and the CH$_4$ background (1859.4 ppbv) values according to the calculation of the CH$_4$ threshold in section 3.1. In the horizontal CO distribution also the CH$_4$ threshold is added. Now the position of the AMA is easier to identify with respect to the flight tracks.

Authors changes in manuscript:

10    Figures 7,8,(P31-32), 18-21 (P42-45), 22 (P47),23 (P48)

-Reviewer 1:

A comparison between footprints of last PBL contact derived from 10 day backward trajectories from FLEXPART and the surface emissions from EMAC could be much more efficient, when footprints would be graphically added to the surface

15    emission charts.

Author:

Footprint is now added as white contour lines for the number of particles per grid cell = 2 to the surface emission charts for CH$_4$ and CO (Figures14 and 15).

Authors changes in manuscript:

20    Figure 14,15 (P38 and P 39)

-Reviewer 1:

The analysis with respect to the different AMA modes defined by the CH4 distribution of the EMAC simulation leads to very interesting results, which are almost impossible to interpret from the values of table 2 without the knowledge of the

flight tracks and the position of the AMA. Probably one could use the distance not to the anticyclonic centers but to the boundaries of the anticyclones.

Author:

In Table 2 we add a column for the relative position to the AMA, which is quite descriptive. As most of the flight tracks are in and outside the AMA a more detailed geographical location with respect to the AMA can be realized better in a graphical way. Thus we added for each flight in the supplement the $CH_4$ threshold (1879.8ppbv) for the AMA-influence and the background value (1859.4ppbv) as contour lines in the EMAC $CH_4$ and CO distributions as already done in the manuscript, e.g. Figure 7 and 8 for flight 19. In these plots the position of the flight track with respect to the AMA is more obvious.

Authors changes in manuscript:

Column added in table 2.P46

**Table 2: In situ CO and CH₄ for the four different anticyclone situations. Differentiation between AMA and background for each flight between 300-140 hPa.**

| meteorological situation | flight no. | date | position relative to AMA | in situ at 300-140 hPa | | | | | | | |
|---|---|---|---|---|---|---|---|---|---|---|---|
| | | | | CO [ppbv] | | | | CH$_4$ [ppbv] | | | |
| | | | | background | sigma | monsoon | sigma | background | sigma | monsoon | sigma |
| double anticyclone | #8 | 21.07.2015 | partly in the western AMA | 67.8 | 8.7 | 89.8 | 7.4 | 1847.1 | 12.3 | 1898.6 | 7.8 |
| | #9 | 25.07.2015 | in the western AMA | 83.1 | 9.4 | 94.5 | 6.1 | 1870.0 | 11.4 | 1913.7 | 16.7 |
| | #10 | 28.07.2015 | in the western AMA | 76.1 | 16.4 | 91.4 | 5.1 | 1856.4 | 24.8 | 1896.4 | 12.4 |
| | #11 | 01.08.2015 | partly in residuals of the AMA | 92.8 | 6.8 | 108.6 | 4.5 | 1823.5 | 21.0 | 1889.0 | 4.8 |
| | | | | 80.0 | 10.3 | 96.1 | 5.8 | 1849.3 | 17.4 | 1899.4 | 10.4 |
| central mode | #12/13 | 06.08.2015 | in outflow region | 78.6 | 33.3 | 117.3 | 22.2 | 1827.4 | 26.8 | 1893.5 | 9.8 |
| | #14 | 08.08.2015 | in background south of the AMA | 76.3 | 8.0 | | | 1788.2 | 9.2 | | |
| | #15/16 | 09.08.2015 | at the south western edge | 77.5 | 12.0 | | | 1812.6 | 34.3 | | |
| | #17/18 | 10.08.2015 | at the south eastern edge and in outflow region | 76.5 | 7.9 | 98.3 | 7.8 | 1832.0 | 19.5 | 1909.3 | 15.0 |
| | | | | 77.2 | 15.3 | 107.8 | 15.0 | 1815.1 | 22.5 | 1901.4 | 12.4 |
| Tibetan mode | #19 | 13.08.2015 | at the western edge of the AMA | 74.7 | 10.4 | 99.4 | 13.8 | 1848.0 | 16.3 | 1907.3 | 20.8 |
| | #20 | 15.08.2015 | at the western edge of the AMA | | | | | 1855.2 | 11.6 | 1905.2 | 13.9 |
| | #21 | 18.08.2015 | in and outside the AMA | 87.9 | 16.3 | 104.8 | 9.8 | 1853.0 | 12.9 | 1917.1 | 20.6 |
| | | | | 81.3 | 13.4 | 102.1 | 11.8 | 1852.1 | 13.6 | 1909.9 | 18.4 |
| double anticyclone | #22 | 23.08.2015 | at the western edge of the western AMA | | | | | 1857.0 | 8.2 | 1927.9 | 22.6 |
| | #23 | 25.08.2015 | at the western edge of the western AMA | 65.7 | 12.4 | 93.8 | 7.6 | 1855.9 | 8.5 | 1926.4 | 21.0 |
| | #24 | 27.08.2015 | outside the AMA | | | | | 1853.7 | 14.6 | 1889.1 | 8.8 |
| | | | | 65.7 | 12.4 | 93.8 | 7.6 | 1855.5 | 10.4 | 1914.4 | 17.5 |

-Reviewer 1:

The last case study focusing on an outflow event tracked with trajectories and probed twice within 4 days seem to give better agreement with EMAC results (again only judged by visual measure). Maybe an additional figure showing observed and simulated tracer distributions would complement this very interesting manuscript.

Author:

According to the suggestion of the reviewer the CO and $CH_4$ distributions along the flight track as a time line are added in Figures 24 and 25. The trace gas mixing ratios (observed and simulated) show clear enhancement due to the outflow event for both flights (flight 12/13 and flight 17/18). The outflow regions are marked in grey in the Figures. Additionally, we add in the manuscript the average CO and CH4 mixing ratios calculated from EMAC for the outflow periods for both flights (flight 12/13: CO=112.2±1.2 ppbv and $CH_4$=1891.7±1.2 ppbv and flight 17/18: CO=90.8±3.1 ppbv and $CH_4$=1864.6±5.9 ppbv) for a better comparison with the measured data in the outflow.

Authors changes in manuscript:

Figures 24 and 25 added to manuscript P49

P17 L18-22: In the air mass CO and $CH_4$ mixing ratios increased to 117.3±22.2 ppbv and 1893.5±9.8 ppbv, respectively (background: CO=78.6±33.3 ppbv and $CH_4$=1827.4±26.8 ppbv), which can be seen in Figure 24. The second probing of this air mass took place at August 10 (flight 17/18, Figure 23) over the Red Sea yielding mixing ratios of 94.2±6.8 ppbv and 1903.7±19.2 ppbv. This corresponds to the increase at around 12-13 UTC in Figure 25.

P17 L25-28: Comparing the EMAC simulations with the in situ data along the flight tracks (Figure 24 and 25), the trends for the outflow agree. The EMAC average mixing ratios for CO and $CH_4$ are 112.2±1.2 ppbv and $CH_4$=1891.7±1.2 ppbv for flight 12/13 and 90.8±3.1 ppbv and $CH_4$=1864.6±5.9 ppbv for flight 17/18, respectively. Thus also the values agree within their standard deviation beside $CH_4$ in flight 17/18, where the outflow is underestimated by the model.

[Figure]

**Figure 24: Flight 12/13 (August 6, 2015) in situ CH₄ and CO data and EMAC results along the flight track, as well as the flight altitude. The AMA is colour coded by CH₄>1879.8 ppbv. Outflow region is marked in grey.**

[Figure]

**Figure 25: Flight 17/18 (August 10, 2015) in situ CH₄ and CO data and EMAC results along the flight track, as well as the flight altitude. The AMA is colour coded by CH₄>1879.8 ppbv. Outflow region is marked in grey.**

-Reviewer 2:

The goal of this study is not clearly stated. Is it to explore transport pathways inside the anticyclone or in the vicinity? For instance, flight 19 suggests the measurements took place outside the anticyclone based on the

boundaries estimated from the model simulations (Fig. 7 & 8). I think it is important to clarify the goal of this study and explain different transport pathways separately.

Author:

The goal of the present study is to understand the transport pathways from the source regions into the upper troposphere via the convective uplift into the AMA and further within the UT, especially towards the southern and western areas of the AMA. The transport pathways in the UT include the transport along the edges of the AMA, the circulation in the AMA where air masses are trapped and the transport across the AMA edges, and the outflow out of the anticyclone due to instabilities in the strong circulation. For instance, flight 19 took place outside and at the western edge of the AMA, which is now better visible in Figures 7 and 8 due to addition of a contour line for the $CH_4$ threshold.

Authors changes in manuscript: Adapted Figures 7 and 8

P2-3 L32-3: The measurement campaign OMO (Oxidation Mechanism Observations) took place in July/August 2015 with the German High Altitude and Long range (HALO) research Aircraft, performing flights at altitudes between 11 km and 15 km over the above-mentioned regions to investigate the dynamics and atmospheric chemistry in the upper troposphere over five weeks during the monsoon season.

-Reviewer 2:

In introduction, brief background of the Asian monsoon anticyclone and its role in chemical transport in the UTLS region should be mentioned first. Then why in-situ measurements are so valuable but challenging and limited should be mentioned along with pros and cons of other data sources, including, satellite measurements. The purpose of utilizing two separate models should be emphasized. The key factors of OMO field campaign should be included with proper citations as well. Additionally, the goal of this paper and why this paper is unique compared to previous work should be mentioned clearly.

Author:

The introduction is reorganized according to the suggestions of the reviewer.

We used the EMAC model simulations to extend our view on trace gas distributions from the regional scale along flight tracks to a global scale, i.e. horizontal and vertical trace gas distributions, and also to separate different AMA modes. With the second model (FLEXPART) we calculated back trajectories to investigate the emission sources and the transport pathways from the source regions, via the convection into the AMA in the upper troposphere and further westward towards the flight tracks. Thus the back trajectories are mainly used for dynamical processes.

Authors changes in manuscript:

[revised manuscript text omitted]

-Reviewer2:

Abstract of this paper should be a summary of what is shown in this work without including general statements. In the current form, most of the information exists without clearly stating what the goal of this paper is.

Author:

The abstract is revised.

Authors changes in manuscript:

P1 L6-22: The Asian monsoon anticyclone (AMA) is a yearly recurring phenomenon in the northern hemispheric upper troposphere and lower stratosphere. It is part of the South Asian summer monsoon system, and it has a clearly

observable signature due to vertical transport of polluted air masses from the surface to the upper troposphere by the monsoon convection. We performed in situ measurements of carbon monoxide (CO) and methane ($CH_4$) in the region of monsoon outflow and in background air in the upper troposphere (Mediterranean, Arabian Peninsula, Arabian Sea) by optical absorption spectroscopy on board the High Altitude and Long range (HALO) research aircraft during the OMO (Oxidation Mechanism Observations) mission in summer 2015. We identified the transport pathways and the origin of the trace gases with back trajectories, calculated with the Lagrangian particle dispersion model FLEXPART, and we compared the in situ data with simulations of the atmospheric chemistry general circulation model EMAC. $CH_4$ and CO mixing ratios were found to be enhanced within the AMA, the in situ data increased on average by 72.1 ppbv and 20.1 ppbv, respectively, originating in the South Asian region (Indio-Gangetic Plain, North East India, Bangladesh and Bay of Bengal). It appears that $CH_4$ is an ideal monsoon tracer in the upper troposphere due to its extended lifetime and the strong South Asian emissions. Furthermore, we used the measurements and model results to study the dynamics of the AMA over several weeks during the monsoon season, with an emphasis on the southern and western areas in the upper troposphere. We distinguished four AMA modes based on different meteorological conditions. During one occasion we observed that under the influence of dwindling flow the transport barrier between the anticyclone and its surroundings weakened, expelling air masses from the AMA. The trace gases exhibited a distinct fingerprint of the AMA, and we also found that $CH_4$ accumulated over the course of the OMO campaign.

-Reviewer2:

Section 2 (methods) should include general information about OMO field campaign, including its science goal. What other species were measured during the campaign? What were the science questions? Are there any references?

Author:

General information about the OMO mission are added in the manuscript in the method part in section 2.1 including references.

Authors changes in manuscript:

P3-4 L23-7: The Oxidation Mechanism Observation (OMO) aircraft measurement campaign focused on the self-cleaning capacity of the atmosphere in connection with the Indian summer monsoon. The mission took place in July and August 2015 with flight tracks in the upper troposphere (10-15 km) over the Mediterranean, the Arabian Peninsula, and the Indian Ocean (Figure 1). In South Asia the pollution emissions are growing and during the monsoon season they are uplifted into the upper troposphere. The pollution is partly removed by wet deposition or transformation into soluble gases, or they are involved in air chemistry and transported downwind of the sources. For a broad analysis of the efficiency of the self-cleaning mechanism a large variety of chemical compounds, like $CH_4$, CO, OH, $HO_2$, $NO_y$, $SO_2$, $RO_2$, $H_2O_2$, and total peroxides, were measured during the multi-institutional campaign,

involving the Max-Planck-Institute for Chemistry, Mainz, the Research Centre Jülich, the German Aerospace Center, the Research Centre Karlsruhe, and the universities of Bremen, Heidelberg, and Wuppertal. The main objectives were the oxidation processes and free radical chemistry, the efficiency of convective cloud transport and wet deposition, as well as long-distance transport of air pollution and impacts on air quality and climate change. The OMO mission comprised 111 flight hours during 17 flights. HALO was based alternately at Paphos (Cyprus) and on Gan (Maldives) with refueling stops at the airport of Bahrain. Further information about OMO can be found in Lelieveld et al. (2018) and on the webpage http://www.halo.dlr.de/science/missions/omo/omo.html.

-Reviewer 2:

Section 3.5 (AMA mode) should include discussions of bimodal mode of the monsoon anticyclone shown in Zhang et al. (2002) and Nützel et al. (2016). Also, it should be justified why it is necessary to have four modes instead of two. Is bimodal distribution of the anticyclone wrong?

Author:

A short discussion about bimodality of the AMA is now added I section 3.5.

Authors changes in manuscript:

P15 L8-17: Zhang et al. (2002) presented a bimodality of the AMA with a center position of the anticyclone over the Iranian or the Tibetan Plateau. During OMO we found both positions, which in line with the bimodality assumption. In contrast, Nützel et al. (2016) reported different center positions of the AMA in several models, but most of them did not simulate a preferred bimodality. Regarding the eastern anticyclones during the double anticyclones modes, the positions were in-between the Iranian and Tibetan Plateau (first mode) and in the fourth mode over the Tibetan Plateau. Consequently, they do not support a preferred bimodality. In Zhang et al. (2002) and Nützel et al. (2016) the Iranian and the Tibetan mode are further distinguished by parameters, like diabatic heating, rain patterns or areas of convection, which are out of scope in the present study. Here the focus is on the dynamics with respect to the trace gas distributions. The subdivision into four modes represents the dynamics of the AMA over the course of the campaign.

-Reviewer 2:

The abstract includes a few general statements, which makes abstract sound rather like introduction. For instance, L9-11 (However: : :expected) can be removed.

Author:

L9-11 Sentence is removed.

Authors changes in manuscript: P1, L9-11

-Reviewer 2:

P1, L7 – It is connected to -> It is part of the South Asian summer monsoon system

Author: This has been changed.

Authors changes in manuscript: P1, L7: It is part of the South Asian summer monsoon system

-Reviewer 2:

P1, L17-19 – Are those based on the in-situ measurements?

Author:

Yes, these values are representing the in situ data, but also the simulated data show increased mixing ratios with AMA-influence, as mentioned in section 3.4 The AMA during OMO. In situ increase 72.1 ppbv and 20.1 ppbv and EMAC increase 24.0 ppbv and 14.7 ppbv for $CH_4$ and CO, respectively.

Authors changes in manuscript:

P1, L14-15: the in situ data increased on average by 72.1 ppbv and 20.1 ppbv, respectively,

-Reviewer 2:

P1, L21 – areas within the upper troposphere -> areas in the upper troposphere

Author:

This has been changed.

Authors changes in manuscript:

P1, L19: areas in the upper troposphere

-Reviewer 2:

P2, L3 – Park et al. (2008) might be relevant here.

Author:

The reference has been added as it is relevant here.

Authors changes in manuscript:

P2, L2: Stratospheric tracers, like ozone, show generally lower concentrations inside the AMA than outside (Park et al., 2008, Randel and Park, 2006).

-Reviewer 2:

 P2, L4 – within the strong: : :monsoon -> by the strong monsoon convection

Author:

This has been changed.

Authors changes in manuscript:

P2, L3-: to the upper troposphere by the strong monsoon convection.

-Reviewer 2:

P2, L5 – Park et al. (2007) might be relevant here.

Author:

The reference has been added as it is relevant here.

Authors changes in manuscript:

P2,L5: clearly signify the monsoon influence (Park et al., 2007)

-Reviewer 2:

P2, L9 – physical -> physically

Author:

This has been changed.

Authors changes in manuscript:

P2, L20: A more physically motivated criterion

-Reviewer 2:

P2, L17-18 – Full name for CARIBIC and IAGOS-MIZAIC should be provided here as well.

Author:

The full names are added.

Authors changes in manuscript:

P2, L7-10: CARIBIC (Civil Aircraft for the Regular Investigation of the atmosphere Based on an Instrument Container; e.g. Schuck et al., 2012, Rauthe-Schöch et al., 2016) and IAGOS-MOZAIC (IAGOS (In-service Aircraft for a Global Observing System) and MOZAIC (Measurements of OZone by Airbus In-service aircraft); Barret et al., 2016, Dethof et al., 1999)

-Reviewer 2:

P3, L1- It is also important to mention that there is a big uncertainty in source estimates of methane (Bloom et al., 2017 GMD and references there in).

Author:

The information has been added including the reference.

Authors changes in manuscript:

P3L9-11: Further sources are rice cultivation and ruminants, but also swamps and flood areas. For wetlands, the uncertainty in $CH_4$ emissions is still a large concern in atmospheric chemical transport models (Bloom et al., 2017, and references there in).

-Reviewer 2:

P3, L8 – 'variability of the AMA' can be explained more detail here.

Author:

A more detailed explanation is now added.

Authors changes in manuscript:

P4:L20-21: Finally, we investigated the variability of the AMA over several weeks as the anticyclone changes its position, extent, and strength due to the monsoon dynamics.

-Reviewer 2:

P4, L8 (section 2.2) – I assume the trajectory calculations are done backward. Where is the initialization location?

Author:

We calculated back trajectories and the initializations are along the flight tracks. The subtitle for section 2.3 is now "FLEXPART back trajectories".

Authors changes in manuscript:

P5 L8: 2.3 FLEXPART back trajectories

-Reviewer 2:

P5, L11 (section 2.4) – The reason why MODIS cloud top pressure is used is missing. Is this used as convective proxy?

Author:

Yes it is used as a proxy for the location of convection to compare the region with the calculated updraft of the back trajectories. This information is added in the actual section 2.5.

Authors changes in manuscript:

P6 L23-24: Cloud top pressure information is used as a proxy for convection. We compared the location of the convective clouds with the location of the uplift of the back trajectories simulated by FLEXPART. The cloud top pressure data are collected from the MODIS instrument on board of AQUA

-Reviewer 2:

P5, L28 – I would like to know if there are any in-situ measurements of methane and if so how the mixing ratios compare with them even over different regions in different season.

Author:

Yes, there are other in situ profiles. Lelieveld et al. (2002) measured profiles over the Mediterranean in summer 2001 during MINOS. They have observed enhanced $CH_4$ and CO values in the UT, especially during stronger influence from the AMA in the UT with $CH_4$ mixing ratios up to ca. 1890 ppbv. Bergamaschi et al. (2013) presented $CH_4$ profiles over the pacific in dependence of the latitude observed in 2009. The $CH_4$ mixing ratios decrease from the northern hemisphere to the southern hemisphere. The highest values are reported for the lower troposphere in the northern hemisphere (around 1882 ppbv). In the UT $CH_4$ increases towards the tropics to around 1800 ppbv.

Authors changes in manuscript:

P7 L10: is now the position for the authors answer in the manuscript.

-Reviewer 2:

P5, L30 – I have tried to find CO observations from satellite in Randel and Park (2006) but they seem to have used only ozone and water vapor.

Author:

The reference was wrong and the right one is Park et al. (2007).

Authors changes in manuscript:

P7,L12: Park et al. (2007) used CO observations from satellites and wind fields

-Reviewer 2:

P5, L31 – to identified -> to identify

Author:

This has been changed.

Authors changes in manuscript:

P7 L13: to identify monsoon influenced

-Reviewer 2:

P6, L8 -12 – This paragraph is not convincing to me without supporting material or references

Author:

The paragraph is rewritten including supporting material and references.

Authors changes in manuscript:

P7 L22-26: The observed $CH_4$ increase with height can be explained by the global circulation. . In the boundary layer $CH_4$ mixing ratios are influenced by turbulent mixing close to emission sources or by horizontal advection in remote places (Saito et al., 2013). At the surface the air at Gan is influenced by wind from southern directions with low $CH_4$ mixing ratios originating from the southern Indian Ocean. High altitude advection leads to interhemispheric transport (Saito et al., 2013) thus to transfer of higher $CH_4$ mixing ratios from the NH into the SH, which have been convectively uplifted from the boundary layer.

-Reviewer 2:

P6, L18 – This is in consistent -> This is consistent

Author: This has been changed.

Authors changes in manuscript:

P8 L1: This is consistent with the observed upper tropospheric increase of CO and $CH_4$ in the NH background profiles

-Reviewer 2:

L6, L20-22 – Do the mixing ratios of CO in the upper troposphere agree as well?

Author:

Park et al., 2008 reported CO MR in the UT (10-15km) of around 100ppbv in the AMA and 65-90 ppbv outside. We measured in 10-14km around 74.0±15.2 ppbv and outside of 71.2±10.0 ppbv. Park et al., 2008 defined the AMA by a CO threshold opposite to our $CH_4$ approach and in our profiles inside and outside events are included, only separated by their location, which leads to a smaller difference in the CO mixing ratios for background and AMA-influence in the UT.

Authors changes in manuscript:

P8 L1-5-6: CO mixing ratios in the upper troposphere inside the AMA (around 100 ppbv in 10-15 km) in comparison to air outside the AMA (65-90 ppbv in 10-15 km).

Reviewer 2:

P6, L30-31- Does this problem prevented from using the measurement or only degraded the data quality of CO measurements?

Author:

This problem only degraded the data quality of CO measurements.

Authors changes in manuscript:

P8 L14-15:  is now the position for the authors answer in the manuscript.

-Reviewer 2:

P7, Eq. (1) – I think this threshold is somewhat subjective. At least it should be mentioned that this might introduce uncertainty in the analyses and also how sensitive the results are depending on the threshold values.

Author:

The threshold is a simple tool to distinguish between background air masses and air masses influenced by the monsoon. It is based on in situ measurements and it is subjectively chosen, however its application to the in situ data showed a reasonable differentiation. The threshold itself was not applied to the EMAC data along the flight tracks and in the histograms (Figures 16 and 17) as the model underestimated the in situ measurements. The in situ CO and

the EMAC data are distinguished into AMA-influence and background according to the time, when the in situ $CH_4$ was above or below the $CH_4$ threshold. Nevertheless the $CH_4$ threshold is represented in the EMAC horizontal trace gas distributions as a contour line for a better orientation of the AMA position. Consequently the analyses depend on the threshold. A change in the absolute value would increase or decrease the region which we assumed to be influenced by the monsoon.

Authors changes in manuscript:

P8 L22: In situ $CH_4$ mixing ratios

P8 L25-29: Further evaluation depends on the $CH_4$ threshold and thus the results are sensitive to it. Nevertheless also other compounds measured during OMO showed the isolation of the anticyclone in the UT (Lelieveld et al., 2018) which confirms the usefulness of $CH_4$. With a change in the absolute value the region which is supposed to be AMA-influenced will be either larger or smaller, thus the edge of the anticyclone would be differently defined but the whole dynamical process is not significantly changing.

-Reviewer 2:

P7, L28 – Does the difference between Scheeren et al. (2003) and this study agrees with the values in Zimmermann et al. (2018) quantitatively?

Author:

Zimmermann et al. (2018) calculated a $CH_4$ mixing ratio of 1781 ppbv for the upper troposphere between 2000 and 2006. The $CH_4$ values in Scheeren at al. (2003) are 1819±26 ppbv for North America/North Atlantic origin and 1882±21 oobv for South Asia origin. The value in Zimmermann et al. (2018) is a global average over seven years in contrast to the values of Scheeren et al. (2203), which represent only one summer month of northern hemispheric origin, thus not accounting for the lower southern hemispheric $CH_4$ mixing ratios. Zimmermann et al. (2018) increased the $CH_4$ mixing ratio due to additional $CH_4$ emissions starting in 2007 up to 1815 ppbv for 2015. In this study the $CH_4$ mixing ratio is in average 1866.4±43.0 ppbv.

Authors changes in manuscript:

P9 L21: is now the position for the authors answer in the manuscript.

-Reviewer 2:

P8, L13 – cloud top height pressure -> cloud top pressure (also in P10, L23)

Author:

This has been changed.

Authors changes in manuscript:

P10 L6: cloud top pressure

P12 L23: cloud top pressure

-Reviewer 2:

P8, L16-17 – This sentence should be revised for clarity.

Author:

The sentence is rewritten.

Authors changes in manuscript:

P11 L9-10: Matches were generally found over the Bay of Bengal, the Indo-Gangetic Plain, Bangladesh, the north easern region of India, and Myanmar. During the days when the back trajectories passed over central India, convection occurred also in this area, but the cloud top pressure was at a lower altitude than the height of the trajectories.

-Reviewer 2:

P8, L34 – high pressure -> anticyclonic

Author:

This has been changed.

Authors changes in manuscript:

P11 L27: the anticyclonic circulation

-Reviewer 2:

P9, L13 (Figs. 7 & 8) – Here, it looks like the flight path is outside the anticyclone based on the model simulations. The high values from the flight almost should be at the center of the anticyclone. I am not sure how to understand those comparisons.

Author:

We added contour lines for the $CH_4$ threshold (1879.8 ppbv) and the $CH_4$ background (1859.4 ppbv) value in Figure 7 and 8 according to the suggestion of reviewer 1. The flight track crosses the edge of the AMA with higher mixing ratios inside the AMA, which can be seen in the measured and the modeled data. The difference between the in situ and simulated values show that on a regional scale the model is not able to reproduce the reality with respect to the absolute values.

Authors changes in manuscript:

Figure 7 and 8(P31, P32) (P11 L8-9: is now the position for the authors answer in the manuscript.)

-Reviewer 2:

P15, L29 – Instead of 'these transport' describe specific transport processes here

Author:

A detailed description of the transport processes is added.

Authors changes in manuscript:

P1 L19-21: In the present work, we address the transport pathways, including the convective transport from the boundary layer into the UT, the circulation in the AMA, the transport at and across the edges of the AMA, associated with outflow events and further transport in the UT partly in connection with the jet streams.

Revised Submission

Co-Editor Decision: Publish subject to minor revisions (review by editor) (25 Jan 2019) by Jens-Uwe Grooß

Comments to the Author:

Dear Laura Tomsche et al.,

Thank you for the work put into the answers to the reviewers and the detailed revision of the manuscript.

However, I do see still some room of needed improvement.

Co-Editor:

Even though the reviewers did not point to it, I think, you should mention also the study by Vogel et al. (ACP, 2014) who also show CO and CH4 enhancements and the transport pathways due to AMA outflow observed during the ESMVAL campaign, as this is very similar work.

> Answer:

> The study of Vogel et al. (2014) is included in the manuscript, where it fits to the present study.

> Authors changes in manuscript:

> P2 L16-17: and the Transport and Composition in the Upper Troposphere and Lowermost Stratosphere (TACTS) campaign (Vogel et al., 2014).

> P10 L19-24: Furthermore, similar increases in CO and CH4 mixing ratios, caused by an outflow event of the AMA, were found during a flight over Northern Europe during TACTS (Vogel et al., 2014). They reported enhancements of approximately 25 ppbv for CO (background: 15-25 ppbv, outflow air: 40-50 ppbv) and 65 ppbv for CH4 (background: 1700-1750 ppbv, outflow air: 1770-1810 ppbv). The absolute values for CO and CH4 are lower in comparison to the present study. The transport time was about five weeks so the air mass of the outflow event could be mixed with background air. The background itself has a different characteristic as the flight during TACTS took place in the lower stratosphere and flight 19 took place in the upper troposphere.

> P19 L21-26: Beside outflow events happening at the western edge of the AMA, as documented here, it is also possible at the eastern edge of the AMA as described in Vogel et al. (2014). During TACTS they probed an air mass with enhanced CO and CH4 mixing ratios over Northern Europe in the lower stratosphere. They used back trajectories to analyse the transport pathways of this air mass. The air mass was injected at the south eastern edge of the AMA, streamed clockwise around the AMA, and flowed out of the AMA at the north eastern part by eastward eddy shedding. Afterwards the air was transported eastward with the subtropical jet and reached after ca. five weeks the flight track over Northern Europe.

Co-Editor:

Reviewer #2 states that "most of the information exists without clearly stating what the goal of this paper is"

You state that the goal is "to understand the transport pathways from the source regions into the upper troposphere..."

As it stands, the manuscript contains still a lot of figures and it is not clear in every case, whether all of these are needed to underline the goal. Therefore I would ask you to review in what respect the individual figures support your goal.

For example, figs. 4 and 5 show altitude and transport time at the locations of the back trajectories from the complete flight path. That is for perturbed and unperturbed air. You refer then to the (yet not explained) figures 7 and 8 and the reader has to deduce which of the trajectories are perturbed. Likely it would be better in the sense of that goal to mark these trajectories, e.g. by different symbols for below and above your threshold.

> Answer:
>
> The back trajectories for AMA-influence and background are distinguished by different symbols with an explanation in the caption in Figure 4 and 5, and also in the trajectory plots in the supplement.
>
> After reviewing the manuscript with respect to the goal of the paper, Figures 11 and 12 (vertical transect of CO) are removed.
>
> Authors changes in manuscript:
>
> P28 Figure 4 Triangles are back trajectories for CH4 mixing ratios above the CH4 threshold and circles for below the CH4 threshold.
>
> P29 Figure 5 Triangles are back trajectories for CH4 mixing ratios above the CH4 threshold and circles for below the CH4 threshold.
>
> P13 L6: Comment to the removal of Figures 11 and 12 removed

Co-Editor:

The explanation of figure 6 (cloud top pressure) should better include a (rough) pressure value that you would assign to "regions with strong convection"

> Answer:
>
> > Added a value in manuscript and figure capture.
>
> Authors changes in manuscript:
>
> > P11 L8: to identify regions with strong convection (here with cloud top pressure above 200 hPa)
> >
> > P31 Figure 6: Satellite-derived cloud top pressure 10 days prior (August 03, 2015) to flight 19. Pressure below 250 hPa represents strong convection.

Co-Editor:

figure 13/ particle density in PBL: These are for those trajectories starting an the flight path at locations that have CH4 mixing ratios above the threshold. That should also be mentioned in the figure caption.

Answer:

Explanation added in figure caption.

Authors changes in manuscript:

P45: Figure 13: Last boundary layer contact of parcels from trajectories, which start along the flight track at locations with CH4 mixing ratios above the threshold, before they were transported to the track of flight 19 (10 days prior to flight).

Co-Editor:

figs 24/25 It is somewhat confusing that you use blue/green colors for CO and CH4 opposite for model and measurements. I think it is more suggestive to use similar colors for the same molecule (e.g. light blue and dark blue)

Answer:

Model and in situ data are color coded with similar color for the same molecule in Figure 24 and 25, and consequently also in Figure 3 and in the supplement.

Authors changes in manuscript:

P27 Figure 3

P67 Figure 24 and Figure 25

Co-Editor:

p17 l2ff. fig 26 shows a few example trajectories from points of enhanced CH4. As there are likely more points of enhanced CH4 along the flight path, how do you select the shown trajectories? How do you show it is outflow from the AMA? It is likely defined by the outflowing airmass.

Answer:

The selection is made by the outflowing air masses with help of the enhanced CH4 in situ measurements and the location of outflow according to the model data (Figure 22 and 23). Position of the outflow is also added in the Figure captions.

Authors changes in manuscript:

P18-19 L31-1: In Figure 26 the 10-day back trajectories for the enhanced $CH_4$ values, associated with the locations of the outflow event mentioned before (Figure 22 and 23), are shown for the flights 12/13 and 17/18.

P64: Figure 22 The enhanced CH4 values over Oman represent the outflow.

P66: Figure 23 The enhanced CH4 values over the Read Sea represent the outflow.

Co-Editor:

p8/l21/equation 1: please don't mix words and the equation. Preferably use superscripts or subscripts like CH_4^{threshold} or CH_4^{average}

Answer:

This has been changed.

Authors changes in manuscript:

P9 L17: $CH_{4\ threshold} = CH_{4\ average} + 2\ \sigma = 1859.4\ \text{ppbv} + 2 * 10.2\ \text{ppbv} = 1879.8\ \text{ppbv}$

Co-Editor:

p8/l27 "which confirms the usefulness of CH4" is rather sloppy language. I would rather say "which confirms the possibility of this threshold to divide the air mass origin between inside and outside the AMA" or so.

Answer:

This has been changed.

Authors changes in manuscript:

P9 L24-25: which confirms the possibility of this threshold to divide the air mass origin between inside and outside the AMA.

**Upper tropospheric CH$_4$ and CO affected by the South Asian summer monsoon during OMO**

Laura Tomsche[1], Andrea Pozzer[1], Narendra Ojha[1], Uwe Parchatka[1], Jos Lelieveld[1], Horst Fischer[1]

[1]Department of Atmospheric Chemistry, Max-Planck-Institute for Chemistry, Mainz, 55128, Germany

5  *Correspondence to*: Laura Tomsche (laura.tomsche@mpic.de)

**Abstract.** The Asian monsoon anticyclone (AMA) is a yearly recurring phenomenon in the northern hemispheric upper troposphere and lower stratosphere. It is part of the South Asian summer monsoon system, and it has a clearly observable signature due to vertical transport of polluted air masses from the surface to the upper troposphere by the monsoon convection.  We performed in situ measurements of

10  carbon monoxide (CO) and methane (CH$_4$) in the region of monsoon outflow and in background air in the upper troposphere (Mediterranean, Arabian Peninsula, Arabian Sea) by optical absorption spectroscopy on board the  High Altitude and Long range (HALO) research aircraft during the OMO (Oxidation Mechanism Observations) mission in summer 2015. We

15  identified the transport pathways and the origin of the trace gases with back trajectories, calculated with the Lagrangian particle dispersion model FLEXPART, and we compared the in situ data with simulations of the atmospheric chemistry general circulation model EMAC. CH$_4$ and CO mixing ratios were found to be enhanced within the AMA, the in situ data increased on average by 72.1 ppbv and 20.1 ppbv, respectively, originating in the South Asian region (Indio-Gangetic Plain, North East India, Bangladesh and Bay of Bengal). It appears that CH$_4$ is an ideal monsoon tracer in the upper troposphere due to its

20  extended lifetime and the strong South Asian emissions. Furthermore, we used the measurements and model results to study the dynamics of the AMA over several weeks during the monsoon season, with an emphasis on the southern and western areas in the upper troposphere. We distinguished four AMA modes based on different meteorological conditions. During one occasion we observed that under the influence of dwindling flow the transport barrier between the anticyclone and its surroundings weakened, expelling air masses from the AMA. The trace gases exhibited a distinct fingerprint of the AMA,

25  and we also found that CH$_4$ accumulated over the course of the OMO campaign.

**1 Introduction**

The Asian monsoon anticyclone (AMA) is an annual, large-scale weather  phenomenon in the upper troposphere and lower stratosphere during the boreal summer. It is enclosed by the westerly subtropical jet in the north and the

easterly jet in the south and extends over southern Asia and the Middle East up to the Mediterranean. It is formed by diabatic heating in the South Asian monsoon region (Gill, 1980, Hoskins and Rodwell, 1995). The anticyclone is a strong and nearly closed circulation system, which is variable in strength and location (Hsu and Plumb, 2000, Popovic and Plumb, 2001, Garny and Randel, 2013, Ploeger et al., 2015). The strong winds at its edges act as transport barrier for chemical constituents in the upper troposphere. Stratospheric tracers, like ozone, show generally lower concentrations inside the AMA than outside (Park et al., 2008, Randel and Park, 2006). Tropospheric tracers, like CO and $CH_4$, are uplifted to the upper troposphere by within the strong monsoon convection of the monsoon. These chemical constituents can be trapped in the anticyclone, changeing the atmospheric chemistry in the upper troposphere and lower stratosphere and clearly signify the monsoon influence ((Park et al., 2007)Park et al., 2007). The signature of the anticyclone has been identified from different measurement platforms, like satellites and aircrafts. Airborne measurements are rare and limited in time and space but they resolve small scales. There areFor example, the in-service airborne projects CARIBIC (Civil Aircraft for the Regular Investigation of the atmosphere Based on an Instrument Container; e.g. Schuck et al., 2012, Rauthe-Schöch et al., 2016) and IAGOS-MOZAIC (IAGOS (In-service Aircraft for a Global Observing System) and MOZAIC (Measurements of OZone by Airbus In-service aircraft); Barret et al., 2016, Dethof et al., 1999), which reported trace gas measurements in the Asian monsoon region. In addition aircraft campaigns investigated the Asian monsoon, like during the aircraft campaign MINOS (Lelieveld et al., 2002, Scheeren et al., 2003),andor the Earth System Model Validation (ESMVal) campaign (Gottschaldt et al., 2017), and the Transport and Composition in the Upper Troposphere and Lowermost Stratosphere (TACTS) campaign (Vogel et al., 2014). Airborne measurements are rare and represent only on temporary and spacial restricted scales but resolve small scales. In contrast, satellite data cover a larger spaetial area and can be used for long term measurements, nevertheless they are limited to theirrepresent mostly one overpassing trackflight time and they have a coarser resolution. The obscured view fromby clouds induring the South Asian monsoon restricts additionally restricts the satellite dataview (e. g. Ojha et al., 2016), which requires long-term averaging in time and should be complemented by in situ measurements. Moreover, sSatellite data for different trace gases, like $H_2O$ (Park et al., 2004, Randel and Park, 2006), CO (Li et al., 2005, Park et al, 2008) and $CH_4$ (Park et al., 2004), show the vertical and horizontal extension of the AMA and which are generally in agreement with model simulations (e.g. Pan et al., 2016, Nützel et al., 2016, Bergman et al., 2013). Global models simulate the AMA as a dynamic pattern on a global scale, but a detailed resolution (e.g. the distribution and the absolute mixing ratios of trace gases) is more dependent on the grid size of the model itself, the parameterization of the convection, and the emission inventory. To improve model outputs and satellite data retrievals, airborne measurements are necessary.,

A more physically motivated criterion to distinguish between the AMA and its surrounding in the upper troposphere is the potential vorticity (PV) (e.g. Ploeger et al., 2015, Garny and Randel, 2013). In the anticyclone PV values on isentropic surfaces are lower than outside. Therefore, a maximum in the PV gradient can be used to identify the horizontal transport barrier associated with the AMA. However, applying the PV criterion is not straightforward since PV values in the AMA increase

**Kommentiert [TL7]:** The reference has been added as it is relevant here.

**Kommentiert [TL8]:** This has been changed.

**Kommentiert [TL9]:** The reference has been added as it is relevant here.

**Kommentiert [TL10]:** The full names are added.

**Kommentiert [TL11]:** The study of Vogel et al. (2014) is included in the manuscript, where it fits to the present study.

**Kommentiert [TL12]:** This has been changed.

[revised manuscript text omitted]

Kommentiert [TL14]: The information has been added including the reference.

Kommentiert [TL15]: A more detailed explanation is now added.

Kommentiert [TL16]: We used the EMAC model simulations to extend our view on trace gas distributions from the regional scale along flight tracks to a global scale, i.e. horizontal and vertical trace gas distributions, and also to separate different AMA modes. With the second model (FLEXPART) we calculated back trajectories to investigate the emission sources and the transport pathways from the source regions, via the convection into the AMA in the upper troposphere and further westward towards the flight tracks. Thus the back trajectories are mainly used for dynamical processes.

Kommentiert [TL17]: General information about the OMO mission are added in the manuscript in the method part in section 2.1 including references.

**2.2 Trace gas measurements**

[revised manuscript text omitted]

**Kommentiert [TL20]:** Yes it is used as a proxy for the location of convection to compare the region with the calculated updraft of the back trajectories. This information is added in the actual section 2.5.

**Kommentiert [TL21]:** In the classification of the profiles for Northern hemisphere and AMA-influenced profiles the geographical location but also the meteorological situations are accounted. The profiles over Egypt were only sampled during the second double anticyclone mode with the westerly extending over Egypt. Profiles over Paphos spread over a longer period with representing the background atmosphere and partly also AMA-influenced air masses, which leads within the standard deviation to an average of a background profile. The figures 18-21 are only examples for the specific mode and they are in a height of 204hPa. Not necessarily representative for the atmosphere below or representative for different meteorological situations.

over Paphos, Etna, and Oberpaffenhofen are used to derive a northern hemisphere (NH) background, while profiles over Egypt and Bahrain are used to derive altitude dependent information under monsoon influence (AMA profiles), and profiles over Gan are used to derive a southern hemisphereeie (SH) background (Figure 2). Average profiles were calculated in 500 -m bins, starting above 4 -km to avoid boundary layer effects. Inspection of the $CH_4$ AMA profile indicates a significant enhancement in the upper troposphere between 9 and 12.5 km corresponding to pressure levels between 300 and 170 hPa. Randel and Park et al. (20076) used CO observations from satellites and wind fields to identifyied monsoon influenced air masses inside the AMA at a similar pressure range (200-100 hPa). The average $CH_4$ mixing ratio of the AMA profile between 9 km and 12.5 -km is (1919.0±17.2) ppbv, while the average $CH_4$ mixing ratio for the  northern hemisphere (NH) background is 1863.4±14.0 ppbv, comparable to $CH_4$ mixing ratios below 9 km measured for the AMA profile (1876.5±8.7) ppbv. The average $CH_4$ mixing ratio for the SH background is 1778.3±19.5 ppbv, significantly lower than either the NH background or the AMA profiles. While the NH background shows only a small increase of $CH_4$ above 11 km, the SH background profile steadily increases with height. Gan is located at the equator and thus influenced by the southern hemisphere during boreal summer when the ITCZ is shifted to the north (Waliser and Gautier, 1993). Since most of the methane sources are in the northern hemisphere north of the ITCZ, the profile over Gan thus to some extend represents the southern hemisphere (SH) background. The observed $CH_4$ increase with height can be explained by the global circulation. In the boundary layer $CH_4$ mixing ratios are influenced by turbulent mixing close to emission sources or by horizontal advection in remote places (Saito et al., 2013). At the surface the air at Gan is influenced by wind from southern directions with low $CH_4$ mixing ratios originating from the southern Indian Ocean. High altitude At high altitudes the advection leads to interhemispheric transport (Saito et al., 2013) thus to a transporttransfer of higher $CH_4$ mixing ratios from the NH into the SH, which have been convectivleyconvectively uplifted from the boundary layer. upper branch of the Hadley circulation leads to air transport from the ITCZ to the SH. These air masses are influenced by local pollution from the NH air mixed into the SH background in the ITCZ. The observed difference in $CH_4$ background between the NH and the SH is 85.1 ppbv, which agrees with an interhemispheric gradient of 86-90 ppbv for the period 2007 to 2010 given in Bergamaschi et al. (2013).

The measured mean CO profiles for AMA (74.2±10.9 ppbv), NH background (68.8±7.3 ppbv) and SH background (63.2±4.3 ppbv) are rather similar and agree within the standard deviations. Nevertheless, in the upper troposphere the AMA profile indicates a slight increase of CO mixing ratios relative to the background. Enhanced CO and $CH_4$ mixing ratios in the upper troposphere over the eastern Mediterranean in summer 2001, associated with air masses influenced by the monsoon, were also observed during the MINOS aircraft campaign (Lelieveld et al, 2002, Scheeren et al., 2003). This is in consistent with the observed upper tropospheric increase of CO and $CH_4$ in the NH background profiles, which are found during ascends and descends over Paphos but not over Oberpfaffenhofen. In general, our observation of enhanced CO mixing ratios under monsoon influence are consistent with Park et al. (2008), who showed that satellite-based averaged CO profiles exhibit

**Kommentiert [TL22]:** Yes, there are other in situ profiles. Lelieveld et al. (2002) measured profiles over the Mediterranean in summer 2001 during MINOS. They have observed enhaved CH4 and CO values in the UT, especially during stronger influence from the AMA in the UT with CH4 mixing ratios up to ca. 1890 ppbv. Bergamaschi et al. (2013) presented CH4 profiles over the pacific in dependence of the latitude observed in 2009. The CH4 mixing ratios decrease from the norhtern hemisphere to the southern hemisphere. The highest values are reported for the lower troposphere in the northern hemisphere (around 1882 ppbv). In the UT CH4 increases towards the tropics to around 1800 ppbv.

**Kommentiert [TL23]:** The reference was wrong and the right one is Park et al. (2007).

**Kommentiert [TL24]:** This has been changed.

**Kommentiert [TL25]:** The paragraph is rewritten including supporting material and references.

**Kommentiert [TL26]:** This has been changed.

increased CO mixing ratios in the upper troposphere inside the AMA (around 100 ppbv in 10-15 km) in comparison to air outside the AMA (65-90 ppbv in 10-15 km).

To differentiate between air masses in and outside the AMA various approaches have been used in the literature. Often potential vorticity (PV) is used for this purpose (e.g. Randel and Park, 2006, Garny and Randel, 2013 or Ploeger et al., 2015). Ploeger et al. (2015) calculated PV from reanalysis data to determine a transport barrier isolating the AMA. In the restricted area of interest low PV values are found inside the anticyclone while higher PV values represent the background. A more direct approach is the use of a CO threshold (Park et al., 2008). Based on satellite data, Park et al. (2008) found that CO mixing ratios < 60 ppbv represent background air while CO mixing ratios > 60 ppbv represent air inside the AMA at 16.5 km. In our study the monsoon influence in the upper troposphere is most obvious in the $CH_4$ profile, while CO is less suitable due to its larger atmospheric variability associated with its shorter lifetime (Junge, 1974) and the instrumental problems experienced for CO during the second half of the campaign. Therefore, a methane threshold was derived to signify monsoon influenced air masses from the NH background profile. To avoid boundary layer effects and the above mentioned slight increase in the NH background profile above 11 km due to a small contribution of monsoon influenced air above the eastern Mediterranean, only data between 4 km and 10 km were used, yielding an average $CH_4$ mixing ratio for the background of 1859.4±10.2 ppbv, which is slightly lower than the above mentioned mixing ratio covering the whole altitude range. The $CH_4$ threshold is then defined as this average plus twice the standard deviation:

$$CH_{4\,threshold} = CH_{4\,average} + 2\,\sigma = 1859.4\ \text{ppbv} + 2 * 10.2\ \text{ppbv} = 1879.8\ \text{ppbv},$$
(1)

In situ $CH_4$ mixing ratios that exceed this threshold are assumed to be influenced by the South Asian monsoon and are therefore being representative of the AMA, in the following denoted as being AMA-influenced. While in the NH background profile, $CH_4$methane mixing ratios in the upper troposphere are generally smaller than this threshold $CH_4$ mixing ratios in the AMA profile significantly exceed this threshold above 9 km (Figure 2). The fFurther evaluation depends on the $CH_4$ threshold and thus the results are sensitive to it. Nevertheless, also other compounds measured during OMO showed the isolation of the anticyclone in the upper troposphereUT (Lelieveld et al., 2018) which confirmesconfirms the possibility of this threshold to divide the air mass origin between inside and outside the AMAusefulness of $CH_4$. With a change in the absolute value the region which is supposed to be AMA-influenced will be either larger or smaller, thus the edge of the anticyclone would be differently defined but the whole dynamical process is not significantly changing.

**3.2 Case study flight 19**

To illustrate the connection between enhanced $CH_4$ mixing ratios, monsoon convection, and South Asian pollution sources at the surface we performed a case study on flight 19 data (August 13, 2015). The flight took place over the Arabian Peninsula.

**Kommentiert [TL27]:** Park et al., 2008 reported CO MR in the UT (10-15km) of around 100ppbv in the AMA and 65-90ppbv outside. We measured in 10-14km around 74.0±15.2 ppbv and outside of 71.2±10.0 ppbv. Park et al., 2008 defined the AMA by a CO threshold opposite to our $CH_4$ approach and in our profiles inside and outside events are included, only separated by their location, which leads to a smaller difference in the CO mixing ratios for background and AMA-influence in the UT.

**Kommentiert [TL28]:** This problem only degraded the data quality of CO measurements.

**Kommentiert [TL29]:** This has been changed.

**Kommentiert [TL30]:** This has been changed.

**Kommentiert [TL31]:** The threshold is a simple tool to distinguish between background air masses and air masses influenced by the monsoon. It is based on in situ measurements and it is subjectively chosen, however its application to the in situ data showed a reasonable differentiation. The threshold itself was not be applied to the EMAC data along the flight tracks and in the histograms (Figures 16 and 17) as the model underestimated the in situ measurements. The in situ CO and the EMAC data are distinguished into AMA-influence and background according to the time, when the in situ $CH_4$ was above or below the $CH_4$ threshold. Nevertheless the $CH_4$ threshold is represented in the EMAC horizontal trace gas distributions as a contour line for a better orientation of the AMA position. Consequently the analyses depend on the threshold. Nevertheless a change in the absolute value would increase or decrease the region which we assumed to be influenced by the monsoon.

After take-off from Paphos HALO headed towards Oman before returning back to Paphos. Enhanced mixing ratios for CO and $CH_4$ were measured between 10-11UTC (Figure 3). Over Oman at a pressure level of 175 hPa mixing ratios for CO and $CH_4$ increased from background levels of (74.3±10.6) ppbv and (1846.7±16.1) ppbv to (99.5±14.3) ppbv and (1905.2±13.9) ppbv, respectively. According to the classification defined in the previous chapter $CH_4$ mixing ratios reached
5   values well above the threshold indicating that air masses influenced by the monsoon were probed. Elevated mixing ratios were still observed after a flight level change (200 hPa). Accordingly, both flight levels were within the altitude range of the AMA confinement of 200-100 hPa reported by Randel and Park et al. (2006). Within the AMA the average increase relative to background for CO is around 25 ppbv and for $CH_4$ around 58 ppbv. The increase in $CH_4$ is rather sharp, indicating a rather well defined edge of the AMA, as has been reported in previous studies (e.g. Park et al., 2008).

10  For the MINOS campaign over the eastern Mediterranean, Scheeren et al. (2003) distinguished between air masses that originated over South Asia and those over North America/North Atlantic, corresponding to our classification aof AMA air masses and NH background, respectively. Note that Scheeren's South Asia air mass would be incorporated in our background, due to its location over the eastern Mediterranean. Scheeren et al. (2003) reported in situ trace gas measurements for the 6-13 km altitude range. Mean CO mixing ratios were (74±12) ppbv for North American/North Atlantic origin and (102±4) ppbv
15  for air masses with a South Asian origin, resulting in a difference of 28 ppbv. The relative difference in $CH_4$ observed by Scheeren is 63 ppbv (North America/North Atlantic: (1819±26) ppbv, South Asia: (1882±21) ppbv). The enhancements observed during MINOS are similar to those observed during OMO flight 19, although absolute mixing ratios in particular for $CH_4$ are higher, since global $CH_4$ concentrations have been increasing since summer 2001 (Zimmermann et al., 2018). Furthermore, similar increases in CO and $CH_4$ mixing ratios, caused by an outflow event of the AMA, were documented over
20  Northern Europe during a TACTS flight by Vogel et al. (2014). They reported enhancements of approximately 25 ppbv for CO (background: 15-25 ppbv, outflow air: 40-50 ppbv) and 65 ppbv for $CH_4$ (background: 1700-1750 ppbv, outflow air: 1770-1810 ppbv). The absolute values for CO and $CH_4$ are lower in comparison to the present study. The transport time was about five weeks so the air mass of the outflow event could be mixed with background air. The background itself has a different characteristic as the flight during TACTS took place in the lower stratosphere and flight 19 took place in the upper troposphere.
25  Using FLEXPART, 10-day centroid back trajectories along the flight track were calculated. An analysis indicates that in general enhanced $CH_4$ mixing ratios are associated with an air mass origin inside the AMA, while lower $CH_4$ mixing ratios are associated with background air (Figure 4). In particular the back trajectories starting at release points with the highest $CH_4$ mixing ratios measured along the flight track (Figure 7, 8) have been confined in the AMA for several days with their origin over Northern India and Bangladesh. Between five to ten days before observations the back-trajectories are found in the
30  boundary layer or the lower troposphere, before they are uplifted into the upper troposphere by deep convection (> 200 hPa) (Figure 5). This finding is in good agreement with Bergman et al. (2013), who calculated trajectory transit times of 2-22 days from the surface to the 200 hPa level in the region of the Tibetan Plateau and India/SE Asia. After the convective injection into

**Kommentiert [TL32]:** Zimmermann et al. (2018) calculated a $CH_4$ mixing ratio of 1781 ppbv for the upper troposphere between 2000 and 2006. The $CH_4$ values in Scheeren at al. (2003) are 1819±26 ppbv for North America/North Atlantic origin and 1882±21 oobv for South Asia origin. The value in Zimmermann et al. (2018) is a global average over seven years in contrast to the values of Scheeren et al. (2003), which represent only one summer month of northern hemispheric origin, thus not accounting for the lower southern hemispheric $CH_4$ mixing ratios. Zimmermann et al. (2018) increased the $CH_4$ mixing ratio due to additional $CH_4$ emissions starting in 2007 up to 1815 ppbv for 2015. In this study the $CH_4$ mixing ratio is in average 1866.4±43.0 ppbv.

**Kommentiert [TL33]:** The study of Vogel et al. (2014) is included in the manuscript, where it fits to the present study.

[revised manuscript text omitted]

Since the 200 hPa level is representative for the dominant flight altitude, the in situ observations along the flight track can also be compared to the simulated 2D fields. Further, we assume that the 200 hPa level is where most of the convective outflow takes place, and therefore pollution levels are expected to be highest (Park et al., 2009). Figure 7 and 8 show that OMO flight 19 only scratches the western edge of the AMA. Measured $CH_4$ mixing ratios are higher inside the AMA than the simulated ones. This is in line with the above mentioned $CH_4$ underestimation by the model. In the model simulations the anticyclone shows a more distinct signal in $CH_4$ compared to CO, since the edge in $CH_4$ is well-defined, with mixing ratios dropping off significantly outside the anticyclone. In contrast, the CO pattern is more diffuse. The simulated CO pattern, especially the enhanced values over Oman, fits well to the observed CO mixing ratios along the flight track, which can be partly due to the effect that the model overestimates CO while $CH_4$ values are generally underestimated by the model. The EMAC model underestimates $CH_4$ and CO in the upper troposphere. As shown by Krol et al. (2018), EMAC seems to have a weaker transport of surface tracers than other models. There are two potential reasons for that, but it is difficult to distinguish them. First, a too slow vertical velocity, thus the convective updraft is too ineffective, or second, the numerical diffusion implied by the coarse resolution restricts the updraft too strong. Nevertheless we would like to notice that for the comparison of CO with the model, the results are in line with other literature studies at such resolution (e.g. Baret et al., 2016).

Additional vertical transects along 23.7°N latitude and 56.2°E longitude on August 13, 2015 complete the picture of the AMA with respect to its extension. In a vertical $CH_4$ transect along 23.7°N (Figure 9) it is obvious that the flight touches only the western edge of the anticyclone in the upper troposphere and that the majority of the flight took place outside the anticyclone. According to the model simulation, convective uplift of $CH_4$ takes place between 75°E and 95°E, which corresponds to India and the Bay of Bengal. Moreover, the upward transport of polluted air masses is only simulated in a rather restricted area, analogous to a chimney, as reported by Bergman et al. (2013). This area was also the preferred location for convection 10 days prior to the flight as reported above. Rauthe-Schöch et al. (2016) reported a similar longitudinal position for convection between 80°E and 100°E in summer 2008 for CARIBIC flights over India. In the vertical transect along 56.2°E (Figure 10)

**Kommentiert [TL38]:** We added contour lines for the $CH_4$ threshold (1879.8 ppbv) and the $CH_4$ background (1859.4 ppbv) value in Figure 7 and 8 according to the suggestion of reviewer 1. The flight track crosses the edge of the AMA with higher mixing ratios inside the AMA, which can be seen in the measured and the modeled data. The difference between the in situ and simulated values show that on a regional scale the model is not able to reproduce the reality with respect to the absolute values.

**Kommentiert [TL39]:** Indeed, the referee is correct in mentioning a possible too low transport of methane and carbon monoxide as a reason for underestimation in the upper troposphere. As shown by Krol et al. (2018), EMAC seems to have a weaker transport of surface tracers than other models. Both reasons suggested by the referee are possible, and it is difficult (if not impossible) to really distinguish the real reason for the underestimation of the transport. Nevertheless we would like to notice that for the comparison of CO with the model, the results are in line with other literature studies at such resolution (e.g. Baret et al., 2016). Horizontal wind components are added to the cross sections in figures 9-12, in detail: eastward wind component in cross sections along a longitude and northward wind component in cross sections along a latitude.

CH$_4$ mixing ratios show an increase at the surface from the equator towards higher northern latitudes. In the upper troposphere the AMA is located approximately between 15°N and 30°N, which fits well with the location of the AMA in summer 2008 identified by enhanced CH$_4$ mixing ratios observed on a CARIBIC flight between 10-40°N (Baker et al., (2012). In vertical transects at longitudes between 75-95°E (not shown) the convection can be determined to occur between 20°N to 35°N, which

5 reflects the area Indo-Gangetic Plain, Tibetan Plateau, Bangladesh, and the north eastern part of India. In the vertical transects for CO (not shownFigure 11, 12) the same patterns are found, although less pronounced compared to CH$_4$. In athe CO latitudinal transect along 23.7°N (Figure 11) the enhanced mixing ratios range from around 90 ppbv inside the anticyclone to over 400 ppbv at the surface, while CH$_4$ mixing ratios scale from around 1850 ppbv inside the anticyclone to surface values above 2250 ppbv (Figure 9).

**Kommentiert [TL40]:** After reviewing the manuscript with respect to the goal of the paper, Figures 11 and 12 (vertical transect of CO) are removed.

[revised manuscript text omitted]

**Kommentiert [TL43]:** According to the suggestion of the reviewer the CO and CH4 distributions along the flight track as a time line are added in Figures 24 and 25. The trace gas mixing ratios (observed and simulated) show clear enhancement due to the outflow event for both flights (flight 12/13 and flight 17/18).

11.9 km and flight 17/18: 12.4 km). Note that the standard deviation for CO during flight 12/13 is larger than for flight 17/18 due to technical problems with the CO. Comparing the EMAC simulations with the in situ data along the flight tracks (Figure 22 and 23), the trends for the outflow agree. The EMAC average mixing ratios for CO and $CH_4$ are 112.2±1.2 ppbv and 1891.7±1.2 ppbv, respectively, for flight 12/13 and 90.8±3.1 ppbv and 1864.6±5.9 ppbv for flight 17/18. Thus also the values

5 agree within their standard deviation beside $CH_4$ in flight 17/18, where the outflow is underestimated by the model.

To check if the expelled air masses probed in the two flights were connected in a Lagrangian sense, we use centroid back trajectories. In Figure 244 the 10-day back trajectories for the enhanced $CH_4$ values, associated with the locations of the outflow event mentioned before (Figure 20 and 21), are shown for the flights 12/13 and 17/18. The trajectories have their origin in the lower troposphere (below ~550 hPa) over the Arabian Sea and the Indian subcontinent. In the area of Bangladesh, the

10 trajectories are convectively uplifted to the upper troposphere. From there they follow the tropical jet towards the west, towards Oman and the Red Sea. The trajectories fit well to the simulated movement of the air mass with the enhanced $CH_4$ values. Besides the similar geographical and altitude position of the trajectories of both flights, the positions also agree in time, which means that the back trajectories of flight 17/18, released on August 10, needed four days between their release points and the crossing with the release points of flight 12/13. This travel duration is exactly the time between the two flights. Therefore, in

15 Figure 255 the trajectories are colour-coded with time, starting from August 10 (day zero) counting backward in time. The trajectories are uplifted in the same period and exceed the 300 hPa level around -9 days. Subsequently, the trajectories of both flights travel together westwards in the same latitudinal band. On August 6 (-4 days), when the trajectories of flight 12/13 are released, they reach Oman and accordingly the release points of flight 12/13. Thus, the trajectories for both flights coincide in time and space. Therefore, they complete the picture of the outflow observed in the simulations and confirm the in situ

20 measurement analysis.

Beside outflow events at the western edge of the AMA, as documented here, they are also possible at the eastern edge of the AMA as described in Vogel et al. (2014). During TACTS they probed an air mass with enhanced CO and $CH_4$ mixing ratios over Northern Europe in the lower stratosphere. They used back trajectories to analyse the transport pathways of the expelled air mass. The air mass was injected at the south eastern edge of the AMA, streamed clockwise around the AMA, and flowed

25 out of the AMA at the north eastern part by eastward eddy shedding. Afterwards the air was transported eastward with the subtropical jet. After ca. five weeks the expelled air mass reached the flight track over Northern Europe.

**4 Summary and Conclusion**

The AMA is a  circulation system in the upper troposphere and lower stratosphere, which appears over Asia during

30 boreal summer. The anticyclone is coupled to deep convection (Hoskins and Rodwell, 1995), which pumps up polluted air masses. The relatively strong anticyclonic circulation traps the pollutants inside the AMA and constitutes a clear chemical

Kommentiert [TL44]: The outflow regions are marked in grey in the Figures. Additionally, we add in the manuscript the average CO and CH4 mixing ratios calculated from EMAC for the outflow periods for both flights (flight 12/13: CO=112.2±1.2 ppbv and CH4=1891.7±1.2 ppbv and flight 17/18: CO=90.8±3.1 ppbv and CH4=1864.6±5.9 ppbv) for a better comparison with the measured data in the outflow.

Kommentiert [TL45]: The selection is made by the outflowing air masses with help of the enhanced CH4 in situ measurements and the location of outflow according to the model data (Figure 22 and 23). Position of the outflow is also added in the Figure captions.

Kommentiert [TL46]: The study of Vogel et al. (2014) is included in the manuscript, where it fits to the present study.

[revised manuscript text omitted]

**Kommentiert [TL48]:** Model and in situ data are color coded with similar color for the same molecule in Figure 24 and 25, and consequently also in Figure 3 and in the supplement.

[Figure]

**Figure 4: Centroid trajectories for flight 19 (August 13, 2015) with colour coded altitude.** Triangles are back trajectories for CH₄ mixing ratios above the CH₄ threshold and circles for below the CH₄ threshold.

**Kommentiert [TL49]:** The back trajectories for AMA-influence and background are distinguished by different symbols with an explanation in the caption in Figure 4 and 5, and also in the trajectory plots in the supplement.

[Figure]

**Figure 5: Centroid trajectories for flight 19 (August 13, 2015) with colour coded transport time. Triangles are back trajectories for CH$_4$ mixing ratios above the CH$_4$ threshold and circles for below the CH$_4$ threshold.**

**Kommentiert [TL50]:** The back trajectories for AMA-influence and background are distinguished by different symbols with an explanation in the caption in Figure 4 and 5, and also in the trajectory plots in the supplement.

[Figure]

Cloud_Top_Pressure_Day_Mean 20150803

[Figure]

**Figure 6: Satellite-derived cloud top pressure 10 days prior (August 03, 2015) to flight 19.** Pressure below 250 hPa represents strong convection.

**Kommentiert [TL51]:** Added a value in manuscript and figure capture.

[Figure]

[Figure]

**Figure 7:** EMAC modelled CH$_4$ and wind field; daily means at 204 hPa, and in situ CH$_4$ (above 300 hPa) along the flight track for flight 19 (August 13, 2015). White contours represent CH$_4$ threshold and background values according to section 3.1.

**Kommentiert [TL52]:** In Figures 7,8, 18-21 and also in Figures 22 and 23 now contour lines are added for the CH4 threshold (1879.8 ppbv) and the CH4 background (1859.4 ppbv) value according to the calculation of the CH4 threshold in section 3.1. In the horizontal CO distribution also the CH4 threshold is added. Now the position of the AMA is easier to identify with respect to the flight tracks.

[Figure]

[Figure]

**Figure 8:** EMAC modelled CO and wind field; daily mean at 204 hPa, and in situ CO (above 300 hPa) along the flight track for flight 19 (August 13, 2015). White contours represent CH$_4$ threshold and background values according to section 3.1.

**Kommentiert [TL53]:** In Figures 7,8, 18-21 and also in Figures 22 and 23 now contour lines are added for the CH4 threshold (1879.8 ppbv) and the CH4 background (1859.4 ppbv) value according to the calculation of the CH4 threshold in section 3.1. In the horizontal CO distribution also the CH4 threshold is added. Now the position of the AMA is easier to identify with respect to the flight tracks.

[Figure]

[Figure]

**Figure 9:** EMAC modelled CH₄; daily mean transect along 23.7°N, and measured CH₄ along the aircraft track for flight 19. Additional EMAC pressure (black lines in hPa) and EMAC northward wind component (blue lines in m/s; southward wind dashed lines).

**Kommentiert [TL54]:** Horizontal wind components are added to the cross sections in figures 9-12, in detail: eastward wind component in cross sections along a longitude and northward wind component in cross sections along a latitude.

[Figure]

[Figure]

**Figure 10:** EMAC calculated CH₄; daily mean transect along 56.2°E, and measured CH₄ along the aircraft track for flight 19. Additional EMAC pressure (black lines in hPa) and EMAC eastward wind component (blue lines in m/s; westward wind dashed lines).

**Kommentiert [TL55]:** Horizontal wind components are added to the cross sections in figures 9-12, in detail: eastward wind component in cross sections along a longitude and northward wind component in cross sections along a latitude.

[Figure]

Figure 11: EMAC modelled CO; daily mean transect along 23.7°N, and CO measured along the aircraft track for flight 19.

**Kommentiert [TL56]:** After reviewing the manuscript with respect to the goal of the paper, Figures 11 and 12 (vertical transect of CO) are removed.

**Kommentiert [TL57]:** Horizontal wind components are added to the cross sections in figures 9-12, in detail: eastward wind component in cross sections along a longitude and northward wind component in cross sections along a latitude.

[Figure]

Figure 12: EMAC modelled CO; daily mean transect along 56.2°E, and CO measured along the aircraft track for flight 19.

**Kommentiert [TL58]:** After reviewing the manuscript with respect to the goal of the paper, Figures 11 and 12 (vertical transect of CO) are removed.

**Kommentiert [TL59]:** Horizontal wind components are added to the cross sections in figures 9-12, in detail: eastward wind component in cross sections along a longitude and northward wind component in cross sections along a latitude.

[Figure]

number of parcels in PBL height per grid cell

[Figure]

**Figure 13:** Last boundary layer contact of parcels from trajectories, which start along the flight track at locations with CH$_4$ mixing ratios above the threshold, before they were transported to the track of flight 19 (10 days prior to flight).

**Kommentiert [TL60]:** Explanation added in figure caption.

[Figure]

[Figure]

**Figure 124**: EMAC calculated CO; daily mean at the surface (1008 hPa, August 03, 2015) as an indicator for surface emissions (10 days prior to measurement) and the flight track of flight 19 (black). Additionally, the footprint of the last boundary layer contact as white contour line from Figure 11.

**Kommentiert [TL61]:** Footprint is now added as white contour lines for the number of particles per grid cell = 2 to the surface emission charts for CH4 and CO (Figures 14 and 15).
.

[Figure]

[Figure]

**Figure 15**: EMAC calculated CH$_4$; daily mean at the surface (1008 hPa, August 03, 2015) as an indicator for surface emissions (10 days prior to measurements), and the track of flight 19 **(black). Additionally, the footprint of the last boundary layer contact as white contour line from Figure 11.**

**Kommentiert [TL62]:** Footprint is now added as white contour lines for the number of particles per grid cell = 2 to the surface emission charts for CH4 and CO (Figures14 and 15).
.

[Figure]

**Figure 146: Histogram for in situ measured and EMAC modelled CH₄ within the altitude range 300-140 hPa, both for background and AMA air.**

[Figure]

**Figure 157: Histogram for in situ measured and EMAC modelled CO within the altitude range 300-140 hPa, both for background and AMA air.**

**Table 1: CH$_4$ and CO averages and standard deviations for in situ measured and EMAC data, both for background and monsoon influenced air masses according to the CH$_4$ threshold for altitudes between 300-140 hPa.**

| p=[300-140] hPa | | CH$_4$ [ppbv] | | CO [ppbv] | |
|---|---|---|---|---|---|
| | | avg | std | avg | std |
| monsoon | in situ | 1910.0 | 19.2 | 96.9 | 10.0 |
| | EMAC | 1874.4 | 15.3 | 99.0 | 11.9 |
| background | in situ | 1837.9 | 27.6 | 76.8 | 15.7 |
| | EMAC | 1850.5 | 21.2 | 84.3 | 15.1 |

[Figure]

[Figure]

**Figure 168:** Double anticyclone mode illustrated with wind field and CH$_4$ EMAC daily means at 204 hPa (July 25, 2015) and the associated flight tracks. White contours represent CH$_4$ threshold and background values according to section 3.1.

**Kommentiert [TL63]:** In Figures 7,8, 18-21 and also in Figures 22 and 23 now contour lines are added for the CH4 threshold (1879.8 ppbv) and the CH4 background (1859.4 ppbv) value according to the calculation of the CH4 threshold in section 3.1. In the horizontal CO distribution also the CH4 threshold is added. Now the position of the AMA is easier to identify with respect to the flight tracks.

[Figure]

[Figure]

**Figure 179:** **Central anticyclone mode illustrated with wind field and CH$_4$ EMAC daily means at 204 hPa (August 09, 2015) and the associated flight tracks.** **White contours represent CH$_4$ threshold and background values according to section 3.1.**

**Kommentiert [TL64]:** In Figures 7,8, 18-21 and also in Figures 22 and 23 now contour lines are added for the CH4 threshold (1879.8 ppbv) and the CH4 background (1859.4 ppbv) value according to the calculation of the CH4 threshold in section 3.1. In the horizontal CO distribution also the CH4 threshold is added. Now the position of the AMA is easier to identify with respect to the flight tracks.

[Figure]

[Figure]

**Figure 18:** Tibetan anticyclone mode illustrated with wind field and CH₄ EMAC daily means at 204 hPa (August 15, 2015) and the associated flight tracks. White contours represent CH₄ threshold and background values according to section 3.1.

**Kommentiert [TL65]:** In Figures 7,8, 18-21 and also in Figures 22 and 23 now contour lines are added for the CH4 threshold (1879.8 ppbv) and the CH4 background (1859.4 ppbv) value according to the calculation of the CH4 threshold in section 3.1. In the horizontal CO distribution also the CH4 threshold is added. Now the position of the AMA is easier to identify with respect to the flight tracks.

[Figure]

[Figure]

Figure 1921: **Double anticyclone mode illustrated with wind field and CH₄ EMAC daily means at 204 hPa (August 25, 2015) and the associated flight tracks. White contours represent CH₄ threshold and background values according to section 3.1.**

**Kommentiert [TL66]:** In Figures 7,8, 18-21 and also in Figures 22 and 23 now contour lines are added for the CH4 threshold (1879.8 ppbv) and the CH4 background (1859.4 ppbv) value according to the calculation of the CH4 threshold in section 3.1. In the horizontal CO distribution also the CH4 threshold is added. Now the position of the AMA is easier to identify with respect to the flight tracks.

Table 2: In situ CO and CH$_4$ for the four different anticyclone situations. Differentiation between AMA and background for each flight between 300-140 hPa.

| meteorological situation | flight no. | date | position relative to AMA | in situ at 300-140 hPa | | | | | | | |
|---|---|---|---|---|---|---|---|---|---|---|---|
| | | | | CO [ppbv] | | | | CH$_4$ [ppbv] | | | |
| | | | | background | sigma | monsoon | sigma | background | sigma | monsoon | sigma |
| double anticyclone | #8 | 21.07.2015 | partly in the western AMA | 67.8 | 8.7 | 89.8 | 7.4 | 1847.1 | 12.3 | 1898.6 | 7.8 |
| | #9 | 25.07.2015 | in the western AMA | 83.1 | 9.4 | 94.5 | 6.1 | 1870.0 | 11.4 | 1913.7 | 16.7 |
| | #10 | 28.07.2015 | in the western AMA | 76.1 | 16.4 | 91.4 | 5.1 | 1856.4 | 24.8 | 1896.4 | 12.4 |
| | #11 | 01.08.2015 | partly in residuals of the AMA | 92.8 | 6.8 | 108.6 | 4.5 | 1823.5 | 21.0 | 1889.0 | 4.8 |
| - | - | - | - | 80.0 | 10.3 | 96.1 | 5.8 | 1849.3 | 17.4 | 1899.4 | 10.4 |
| central mode | #12/13 | 06.08.2015 | in outflow region | 78.6 | 33.3 | 117.3 | 22.2 | 1827.4 | 26.8 | 1893.5 | 9.8 |
| | #14 | 08.08.2015 | in background south of the AMA | 76.3 | 8.0 | - | | 1788.2 | 9.2 | - | - |
| | #15/16 | 09.08.2015 | at the south western edge | 77.5 | 12.0 | - | | 1812.6 | 34.3 | - | - |
| | #17/18 | 10.08.2015 | at the south eastern edge and in outflow region | 76.5 | 7.9 | 98.3 | 7.8 | 1832.0 | 19.5 | 1909.3 | 15.0 |
| - | - | - | - | 77.2 | 15.3 | 107.8 | 15.0 | 1815.1 | 22.5 | 1901.4 | 12.4 |
| Tibetan mode | #19 | 13.08.2015 | at the western edge of the AMA | 74.7 | 10.4 | 99.4 | 13.8 | 1848.0 | 16.3 | 1907.3 | 20.8 |
| | #20 | 15.08.2015 | at the western edge of the AMA | - | - | - | | 1855.2 | 11.6 | 1905.2 | 13.9 |
| | #21 | 18.08.2015 | in and outside the AMA | 87.9 | 16.3 | 104.8 | 9.8 | 1853.0 | 12.9 | 1917.1 | 20.6 |
| - | - | - | - | 81.3 | 13.4 | 102.1 | 11.8 | 1852.1 | 13.6 | 1909.9 | 18.4 |
| double anticyclone | #22 | 23.08.2015 | at the western edge of the western AMA | - | - | - | | 1857.0 | 8.2 | 1927.9 | 22.6 |
| | #23 | 25.08.2015 | at the western edge of the western AMA | 65.7 | 12.4 | 93.8 | 7.6 | 1855.9 | 8.5 | 1926.4 | 21.0 |
| | #24 | 27.08.2015 | outside the AMA | - | | - | | 1853.7 | 14.6 | 1889.1 | 8.8 |
| - | - | - | - | 65.7 | 12.4 | 93.8 | 7.6 | 1855.5 | 10.4 | 1914.4 | 17.5 |

Kommentiert [TL67]: In table 2 a column is added, to describe the relative position of the flight tracks to the position of the AMA.

[Figure]

[Figure]

**Figure 202:** EMAC calculated CH$_4$ and wind field; daily means at 204 hPa, and in situ CH$_4$ (above 300 hPa) along the aircraft track for flight 12/13 (August 06, 2015). White contours represent CH$_4$ threshold and background values according to section 3.1. The enhanced CH$_4$ values over Oman represent the outflow.

**Kommentiert [TL68]:** In Figures 7,8, 18-21 and also in Figures 22 and 23 now contour lines are added for the CH4 threshold (1879.8 ppbv) and the CH4 background (1859.4 ppbv) value according to the calculation of the CH4 threshold in section 3.1. In the horizontal CO distribution also the CH4 threshold is added. Now the position of the AMA is easier to identify with respect to the flight tracks.

**Kommentiert [TL69]:** The selection is made by the outflowing air masses with help of the enhanced CH4 in situ measurements and the location of outflow according to the model data (Figure 22 and 23). Position of the outflow is also added in the Figure captions.

[Figure]

[Figure]

**Figure 23:** EMAC calculated CH$_4$ and wind field; daily means at 204 hPa, and in situ measured CH$_4$ (above 300 hPa) along the aircraft track for flight 17/18 (August 10, 2015). White contours represent CH$_4$ threshold and background values according to section 3.1. The enhanced CH$_4$ values over the Read Sea represent the outflow.

**Kommentiert [TL70]:** In Figures 7,8, 18-21 and also in Figures 22 and 23 now contour lines are added for the CH4 threshold (1879.8 ppbv) and the CH4 background (1859.4 ppbv) value according to the calculation of the CH4 threshold in section 3.1. In the horizontal CO distribution also the CH4 threshold is added. Now the position of the AMA is easier to identify with respect to the flight tracks.

**Kommentiert [TL71]:** The selection is made by the outflowing air masses with help of the enhanced CH4 in situ measurements and the location of outflow according to the model data (Figure 22 and 23). Position of the outflow is also added in the Figure captions.

[Figure]

**Figure 22**: Flight 12/13 (August 6, 2015) in situ CH$_4$ and CO data and EMAC results along the flight track, as well as the flight altitude. The AMA is colour coded by CH$_4$>1879.8 ppbv.

**Kommentiert [TL72]:** Model and in situ data are color coded with similar color for the same molecule in Figure 24 and 25, and consequently also in Figure 3 and in the supplement.

[Figure]

**Figure 23: Flight 17/18 (August 10, 2015) in situ CH$_4$ and CO data and EMAC results along the flight track, as well as the flight altitude. The AMA is colour coded by CH$_4$>1879.8 ppbv.**

[Figure]

**Figure 244: Centroid back trajectories for enhanced CH$_4$ mixing ratios during flight 12/13 (triangles) and flight 17/18 (circles) with colour coded height.**

[Figure]

**Figure 25: Centroid back trajectories for enhanced CH₄ mixing ratios during flight 12/13 (triangles) and flight 17/18 (circles) with colour coded days before the release on August 10, 2015.**

**Supplementary Information for**
**"Upper tropospheric CH$_4$ and CO affected by the Indian summer monsoon during OMO"**

by
Laura Tomsche, Andrea Pozzer, Narendra Ojha, Uwe Parchatka, Jos Lelieveld, Horst Fischer

**Table S1: Uncertainty for CO and CH$_4$ for all flights during OMO.**

| flight no. | date | CO uncertainty [%] | CH$_4$ uncertainty [%] |
|---|---|---|---|
| 8 | 7.21.2015 | 3.52 | 0.33 |
| 9 | 7.25.2015 | 3.47 | 0.34 |
| 10 | 7.28.2015 | 3.23 | 0.33 |
| 11 | 8.1.2015 | 3.41 | 0.23 |
| 12/13 | 8.6.2015 | 19.42 | 0.26 |
| 14 | 8.8.2015 | 4.64 | 0.24 |
| 15/16 | 8.9.2015 | 4.08 | 0.19 |
| 17/18 | 9.10.2015 | 3.78 | 0.21 |
| 19 | 8.13.2015 | 7.89 | 0.34 |
| 20 | 8.15.2015 | | 0.19 |
| 21 | 8.18.2015 | 5.26 | 0.21 |
| 22 | 8.23.2015 | | 0.20 |
| 23 | 8.25.2015 | 3.61 | 0.31 |
| 24 | 8.27.2015 | | 0.23 |

Detailed overview of all OMO flights for p>300 hPa:

a) CO and CH$_4$ in situ and EMAC data along the flight track. The AMA is colour coded due to c[CH$_4$]>=1879.8 ppb (yellow). Further the deviation between EMAC and in situ data are shown for CO and CH$_4$. Additionally, the flight altitude is in grey.

b) 204 hPa EMAC data for CO and wind field and in situ CO along the flight track. White contours represent CH$_4$ threshold and background values according to section 3.1.

c) 204 hPa EMAC data for CH$_4$ and wind field and in situ CH$_4$ along the flight track. White contours represent CH$_4$ threshold and background values according to section 3.1.

d) 10 -day back centroid trajectories for 10 min releases along the flight track (black); colour coded is the altitude in hPa. Triangles are back trajectories for CH$_4$ mixing ratios above the CH$_4$ threshold and circles for below the CH$_4$ threshold.

**Kommentiert [TL1]:** In Table 2 we add a column for the relative position to the AMA, which is quite descriptive. As most of the flight tracks are in and outside the AMA a more detailed geographical location with respect to the AMA can be realized better in a graphical way. Thus we added for each flight in the supplement the CH$_4$ threshold (1879.8ppbv) for the AMA-influence and the background value (1859.4ppbv) as contour lines in the EMAC CH$_4$ and CO distributions as already done in the manuscript, e.g. Figure 7 and 8 for flight 19. In these plots the position of the flight track with respect to the AMA is more obvious.

**Kommentiert [TL2]:** The back trajectories for AMA-influence and background are distinguished by different symbols with an explanation in the caption in Figure 4 and 5, and also in the trajectory plots in the supplement.

[Figure]

a)

[Figure]

[Figure]

b)

[Figure]

c)

[Figure]

d)

5    **Figure S1: Flight 08 (07.21.2015): transfer flight from Oberpfaffenhofen to Paphos.**

[Figure]

a)

[Figure]

[Figure]

b)

[Figure]

[Figure]

c)

[Figure]

d)

**Figure S2: Flight 09 (07.25.2015): measurement flight from Paphos to Paphos over Cyprus.**

[Figure]

a)

[Figure]

[Figure]

b)

[Figure]

[Figure]

c)

[Figure]

d)

**Figure S3: Flight 10 (07.28.2015): measurement flight from Paphos to Paphos over Cyprus.**

[Figure]

a)

[Figure]

[Figure]

b)

[Figure]

[Figure]

c)

[Figure]

d)

**Figure S4: Flight 11 (08.01.2015): transfer flight from Paphos to Gan.**

[Figure]

a)

[Figure]

[Figure]

[Figure]

[Figure]

c)

[Figure]

d)

**Figure S5: Flight 12/13 (08.06.2015): measurement flight from Gan to Bahrain and return to Gan.**

[Figure]

a)

[Figure]

[Figure]

b)

[Figure]

c)

[Figure]

d)

**Figure S6: Flight 14 (08.08.2015): measurement flight from Gan to Gan towards Sri Lanka.**

[Figure]

a)

[Figure]

[Figure]

b)

[Figure]

[Figure]

[Figure]

d)

**Figure S7: Flight 15/16 (08.09.2015): measurement flight from Gan to Bahrain and return to Gan.**

[Figure]

a)

[Figure]

[Figure]

b)

[Figure]

[Figure]

c)

[Figure]

d)

**Figure S8: Flight 17/18 (08.10.2015): transfer flight from Gan to Paphos via refuelling stop in Bahrain.**

[Figure]

a)

[Figure]

[Figure]

b)

[Figure]

c)

[Figure]

d)

**Figure S9: Flight 19 (08.13.2015): measurement flight from Paphos to Paphos towards Oman.**

[Figure]

a)

[Figure]

[Figure]

b)

[Figure]

[Figure]

d)

**Figure S10: Flight 20 (08.15.2015): measurement flight from Paphos to Paphos towards Oman.**

[Figure]

a)

[Figure]

[Figure]

b)

[Figure]

[Figure]

c)

[Figure]

d)

**Figure S11: Flight 21 (08.18.2015): measurement flight from Paphos to Paphos towards Oman.**

[Figure]

a)

[Figure]

[Figure]

b)

[Figure]

[Figure]

c)

[Figure]

d)

**Figure S12: Flight 22 (08.23.2015): measurement flight from Paphos to Paphos over Egypt, Greece, the Mediterranean; with profiles over Egypt.**

[Figure]

a)

[Figure]

[Figure]

b)

[Figure]

c)

[Figure]

d)

**Figure S13: Flight 23 (08.25.2015): measurement flight from Paphos to Paphos over Egypt, Etna, the Mediterranean; with profiles over Egypt and low altitude at the Etna.**

[Figure]

a)

[Figure]

[Figure]

b)

[Figure]

[Figure]

c)

[Figure]

d)

**Figure S14: Flight 24 (08.27.2015): transfer flight from Paphos to Oberpfaffenhofen via Etna with low altitude.**